**Subject Category:**
Biology (whole organism)

ecology

conservation, Ethiopia, fluctuating asymmetry, ptilochronology, shade coffee farming

**Author for correspondence:**
Gelaye Gebremichael
e-mail: gelayegmd@gmail.com

# Fluctuating asymmetry and feather growth bars as biomarkers to assess the habitat quality of shade coffee farming for avian diversity conservation

Gelaye Gebremichael[1,2], Diress Tsegaye[3], Nils Bunnefeld[4,5], Dietmar Zinner[6,7] and Anagaw Atickem[6]

[1]Terrestrial Ecology Unit (TEREC), Ghent University, K.L. Ledeganckstraat 35, 9000 Ghent, Belgium
[2]College of Natural Sciences, Jimma University, PO Box 378, Jimma, Ethiopia
[3]Department of Biosciences, University of Oslo, Postboks 1066 Blindern, 0316 Oslo, Norway
[4]Biological and Environmental Sciences, Faculty of Natural Sciences, University of Stirling, Stirling FK9 4LA, UK
[5]School of Geosciences, University of Edinburgh, Edinburgh EH9 3JN, UK
[6]Cognitive Ethology Laboratory, German Primate Center, Leibniz Institute for Primate Research Kellnerweg 4, 37077 Göttingen, Germany
[7]Leibniz ScienceCampus Primate Cognition, Göttingen, Germany

GG, 0000-0003-3960-6088; DZ, 0000-0003-3967-8014

Shade coffee farming has been promoted as a means of combining sustainable coffee production and biodiversity conservation. Supporting this idea, similar levels of diversity and abundance of birds have been found in shade coffee and natural forests. However, diversity and abundance are not always good indicators of habitat quality because there may be a lag before population effects are observed following habitat conversion. Therefore, other indicators of habitat quality should be tested. In this paper, we investigate the use of two biomarkers: fluctuating asymmetry (FA) of tarsus length and rectrix mass, and feather growth bars (average growth bar width) to characterize the habitat quality of shade coffee and natural forests. We predicted higher FA and narrower feather growth bars in shade coffee forest versus natural forest, indicating higher quality in the latter. We measured and compared FA in tarsus length and rectrix mass and average growth bar width in more than 200 individuals of five bird species. The extent of FA in both tarsus

length and rectrix mass was not different between the two forest types in any of the five species. Similarly, we found no difference in feather growth between shade coffee and natural forests for any species. Therefore, we conclude our comparison of biomarkers suggests that shade coffee farms and natural forests provide similar habitat quality for the five species we examined.

## 1. Introduction

Tropical forests host at least two-thirds of the world's terrestrial biodiversity and offer a wide range of ecosystem services [1]. However, these forests are shrinking at unprecedented rates [2,3]. Agricultural practices compatible with forest conservation have been proposed as a potential strategy to slow down, stop or even reverse the current alarming rate of deforestation [4].

Shade coffee farming, i.e. growing coffee beneath the canopy of shade trees, has recently gained support as a form of agricultural practice that is, at least partly, more compatible with biodiversity conservation than other agricultural practices [5–7]. However, the results of various studies testing this notion are contradictory. For instance, species richness, abundance and community composition of birds in shade coffee forests have been shown to be similar to those in natural forests in some studies [8–10], but not in others [11–14]. But compared with nearby open-sun plantation, bird species diversity in shade coffee forests is notably higher [9,12,13]. Therefore, shade coffee farming might be an important method to preserve biodiversity in agro-forestry systems [15,16].

These studies imply that populations inhabiting shade coffee forests probably also share equal prospects for reproduction or/and long-term survival with their conspecifics residing in pristine forests [17,18]. However, more subtle effects on populations may have been overlooked so far, including demographic changes in the distribution of sex or age classes or changes in rates of survival or reproduction. Such demographic changes are unlikely to be identified by community-level studies on demography and α-diversity alone if surveys are done soon after natural forests have been transformed into shade coffee forests. Communities can be affected without any notable change in diversity and abundance due to time lags in responses or immigration from nearby source populations [19,20]. To overcome these issues, a growing number of studies applied phenotypic or physiological proxies of fitness (biomarkers), such as body size, body mass, fat-free mass, or the rate or precision of growth. These markers are all thought to be able to indicate stress (e.g. nutritional stress), and thus relative levels of habitat quality [21,22]. Stress-mediated changes should preferably be detectable before direct components of fitness are compromised (see [23] for a list of criteria to evaluate biomarkers). Most importantly in a conservation context, collecting such data should not involve lethal sampling, exhaustive training or expensive equipment.

Two potential useful biomarkers are fluctuating asymmetry (FA), i.e. measuring small random deviations from left–right symmetry in bilateral traits [24–27] and feather growth bars, i.e. measuring width of alternating dark and light growth-bars on bird feathers (ptilochronology, [28–31]). While there are exceptions [32], studies have shown that asymmetric individuals exhibit lower fitness affecting populations in the long term [33–37].

For example, in a study of barn swallows (*Hirundo rustica*), Møller [38] reported a significant increase in male tail feather FA in birds captured in Chernobyl after the 1986 nuclear accident compared with pre-1986 museum specimens from the same site. Furthermore, Møller [38] indicated that more asymmetric males of this species bred later than symmetric ones. Also, birds living in low-quality sites (food less abundant) are expected to show narrower growth bars than birds in sites with higher food availability [28,29,39]. Carlson [39] demonstrated a positive correlation between feather growth bar widths with food availability in the territories of white-backed woodpeckers (*Dendrocopos leucotos*). Although the use of growth bars as an index of nutritional condition in wild birds seems plausible and has been used in many studies (e.g. [28,39–41]), others have criticized the validity of this technique for the lack of precision [42,43].

The benefits of these markers are based on their ease, low cost, non-lethality during the measurement process, and their link to fundamental life-history components that are otherwise more laborious to measure [34]. When applied in tandem [44,45], they provide an overall, integrated picture of environmental and nutritional stress effects [21], which in turn can provide information on habitat quality.

Ethiopian coffee mainly grows in tropical rainforest remnants belonging to the Eastern Afromontane biodiversity hotspot. This hotspot is known for its exceptionally high level of biodiversity and endemicity

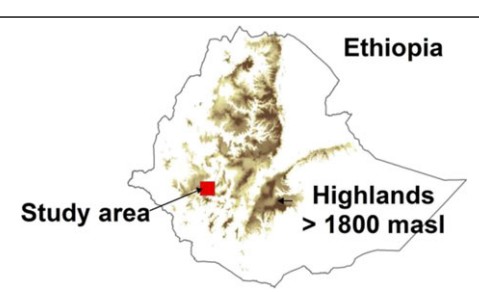

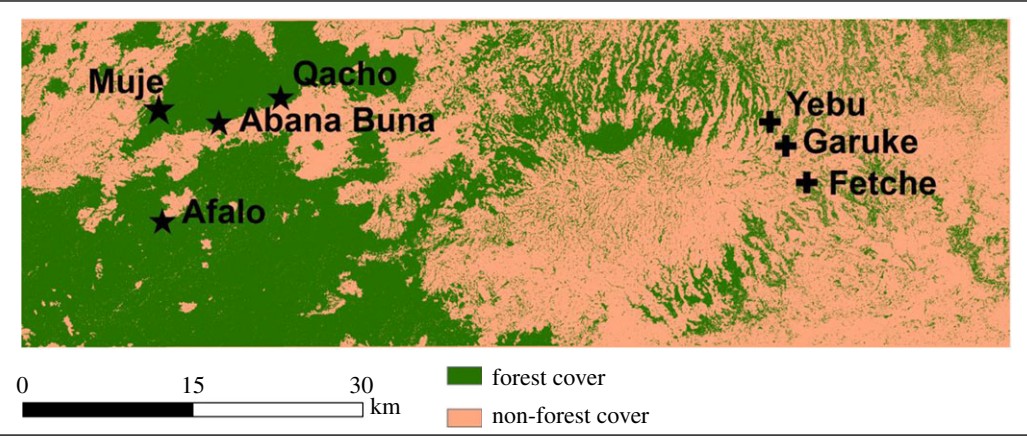

**Figure 1.** Geographical positions of study sites in natural forests (Muje, Afalo, Ababa Buna and Qacho) and in the shade coffee forests (Yebu, Garuke, Fetche) in the highlands of western Ethiopia.

[46]. The respective forests are also the origin and reservoir of wild *Coffea arabica* [47,48]. Buechley *et al*. [49] showed that bird diversity in shade coffee forests in southwestern Ethiopia did not significantly differ from that of ecologically comparable natural moist evergreen Afromontane forests in the same region. The forests used for coffee production actually hosted more than twice the number of species than the latter. While all bird species recorded in the primary forest control sites were also recorded in shade coffee forests, shade coffee plots showed a lower abundance of forest specialists and understorey insectivores. The authors concluded that, while Ethiopian shade coffee farming is perhaps the most 'bird-friendly' coffee in the world, the little-disturbed forest remains critical for sustaining at-risk groups of birds, such as forest specialist birds [49].

To determine whether populations of forest-dependent species that currently persist in Ethiopian shade coffee forest show signs of increased environmental and/or nutritional stress compared with conspecifics from natural forests, we revisited a subset of the study plots sampled by Buechley *et al*. [49] and measured FA of two phenotypic traits (length of tarsi and mass of the second outermost rectrices) and feather growth bars of the second outermost rectrices in 319 individuals belonging to five species. We hypothesized that if shade coffee is of lower quality than natural forests we would observe higher FA and lower feather growth rates (i.e. narrower feather growth bars) in birds from shade coffee forests when compared with birds from undisturbed forests. Studying both traits allowed us to assess possible adverse effects of shade coffee farming during different stages of life. Since tarsi are already fully grown before fledging, this measure provides information on stress during the nestling period [50,51], and probably greatly influences future survival and reproductive success [34,46,52–55]. On the other hand, rectrices are moulted throughout adult life, providing information on stress at a particular time of the adult individuals' life-history.

## 2. Material and method

### 2.1. Study site

The study was conducted in southwestern Ethiopia at three sites in the shade coffee farming area (422 km$^2$): Fetche, Garuke and Yebu, and at four natural forest sites in the Belete-Gera Forest (920 km$^2$): Afalo, Abana Buna, Qacho and Muje (figure 1).

**Table 1.** The number of individual birds assessed for growth bar widths (rectrices masses) and tarsus lengths per species and forest type (natural forest (NF), shade coffee forest (SC)).

| species | common name | growth bar width | | | tarsus length | | |
|---------|-------------|------|------|-------|------|------|-------|
| | | SC | NF | total | SC | NF | total |
| *Muscicapa adusta* | African dusky flycatcher | 10 | 10 | 20 | 21 | 22 | 43 |
| *Camaroptera brevicaudata* | grey-backed camaroptera | 8 | 7 | 15 | 16 | 12 | 28 |
| *Cossypha semirufa* | Rüppell's robin-chat | 66 | 30 | 96 | 85 | 40 | 125 |
| *Turtur tympanistria* | tambourine dove | 37 | 40 | 77 | 47 | 50 | 97 |
| *Zoothera piaggiae* | Abyssinian ground thrush | 5 | 10 | 15 | 13 | 13 | 26 |
| total | | 126 | 97 | 223 | 182 | 137 | 319 |

In the shade coffee production scheme, the forest was modified from the original forest to improve the productivity of coffee plants by reducing tree and shrub density. This leads to obvious differences in the vegetation density and canopy cover compared with the natural forest. In the shade coffee areas, seedling density, stem density and crown cover of indigenous trees were 3000 ha$^{-1}$, 655 ha$^{-1}$ and 79%, respectively, whereas in natural forests respective values were 10 000 ha$^{-1}$, 952 ha$^{-1}$ and 86.7% [56].

The Belete-Gera Forest is one of the largest remaining tracts of forest in southwestern Ethiopia. It is a typical Afromontane evergreen forest, dominated by tree species like *Syzigium guineense*, *Olea welwitchii*, *Prunus africana* and *Pouteria adolfi-friederici*, with over 157 plant species representing 69 families and 135 genera [57,58]. Its animal diversity has never been thoroughly studied, but during a one-month pilot study, 25 mammal, 16 amphibian, 5 reptile, 126 bird and 87 butterfly species have been identified [59]. The forest is considered as a regional forest priority area and 'key biodiversity area' (KBA). Nevertheless, deforestation and habitat modification for shade coffee farming continues in the region.

## 2.2. Study species

The bird species included were chosen if (1) they represented a breeding population in our study area, and (2) their sample sizes from both forest types were adequate and comparable. These criteria were met by five species: Abyssinian ground thrush, Rüppell's robin-chat, grey-backed camaroptera, African dusky flycatcher and tambourine dove (table 1). Based on their primary habitat utilization, the first two species were considered forest species, while the latter three species were considered non-forest species [60–66]. Based on their diet Rüppell's robin-chat, grey-backed camaroptera and African dusky flycatcher were regarded as insectivorous, the Abyssinian ground thrush as omnivorous and the tambourine dove as granivorous [67]. These species are year-round residents in the study area [68], comprising breeding populations [49].

## 2.3. Data collection

From December 2013 to June 2015, we captured 1070 individuals of 64 bird species using mist-netting procedures. We retrieved measures of tarsus length from 319 birds, and measures of rectrix mass and average growth bar width from 223 individuals of our five species (table 1). We opened the nets 30 min before sunrise and kept them open for six hours. We checked the nets routinely at 30 min intervals to remove birds without holding them unnecessarily. Upon capture, we (1) ringed each individual bird with rings from National Museum of Kenya, and aged and weighed them (to the nearest 0.01 g using digital balance), (2) measured tarsus length (to the nearest 0.1 mm with Vernier callipers), and wing length (to the nearest 0.5 mm using wing ruler) and (3) plucked the second outermost (the fifth) of the right and left rectrices feathers from individual captured birds. We released the birds immediately after we collected feather samples and took the measurements on the spot.

### 2.3.1. Measurement of tarsi length and rectrix mass

Fluctuating asymmetry was determined from two traits, tarsi length and rectrix mass. Different traits have different susceptibility for FA based on cost of growth of the trait, defined as the amount of

structural components necessary to form a unit of length of a given character [69] hence we used both traits. We measured tarsi length as the distance from the back of inter-tarsal joint to the point where the toes were bending an angle of 90 degree to the tarsus [70]. Two independent measurements of right and left tarsus lengths for each individual (sequence right-left-right-left or left-right-left-right) were taken. Collected feathers were stored in separately labelled dry paper envelopes and transported to the laboratory, and the rectrices were weighted twice using an analytical balance to the nearest 0.1 mg. Tarsi with anomalies (such as broken or deformed tarsi) were not measured, and rectrices with damaged tips or soiled with faeces were not collected.

### 2.3.2. Measurement of growth bar widths

Growth bar widths and total feather lengths were measured by one of us (GG) as follows: (1) each unwearied feather was pinned on separate, white, polystyrene boards and total feather length was measured to the nearest 0.01 mm with a digital calliper. (2) Each feather was marked at a distance of 7/10 from its proximal end, and the proximate and distal ends of five consecutive growth bars were marked with ultrafine mounting pins. (3) Each marked board was scanned (Oce OP1130) and growth bar widths and total feather lengths were automatically measured with an image analysis software (KS400 Zeiss). The entire procedure was repeated for 66 randomly selected feathers to assess the repeatability of our measurements.

## 2.4. Data validation

Individual birds were caught first as immatures and in subsequent years as adults in the same sites. This indicates that the individual birds have been year-round residents at the respective sites. Therefore, potential effects on the three biomarkers most likely reflect habitat quality and are not disturbed by immigrating birds from other habitats.

Possible data outliers were detected using Cook distance (Cookd) (in SAS) to ensure data quality for all measurements (tarsus length, feather mass, feather length and average growth bar widths), whereby outliers were removed from the data.

The level of repeatability for the growth bar widths was calculated according to Lessells & Boag [71], whereby within-group and between-group mean squares were obtained from one-way ANOVAs, and significance was assessed using $F$-tests. Repeatability estimates were high, for total feather length (right trait side: $r = 0.999$, $N = 66$, $p = <0.001$; left trait side: $r = 0.999$, $N = 66$, $p < 0.001$) and for average growth bar widths (right trait side: $r = 0.992$, $N = 66$, $p < 0.000$; left trait side: $r = 0.983$, $N = 66$, $p < 0.001$).

Since the residuals of the measured traits (unsigned FA values for tarsus length and rectrix mass) deviated from normal (Shapiro–Wilk test; see electronic supplementary material, table S1), the data were transformed using Box–Cox transformation to meet normality assumptions before further analysis. Individual adult birds were used for the subsequent analyses of growth bar width, rectrix mass and rectrix length.

Theoretically, one would expect that the two biomarkers indicate similar trends in habitat quality, i.e. the signed FA values (tarsus length and rectrix mass) should both negatively correlate with average growth bar width. However, we did not find a strong correlation between the two biomarkers (growth bar width versus FA tarsus length: $r = -0.121$, $N = 180$, $p = 0.106$; growth bar width versus FA rectrix mass: $r = -0.073$, $N = 222$, $p = 0.330$).

The lack of strong correlation between the growth bar width and FA tarsus length means the biomarkers used in this study are not equally sensitive for the effects of habitat change.

## 2.5. Data analyses

### 2.5.1. Analysis of fluctuating asymmetry

We quantified individual signed FA for tarsus length and rectrix mass as the difference in measures between right (R) and left (L) tarsi (using average values obtained from the two replicated measurements of each side) [72]. We then calculated the absolute values of differences in measurements (unsigned FA) to describe the magnitude of FA.

Following Van Dongen et al. [73], the levels of significance of signed FA and its repeatability in the measured traits were calculated by the restricted maximum-likelihood (REML) estimation of a mixed regression model where we fitted the models to the repeated measurements of right and left trait

sides [73]. We included sides as fixed effects in the models and measured trait values (tarsi lengths or rectrix masses) as response variables, bird id and sides as a random effect. This procedure allows the separation of measurement error (ME) from bilateral asymmetry analysis [73]. The presence of directional asymmetry was assessed by the F-test of the fixed effects, with degrees of freedom corrected for statistical dependence by Satterthwaite formulae [74]. We used random intercepts and slopes (both estimated within individuals) to estimate the variation in individual trait value and the individual signed FA, respectively, while the random error variance component gave the measurement error. The significance of FA was then calculated by performing a likelihood ratio (LR) test comparing two models: the original, full model and a reduced model without the side (left or right) as a random effect. Then, intra-class correlation coefficients (ICC) were calculated to evaluate the repeatability of FA [73].

The absence of antisymmetry, i.e. a type of asymmetry that occurs in a random direction, producing a bimodal distribution over the population, was verified by testing the signed FA for normal distribution.

### 2.5.2. Comparison of unsigned FA between habitat types

The linear mixed model was fit to the data to test whether unsigned FA levels in tarsus length and rectrix mass differed between birds of shade coffee forests and birds of natural forests. Unsigned FA was included as a response variable, forest type as a fixed effect and sampling location as a random effect.

Model selection was done using Akaike information criterion (AIC) for the species with moderate sample sizes (Rüppell's robin-chat and tambourine dove). For the species with small sample sizes (Abyssinian ground thrush, grey-backed camaroptera and African dusky flycatcher), model selection was done using AICc (the form of AIC corrected for small sample size).

### 2.5.3. Comparison of growth bar width between habitat types

To test whether average growth bar widths differed between birds of shade coffee forests and those of natural forests, we added average growth bar widths in the models as a response variable, forest type as fixed effect, total feather lengths as fixed covariate and sampling location as a random effect. All statistical analyses were performed using SAS (v. 9.4., SAS Institute 2013, Cary, NC, USA).

## 3. Results

We obtained measures of tarsus length from 319 individuals, and measures of rectrix mass and average growth bar width from 223 individuals (table 1). For tarsus length, kurtosis ranged from $K = 0.33$ in tambourine dove to $K = 6.82$ in Abyssinian ground thrush, for rectrix mass, it ranged from $K = 3.551$ in Abyssinian ground thrush to $K = 24.75$ in tambourine dove (electronic supplementary material, table S1). The signed FA distribution was leptokurtic for both tarsus length and rectrix mass in the Abyssinian ground thrush, African dusky flycatcher and grey-backed camaroptera, but for rectrix mass only in Rüppell's robin-chat and tambourine dove ($K > 3.0$) (Shapiro–Wilk test; all $p < 0.03$; electronic supplementary material, table S1), which indicates heterogeneity in developmental stability between individuals [73,75]. Directional asymmetry was detected for tarsus length ($p = 0.0005$), but not for rectrix mass ($p = 0.0967$; electronic supplementary material, table S2).

### 3.1. Fluctuating asymmetry

The signed FA variance estimation was highly significant relative to measurement error in both tarsus length ($\chi^2 = 789.7$, d.f. = 1, $p < 0.0001$) and rectrix mass ($\chi^2 = 449.1$, d.f. = 1, $p < 0.0001$) (electronic supplementary material, table S2). But, the level of within-side measurement error was high compared with between-side difference in both traits (tarsus length: ICC = 22.3% and rectrix mass: ICC = $2.5 \times 10^{-5}$). Furthermore, the level of unsigned FA in tarsus length and rectrix mass did not differ between natural forests and shade coffee forests for four species (all: $p > 0.0521$; tables 2 and 3). Although in Abyssinian ground thrush, the FA in rectrix mass was arguably statically different between shade coffee and natural forest ($p = 0.0521$), the effect size was near zero (estimate = $-9 \times 10^{-4}$; table 3). Although not significant, African dusky flycatcher and tambourine dove demonstrated higher FA in tarsus length and rectrix mass in shade coffee than their conspecifics in natural forest (figure 2a,b). Again, although not significant, Rüppell's robin-chat showed higher FA only in tarsus length in shade coffee forests.

**Table 2.** Unsigned FA estimates of tarsus length for five bird species in two habitat types (natural forest and shade coffee forest). Linear mixed models were fitted to the data where unsigned FA were included in the models as a response variable, forest types as fixed effect and sampling location as random effect for Abyssinian ground thrush (AGT), African dusky flycatcher (ADF), grey-backed camaroptera (GBC), Rüppell's robin-chat (RRC) and tambourine dove (TD).

| species | effect | estimate | s.e. | d.f. | $t$ | $p$-values |
|---|---|---|---|---|---|---|
| AGT | intercept | −2.566 | 0.2491 | 23 | −10.3 | <0.0001 |
| | forest type (natural) | −0.5699 | 0.3454 | 23 | −1.65 | 0.1126 |
| ADF | intercept | 0.4219 | 0.01029 | 17 | 41.0 | <0.0001 |
| | forest type (natural) | −0.0210 | 0.021 | 40 | −1.67 | 0.1031 |
| GBC | intercept | 0.7796 | 0.149 | 28 | 5.24 | <0.0001 |
| | forest type (natural) | −0.0602 | 0.227 | 28 | −0.27 | 0.7926 |
| RRC | intercept | 0.0195 | 0.004 | 10 | 5 | <0.0034 |
| | forest type (natural) | −0.0091 | 0.0085 | 11 | −1.08 | 0.3018 |
| TD | intercept | 3.3257 | 0.3577 | 86 | 9.30 | <0.0001 |
| | forest type (natural) | −0.3899 | 0.4948 | 86 | −0.81 | 0.4225 |

**Table 3.** Unsigned FA estimates in rectrix masses for five bird species in two habitat types (natural forest and shade coffee forest). Linear mixed models were fitted to the data where unsigned FA were included in the models as a response variable, forest types as fixed effect and sampling location as random effect for Abyssinian ground thrush (AGT), African dusky flycatcher (ADF), grey-backed camaroptera (GBC), Rüppell's robin-chat (RRC) and tambourine dove (TD).

| species | effect | estimate | s.e. | d.f. | $t$ | $p$-values |
|---|---|---|---|---|---|---|
| AGT | intercept | $1.51 \times 10^{-5}$ | 0.00037 | 15 | 4.12 | 0.0009 |
| | forest type (natural) | $-9 \times 10^{-4}$ | 0.00045 | 15 | −2.11 | 0.0521 |
| ADF | intercept | $9.68 \times 10^{-4}$ | 0.000568 | 20 | 1.7 | 0.0391 |
| | forest type (natural) | −11.00066 | 0.000733 | 20 | −0.89 | 0.3816 |
| GBC | intercept | $1.35 \times 10^{-6}$ | 0 | 13 | −0.86 | <0.0001 |
| | forest type (natural) | $1.8 \times 10^{-5}$ | $1.4 \times 10^{-6}$ | 13 | −0.86 | 0.4038 |
| RRC | intercept | $7.9 \times 10^{-5}$ | $2.3 \times 10^{-5}$ | 94 | 3.5 | 0.0007 |
| | forest type (natural) | $1.8 \times 10^{-5}$ | $2.7 \times 10^{-5}$ | 94 | 0.67 | 0.5021 |
| TD | intercept | $1.4 \times 10^{-2}$ | 0.004 | 66 | 3.69 | 0.005 |
| | forest type (natural) | $-6.5 \times 10^{-3}$ | 0.005 | 66 | −1.22 | 0.223 |

## 3.2. Growth bar width

Average growth bar width of the five species did not differ between shade coffee and natural forests (all species: $p > 0.14$; table 4). Furthermore, mean rectrix length did not vary between shade coffee and natural forests (all species: $p > 0.15$; electronic supplementary material, table S3). Tambourine doves showed a tendency for wider average growth bar in the natural forests, whereas grey-backed camaroptera showed narrower growth bar in the natural forest (figure 3).

## 4. Discussion

Several studies based on species richness, abundance and community composition have revealed that shade coffee farming can have an important role in biodiversity conservation [8–10,15,16]. Yet, the intensification of coffee management necessitates further monitoring of habitat quality in shade coffee farms with methods that can detect changes before the habitat quality is degraded to such an extent that it affects avian community demography [40,76,77]. Here, we applied biomarkers to re-assess the

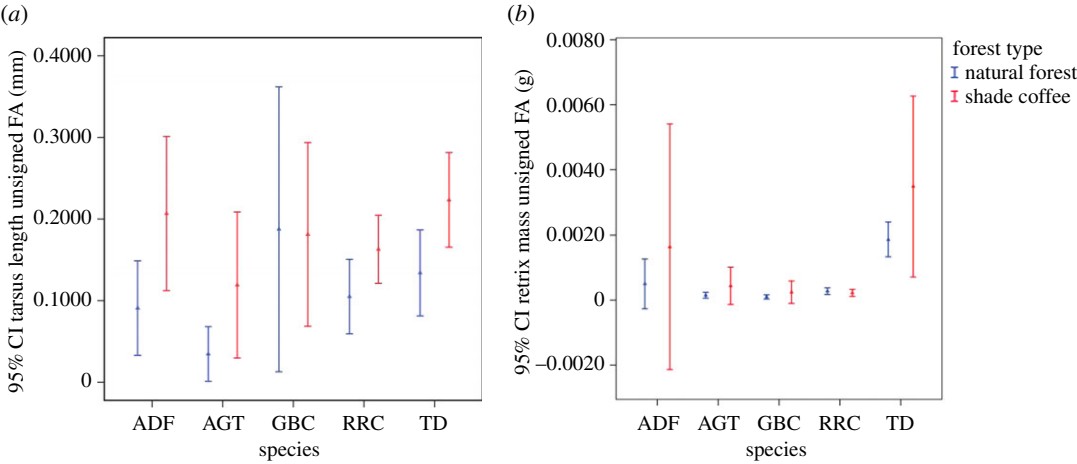

**Figure 2.** Unsigned FA (mean ± s.e.) in five bird species (species: ADF, African dusky flycatcher; AGT, Abyssinian ground thrush; GBC, grey-backed camaroptera, RRC, Rüppell's robin-chat; TD, tambourine dove).

**Table 4.** Estimates of average growth bar width from mixed linear models, where average growth bar widths were included as a response variable, forest types as fixed effect, whereby feather length included as fixed covariate and sampling location as random effect for Abyssinian ground thrush (AGT), African dusky flycatcher (ADF), grey-backed camaroptera (GBC), Rüppell's robin-chat (RRC) and tambourine dove (TD). Natural = natural forest and TFL = mean of the second outermost right and left rectrices.

| species | effect | estimate | s.e. | d.f. | t | p-values |
|---------|--------|----------|------|------|---|----------|
| AGT | intercept | −7.012 | 7.9878 | 13 | 0.880 | 0.3959 |
| | forest type (natural) | 1.456 | 0.9286 | 13 | 1.57 | 0.1409 |
| | TFL | 0.267 | 0.093 | 13 | 2.88 | 0.0128 |
| ADF | intercept | 6.029 | 7.428 | 16 | 0.81 | 0.429 |
| | forest type (natural) | −0.287 | 0.5471 | 16 | −0.53 | 0.429 |
| | TFL | −0.287 | 0.547 | 16 | 0.77 | 0.451 |
| GBC | intercept | −4.687 | 6.44 | 8.42 | −0.73 | 0.486 |
| | forest type (natural) | −0.83 | 1.12 | 6.1 | −0.74 | 0.486 |
| | TFL | 0.38 | 0.105 | 8 | 2.3 | 0.0496 |
| RRC | intercept | 6.70 | 4.9647 | 91 | 2.36 | 0.0206 |
| | forest type (natural) | −5.098 | 4.9647 | 88 | −1.1 | 0.276 |
| | TFL | 0.098 | 0.0443 | 91 | 2.22 | 0.0292 |
| | natural × TFL | 0.088 | 0.0776 | 88 | 2.0 | 0.2574 |
| TD | intercept | −0.316 | 0.138 | 66 | 1.87 | 0.979 |
| | forest type (natural) | 0.984 | 5.038 | 69 | 0.07 | 0.9480 |
| | TFL | 0.2582 | 0.138 | 65 | 1.87 | 0.066 |
| | natural × TFL | −0.0163 | 0.175 | 69 | −0.09 | 0.92 |

quality of Ethiopian tropical rainforest used for shade coffee farming which has previously been reported to support as rich avian biodiversity as natural forest based on demographic studies [49].

FA in both traits (tarsus length and rectrix mass) was not significantly different in birds from shade coffee or natural forests. Similarly, the analysis of feather growth bar widths revealed no significant difference between birds of the two forest types. Although we did not find significant differences between the habitats, there is a certain tendency for such differences. Hence, we should not rule out that shade coffee farming has no negative impact on habitat quality for the studied bird species. A follow-up study with more individuals over a longer time period will probably find whether it is a

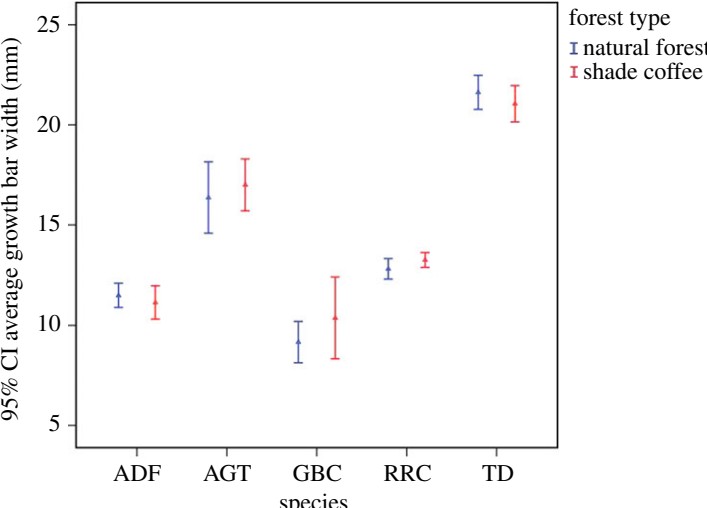

**Figure 3.** Average growth bar width (mean ± s.e.) in five bird species (species: ADF, African dusky flycatcher; AGT, Abyssinian ground thrush; GBC, grey-backed camaroptera, RRC, Rüppell's robin-chat; TD, tambourine dove).

real effect or not. Our results are in contrast to Buechley *et al.* [49] who reported that forest specialist guilds and understorey insectivores were more negatively affected than birds of other guilds in shade coffee when compared with those in natural forest.

Based on the biomarkers we compared, it appears that shade coffee farms and natural forests are of similar quality for the five species we examined. Yet, the long-term impact of the on-going shade coffee plantation modification and management should still be closely monitored. Shade coffee farm management, involving the selective removal of certain tree species, may decrease habitat quality of shade coffee plantation for avian biodiversity [5,78]. In many coffee growing countries across the globe, shade coffee management practices are intensifying, which is believed to be reducing plant diversity and canopy cover [77,79–81], and hence, also reducing the quality or quantity of feeding, nesting or hiding resources for animal populations. Such tendencies underline the need for close monitoring of habitat quality of shade coffee forests, for example, with biomarkers such as FA and feather growth bars, which can detect changes of habitat quality before it is degraded to the extent that it affects avian diversity and community demography [18,40,82,83]. Globally, about 21% of bird species are currently at risk of extinction and 6.5% are functionally extinct [84], underlining the urgency of monitoring efforts.

The lack of strong correlation between the growth bar width and FA tarsus length means the biomarkers used in this study are not equally sensitive for the effects of habitat change. Only a few studies combined FA and feather growth bar which reported consistent result from both markers or only one of the markers revealed the effect of the habitat change [25,33,85]. We recommend future studies combine the use of multiple biomarkers to robustly assess the habitat quality following habitat conversion.

## 5. Conclusion

Our results are consistent with results of many of the demographic studies that shade coffee farming in the Ethiopian highland can be compatible with avian diversity conservation. Similar to these and other biomarker studies, we found no early sign of negative effect was observed from biomarker studies. Buechley *et al.* [49] showed that shade coffee farming can contribute to avian biodiversity conservation in coffee agroecosystem, though forest specialist birds were disproportionally negatively affected in shade coffee farms. Conversely, biomarkers for the two forest species in our study (Abyssinian ground thrush and Rüppell's robin-chat), were not different between shade coffee and natural forests. While the current biomarker-based study did not reveal any sign of a negative effect of shade coffee, it is important to continue evaluating the avian biodiversity conservation value of Ethiopian shade coffee as shade coffee plantations are still under pressure of modification and subject to increasing levels of

fragmentation. We also recommend combining FA and feather growth bars as biomarkers to provide complementary measures of habitat quality for avian biodiversity conservation.

Ethics. This project was carried out in accordance with the ethical standards for research from Jimma University, Ethiopia and Ethiopian Wildlife Conservation Authority (EWCA), and the project was endorsed by EWCA.
Data accessibility. Data available at the Dryad Digital Repository: http://dx.doi.org/10.5061/dryad.cb0d8ft [86].
Authors' contributions. G.G. did the fieldwork and collected the data. G.G. and A.A. drafted a first version of the manuscript which was improved by D.Z. The statistical analyses were done by G.G., D.T. and N.B. All authors reviewed the manuscript and gave their final approval for publication.
Competing interests. We declare we have no competing interests.
Funding. This research is funded by the VLIR IUC-JU project (VLIR-UOS Institutional University Cooperation programme between Jimma University and various Flemish universities under the umbrella of the Flemish Interuniversity Council).
Acknowledgements. We are grateful to Bahir and Reshad Abafita and Jewad Abazinab for assisting us during our fieldwork and all VLIR IUC-JU divers. We thank Luc Lens for important comments on improving the proposal and designing the fieldwork. We also thank the reviewers for their valuable comments and Lynsey Bunnefeld for the help with the language; Kasahun Eba at VLIR IUC-JU programme office at Jimma University for his support throughout our fieldwork period, and the Alexander von Humboldt Foundation for the support given to Anagaw Atickem.

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
