## [Reviewer comments · Royal Society Open Science]

Review History

RSOS-180982.R0 (Original submission)

Review form: Reviewer 1 (Evan Buechley)

Is the manuscript scientifically sound in its present form?

No

Are the interpretations and conclusions justified by the results?

No

Is the language acceptable?

Yes

Is it clear how to access all supporting data?

Yes

Do you have any ethical concerns with this paper?

No

Have you any concerns about statistical analyses in this paper?

I do not feel qualified to assess the statistics

Recommendation?

Major revision is needed (please make suggestions in comments)

Comments to the Author(s)

Overview:

This is a very interesting concept, applying morphometrics and fluctuating asymmetry in a fairly novel fashion to study potential impacts of environmental quality of shade coffee and natural forest on bird communities in SW Ethiopia. The study is novel and has the potential to make a significant and important contribution to biodiversity conservation in Ethiopia, as well as to encourage similar studies worldwide to evaluate early indicators of environmental stresses on populations. On this level, I think it is suitable for publication in Royal Society Open Science.

However, I also have some significant methodological concerns with the manuscript that I hope can be addressed prior to publication. My biggest concern is regarding sample size and guild classifications, as best outlined in Table 1. Is it really suitable to compare fluctuating asymmetry and feather growth bar width of different species, as lumped into very general guilds? The difference in samples sizes of species between shade coffee and natural forest for growth bar width measurements seems particularly great. For example the top 4 species were almost exclusively sampled in shade coffee. Thus how can you compare differences between habitats??? I'm not sure that lumping them into guilds addresses this underlying issue in the dataset. If you have a strong argument for why this is valid to compare, I think it should be very prominently laid out in the methods and discussion. As is, there is really almost no discussion of such lack of uneven sampling and how that might impact results. Otherwise, perhaps the analysis should be limited to species that you have a reasonable sample size in both habitat types...

I also have a bit of a concern with the guild classification for obligate insectivore vs opportunistic insectivore, which is based on a seemingly arbitrary 80% insect-diet cutoff. Is there no way to make a more meaningful guild distinction, perhaps based on the classifications in Buechley et al 2015 (i.e. insectivore / granivore / frugivore).

A couple of other fairly major comments are as follows:

- English was good throughout, up until the discussion, which could use some detailed work simplifying and correcting grammatical errors.
- Discussion has several bold claims about coffee forests in Ethiopia that I strongly suggest toning back. This is an interesting study, but I don't think it is definitive enough to say things like "the negative effects of converting natural forest into shade coffee forests on bird communities are not dramatic [line 294]"

More detailed points here:

Line 71: "random" – if they are random, then how can they be an index of anything?

Line 78: What about using other bio-markers like mass?

Lines 81-82: "the origin and reservoir" – i take issue with the wording here, as arabica also originated and is still found in SW Arabia

Line 94: tarsis length can be a hard measurement to standardize – perhaps address how this was done in methods

Line 95: Mass of the retrace? This seems an odd measurement!

Line 97: replace assumed with hypothesized

Line 118: comma spacing

Line 123: has this directly impacted the study sites? If so how?

Line 134: this 80% cutoff seems arbitrary. Could you not find a more well-defined guild separation, perhaps by following the definitions used in Buechley et al 2015?

Line 143: clarify where the rings are from (i.e. museum of nat hist)

Line 144: is this the only reason?

Table 1: I'm concerned about unbalanced sampling of the study species between habitat types. Is it valid to compare fluctuating asymmetry / feather growth across species?

Lines 154 – 158: I'm a bit skeptical about this method. Retrix feathers are often soiled by feces. Were the feathers cleaned prior to weighing? If so, how? Retrix feathers also undergo major wear. It is not clear to me that tropical birds would necessarily have the same moult pattern, thus different individuals may be under different levels of feather wear at any given time, which could impact feather mass. I strongly suggest adding more detail of methods and citations validating these indices here.

Line 162: “unwearied” – what does this mean??

Line 194: I like that repeatability of the measurements was included in the model

Line 231: spell out acronyms (e.g. FF, DA, etc) at start of new section

Lines 241 – 245: was this caused by scale accuracy? If the scale is accurate, it seems this would be the easier measurement to replicate. Can you explain this result more, perhaps in Discussion?

Lines 252-254: show p value for this comment

Lines 259 – 260: I would cut this sentence off at “natural forest”. It was shown to support high richness, including of forest specialists and understory insectivores.

Line 261: I would lead with the significant result (FF with lower growth bar width) and then follow with null findings.

Line 270-271: “most” – need to add percentage for this claim

Lines 278-279: I would rephrase this sentence away from bold claims

Lines 283 – 284: again, I would back away from this overly-bold claim. I suggest rephrasing to something like “our study identified no difference in FA between...”

Lines 293 – 294: again, I think an overly bold statement. Your study may add some support to this idea, but I wouldn't state that conversion does not have a dramatic impact.

Figure 2 and 3: check lower left heading "insectivore dependy on diet" for misspelling
Evan R. Buechley

Review form: Reviewer 2

Is the manuscript scientifically sound in its present form?

No

Are the interpretations and conclusions justified by the results?

No

Is the language acceptable?

Yes

Is it clear how to access all supporting data?

Yes

Do you have any ethical concerns with this paper?

No

Have you any concerns about statistical analyses in this paper?

No

Recommendation?

Reject

Comments to the Author(s)

This study aims to assess avian habitat quality between natural forests and shade coffee plantations in Ethiopia via individual measures of functional asymmetry and feather growth. The overall aim of the study is excellent and attempts to address an area where much work is needed. However, I am unable to evaluate the study and thus did not proceed to review the results or discussion for one major reason.

The paper does not clarify the phenological life history of the 10 study species used. This is crucial because the measures of local site habitat quality are entirely dependent on the location of 1) where the measured tarsi were developed as nestlings, and 2) where the measured feathers were grown. For example, if all 10 species were known to breed on site, and it was assumed that the individuals did not disperse to or from elsewhere, then the tarsus growth could be assumed to be influenced by the conditions of the study site. However, if the bird was captured after it had migrated from its breeding location elsewhere onto the study site, then the growth differences of the tarsus would reflect the habitat quality or conditions of the site where the development took place. It is worrisome, for example, that juveniles were not included in the study even though these would be an example of individuals for which tarsi development location would have been known. Similarly, it is not clarified where the rectrices feather growth took place and this is typically different depending on species and age. In first year birds, the rectrices would have grown in on the breeding grounds. If the captured birds were after-first-year adults, the rectrices might have molted in during different time periods and at different locations depending on the

species. If this information is unknown then the only way to know for certain that a feather was grown at the study site would be to pluck the feather of unknown origin and then pluck the regrown feather from the recaptured bird. Thus the feather growth rate would reflect conditions at that specific site and not the location where the original plucked feather was grown. It is my assumption that this information was taken into account and it is known but simply not reported. For example, it could be that the ten species are known to breed (and thus develop their tarsi) on site. It could be that the species are migrants from another breeding site but they moult rectrices during the time period in which they are captured on site, or they are year round residents and thus perform their moulting cycles on site (and do not move between the forest and shade coffee). I am happy to review the paper again if this is the case. However, if this type of information is unavailable then I am afraid it is impossible to know whether the tarsi and feather measures reflect what the authors assume they reflect.

I have made other less crucial comments in-text of the pdf that should also be addressed if revised (Appendix A).

Sincerely,
Sacha Heath, PhD
Ecology Graduate Group
University of California, Davis

Decision letter (RSOS-180982.R0)

05-Oct-2018

Dear Ms Gebremichael:

Manuscript ID RSOS-180982 entitled "Fluctuating asymmetry and feather growth rate as biomarkers to assess habitat quality of shade coffee farming for avian diversity conservation" which you submitted to Royal Society Open Science, has been reviewed. The comments from reviewers are included at the bottom of this letter.

In view of the criticisms of the reviewers, the manuscript has been rejected in its current form. However, a new manuscript may be submitted which takes into consideration these comments.

Please note that resubmitting your manuscript does not guarantee eventual acceptance, and that your resubmission will be subject to peer review before a decision is made.

Your resubmitted manuscript should be submitted by 04-Apr-2019. If you are unable to submit by this date please contact the Editorial Office.

Please note that Royal Society Open Science will introduce article processing charges for all new

submissions received from 1 January 2018. Charges will also apply to papers transferred to Royal Society Open Science from other Royal Society Publishing journals, as well as papers submitted as part of our collaboration with the Royal Society of Chemistry (<http://rsos.royalsocietypublishing.org/chemistry>). If your manuscript is submitted and accepted for publication after 1 Jan 2018, you will be asked to pay the article processing charge, unless you request a waiver and this is approved by Royal Society Publishing. You can find out more about the charges at <http://rsos.royalsocietypublishing.org/page/charges>. Should you have any queries, please contact openscience@royalsociety.org.

on behalf of Prof. Kevin Padian (Subject Editor)
openscience@royalsociety.org

Reviewers' Comments to Author:

Reviewer: 1

Comments to the Author(s)

Overview:

This is a very interesting concept, applying morphometrics and fluctuating asymmetry in a fairly novel fashion to study potential impacts of environmental quality of shade coffee and natural forest on bird communities in SW Ethiopia. The study is novel and has the potential to make a significant and important contribution to biodiversity conservation in Ethiopia, as well as to encourage similar studies worldwide to evaluate early indicators of environmental stresses on populations. On this level, I think it is suitable for publication in Royal Society Open Science.

However, I also have some significant methodological concerns with the manuscript that I hope can be addressed prior to publication. My biggest concern is regarding sample size and guild classifications, as best outlined in Table 1. Is it really suitable to compare fluctuating asymmetry and feather growth bar width of different species, as lumped into very general guilds? The difference in samples sizes of species between shade coffee and natural forest for growth bar width measurements seems particularly great. For example the top 4 species were almost exclusively sampled in shade coffee. Thus how can you compare differences between habitats??? I'm not sure that lumping them into guilds addresses this underlying issue in the dataset. If you have a strong argument for why this is valid to compare, I think it should be very prominently laid out in the methods and discussion. As is, there is really almost no discussion of such lack of uneven sampling and how that might impact results. Otherwise, perhaps the analysis should be limited to species that you have a reasonable sample size in both habitat types...

I also have a bit of a concern with the guild classification for obligate insectivore vs opportunistic insectivore, which is based on a seemingly arbitrary 80% insect-diet cutoff. Is there no way to make a more meaningful guild distinction, perhaps based on the classifications in Buechley et al 2015 (i.e. insectivore / granivore / frugivore).

A couple of other fairly major comments are as follows:

- English was good throughout, up until the discussion, which could use some detailed work simplifying and correcting grammatical errors.
- Discussion has several bold claims about coffee forests in Ethiopia that I strongly suggest toning back. This is an interesting study, but I don't think it is definitive enough to say things like

“the negative effects of converting natural forest into shade coffee forests on bird communities are not dramatic [line 294]”

More detailed points here:

Line 71: “random” – if they are random, then how can they be an index of anything?

Line 78: What about using other bio-markers like mass?

Lines 81-82: “the origin and reservoir” – i take issue with the wording here, as arabica also originated and is still found in SW Arabia

Line 94: tarsis length can be a hard measurement to standardize – perhaps address how this was done in methods

Line 95: Mass of the retrace? This seems an odd measurement!

Line 97: replace assumed with hypothesized

Line 118: comma spacing

Line 123: has this directly impacted the study sites? If so how?

Line 134: this 80% cutoff seems arbitrary. Could you not find a more well-defined guild separation, perhaps by following the definitions used in Buechley et al 2015?

Line 143: clarify where the rings are from (i.e. museum of nat hist)

Line 144: is this the only reason?

Table 1: I’m concerned about unbalanced sampling of the study species between habitat types. Is it valid to compare fluctuating asymmetry / feather growth across species?

Lines 154 – 158: I’m a bit skeptical about this method. Retrix feathers are often soiled by feces. Were the feathers cleaned prior to weighing? If so, how? Retrix feathers also undergo major wear. It is not clear to me that tropical birds would necessarily have the same moult pattern, thus different individuals may be under different levels of feather wear at any given time, which could impact feather mass. I strongly suggest adding more detail of methods and citations validating these indices here.

Line 162: “unworned” – what does this mean??

Line 194: I like that repeatability of the measurements was included in the model

Line 231: spell out acronyms (e.g. FF, DA, etc) at start of new section

Lines 241 – 245: was this caused by scale accuracy? If the scale is accurate, it seems this would be the easier measurement to replicate. Can you explain this result more, perhaps in Discussion?

Lines 252-254: show p value for this comment

Lines 259 – 260: I would cut this sentence off at “natural forest”. It was shown to support high richness, including of forest specialists and understory insectivores.

Line 261: I would lead with the significant result (FF with lower growth bar width) and then follow with null findings.

Line 270-271: “most” – need to add percentage for this claim

Lines 278-279: I would rephrase this sentence away from bold claims

Lines 283 – 284: again, I would back away from this overly-bold claim. I suggest rephrasing to something like “our study identified no difference in FA between...”

Lines 293 – 294: again, I think an overly bold statement. Your study may add some support to this idea, but I wouldn’t state that conversion does not have a dramatic impact.

Figure 2 and 3: check lower left heading “insectivore dependy on diet” for misspelling

Evan R. Buechley

Reviewer: 2

Comments to the Author(s)

This study aims to assess avian habitat quality between natural forests and shade coffee plantations in Ethiopia via individual measures of functional asymmetry and feather growth. The overall aim of the study is excellent and attempts to address an area where much work is needed. However, I am unable to evaluate the study and thus did not proceed to review the results or discussion for one major reason.

The paper does not clarify the phenological life history of the 10 study species used. This is crucial because the measures of local site habitat quality are entirely dependent on the location of 1) where the measured tarsi were developed as nestlings, and 2) where the measured feathers were grown. For example, if all 10 species were known to breed on site, and it was assumed that the individuals did not disperse to or from elsewhere, then the tarsus growth could be assumed to be influenced by the conditions of the study site. However, if the bird was captured after it had migrated from its breeding location elsewhere onto the study site, then the growth differences of the tarsus would reflect the habitat quality or conditions of the site where the development took place. It is worrisome, for example, that juveniles were not included in the study even though these would be an example of individuals for which tarsi development location would have been known. Similarly, it is not clarified where the rectrices feather growth took place and this is typically different depending on species and age. In first year birds, the rectrices would have grown in on the breeding grounds. If the captured birds were after-first-year adults, the rectrices might have molted in during different time periods and at different locations depending on the species. If this information is unknown then the only way to know for certain that a feather was grown at the study site would be to pluck the feather of unknown origin and then pluck the regrown feather from the recaptured bird. Thus the feather growth rate would reflect conditions at that specific site and not the location where the original plucked feather was grown.

It is my assumption that this information was taken into account and it is known but simply not reported. For example, it could be that the ten species are known to breed (and thus develop their tarsi) on site. It could be that the species are migrants from another breeding site but they moult rectrices during the time period in which they are captured on site, or they are year round residents and thus perform their moulting cycles on site (and do not move between the forest and shade coffee). I am happy to review the paper again if this is the case. However, if this type of information is unavailable then I am afraid it is impossible to know whether the tarsi and feather measures reflect what the authors assume they reflect.

I have made other less crucial comments in-text of the pdf that should also be addressed if revised.

Sincerely,
Sacha Heath, PhD
Ecology Graduate Group
University of California, Davis

Author's Response to Decision Letter for (RSOS-180982.R0)

See Appendix B.

RSOS-190013.R0

Review form: Reviewer 2

Is the manuscript scientifically sound in its present form?

Yes

Are the interpretations and conclusions justified by the results?

Yes

Is the language acceptable?

No

Is it clear how to access all supporting data?

No

Do you have any ethical concerns with this paper?

No

Have you any concerns about statistical analyses in this paper?

No

Recommendation?

Major revision is needed (please make suggestions in comments)

Comments to the Author(s)

I am happy to once again review this paper which aims to compare avian habitat quality between shade coffee and nearby natural forests in Ethiopia. The paper is an important contribution in that unlike most studies of this sort, it aims to compare biomarkers rather than community measures. Both approaches are important, but as the authors note, occupancy alone is not always the best measure of habitat quality.

This version and the author responses addressed my major concern from my previous review. I

also agreed with the primary comment of the 2nd reviewer that it is more appropriate to compare these biomarkers between habitat types within the same species, rather than between guilds. The reworked paper is much more sound, though I recommend that a major revision is still needed. I think that all of the pieces are here (in terms of the data, the design, and the statistical analysis) to present a simple but important result, but I am afraid that the grammar and writing make the story more complicated than it needs to be.

I think it would be very helpful for the authors to request assistance from a copy editor. I am very familiar with the general topic of the paper, thus with some effort I was able to decipher what the paper was explicitly attempting to convey. I fear that readers less familiar with the topic will not be able to do so. I provided a few comments and suggestions for sentences where I found this to be especially true, but I would recommend a thorough check on both grammar and sentence structure so that the primary message is conveyed more clearly.

I am not familiar with the details of analysis of FA, but I think that too much detail of this analysis is provided in the results section. The main aim of the paper is to compare habitat quality (via biomarkers) between the two habitat types; I suggest leading the results section with this result. As it currently stands, this main result is not addressed until the end of the second paragraph.

In reference to a comment from my previous review, the authors stated that they addressed this in the discussion, however this version does not contain any mention of this in the discussion. I actually suggest that this instead be briefly addressed in the methods instead. The original comment was: "No need to go into an exhaustive review of these two measures, but since the entire study is based on FA and feather growth, it would help the reader to have a bit more justification for the validity of the methods. Please more explicitly explain the links between FA and Feather Growth to fundamental life history components. Also, what do you think about the criticisms of ptilochronology? (for example, see citations below). Again, no need for too much detail but please add in a few more clarifying sentences to make the life history links and very briefly address the criticisms of Murphy.

Murphy and King. 1991. Ptilochronology: a critical evaluation of assumptions and utility. *The Auk* 108:695-704

Murphy. 1992. Ptilochronology: accuracy and reliability of the technique. *The Auk* 109:676-680."

Finally, I see that the paper does link to the Dryad Digital Repository. However, this is done in the literature cited section and I just happened to run across it. I don't believe the citation (#77) is referred to anywhere in the text. This is the reason that I checked "no" for the two questions pertaining to the accessibility of the supporting data. I recommend making this link more clear by adding it as a supplement or mentioning it in the acknowledgements.

I am happy to review this paper again (see Appendix C).

Decision letter (RSOS-190013.R0)

15-Mar-2019

Dear Ms Gebremichael,

The Subject Editor assigned to your paper ("Fluctuating asymmetry and feather growth bars as biomarkers to assess habitat quality of shade coffee farming for avian diversity conservation") has now received comments from reviewers. We would like you to revise your paper in accordance with the referee and Associate Editor suggestions which can be found below (not including confidential reports to the Editor). Please note this decision does not guarantee eventual acceptance.

Please submit a copy of your revised paper before 07-Apr-2019. Please note that the revision deadline will expire at 00.00am on this date. If we do not hear from you within this time then it will be assumed that the paper has been withdrawn. In exceptional circumstances, extensions may be possible if agreed with the Editorial Office in advance. We do not allow multiple rounds of revision so we urge you to make every effort to fully address all of the comments at this stage. If deemed necessary by the Editors, your manuscript will be sent back to one or more of the original reviewers for assessment. If the original reviewers are not available we may invite new reviewers.

When submitting your revised manuscript, you must respond to the comments made by the referees and upload a file "Response to Referees" in "Section 6 - File Upload". Please use this to document how you have responded to each of the comments, and the adjustments you have made. In order to expedite the processing of the revised manuscript, please be as specific as possible in your response.

- Ethics statement

- Data accessibility

<http://datadryad.org/submit?journalID=RSOS&manu=RSOS-190013>

- Competing interests

- Authors' contributions

All submissions, other than those with a single author, must include an Authors' Contributions section which individually lists the specific contribution of each author. The list of Authors

should meet all of the following criteria; 1) substantial contributions to conception and design, or acquisition of data, or analysis and interpretation of data; 2) drafting the article or revising it critically for important intellectual content; and 3) final approval of the version to be published.

- Acknowledgements

- Funding statement

on behalf of Professor Kevin Padian (Subject Editor)
openscience@royalsociety.org

Associate Editor Comments to Author:

One of the reviewers of your original submission has provided commentary on the resubmission, and though they see much improvement, there are still a number of matters that need addressing. You are invited to submit a revision to tackle these. Please note that you will not be entitled to submit a further revision, so please ensure you fully resolve these issues in the revision. Good luck!

Reviewer comments to Author:

Reviewer: 2

Comments to the Author(s)

I am happy to once again review this paper which aims to compare avian habitat quality between shade coffee and nearby natural forests in Ethiopia. The paper is an important contribution in that unlike most studies of this sort, it aims to compare biomarkers rather than community measures. Both approaches are important, but as the authors note, occupancy alone is not always the best measure of habitat quality.

This version and the author responses addressed my major concern from my previous review. I also agreed with the primary comment of the 2nd reviewer that it is more appropriate to compare these biomarkers between habitat types within the same species, rather than between guilds. The reworked paper is much more sound, though I recommend that a major revision is still needed. I

think that all of the pieces are here (in terms of the data, the design, and the statistical analysis) to present a simple but important result, but I am afraid that the grammar and writing make the story more complicated than it needs to be.

I think it would be very helpful for the authors to request assistance from a copy editor. I am very familiar with the general topic of the paper, thus with some effort I was able to decipher what the paper was explicitly attempting to convey. I fear that readers less familiar with the topic will not be able to do so. I provided a few comments and suggestions for sentences where I found this to be especially true, but I would recommend a thorough check on both grammar and sentence structure so that the primary message is conveyed more clearly.

I am not familiar with the details of analysis of FA, but I think that too much detail of this analysis is provided in the results section. The main aim of the paper is to compare habitat quality (via biomarkers) between the two habitat types; I suggest leading the results section with this result. As it currently stands, this main result is not addressed until the end of the second paragraph.

In reference to a comment from my previous review, the authors stated that they addressed this in the discussion, however this version does not contain any mention of this in the discussion. I actually suggest that this instead be briefly addressed in the methods instead. The original comment was: "No need to go into an exhaustive review of these two measures, but since the entire study is based on FA and feather growth, it would help the reader to have a bit more justification for the validity of the methods. Please more explicitly explain the links between FA and Feather Growth to fundamental life history components. Also, what do you think about the criticisms of ptilochronology? (for example, see citations below). Again, no need for too much detail but please add in a few more clarifying sentences to make the life history links and very briefly address the criticisms of Murphy.

Murphy and King. 1991. Ptilochronology: a critical evaluation of assumptions and utility. *The Auk* 108:695-704

Murphy. 1992. Ptilochronology: accuracy and reliability of the technique. *The Auk* 109:676-680."

Finally, I see that the paper does link to the Dryad Digital Repository. However, this is done in the literature cited section and I just happened to run across it. I don't believe the citation (#77) is referred to anywhere in the text. This is the reason that I checked "no" for the two questions pertaining to the accessibility of the supporting data. I recommend making this link more clear by adding it as a supplement or mentioning it in the acknowledgements.

I am happy to review this paper again.

Author's Response to Decision Letter for (RSOS-190013.R0)

See Appendix D.

RSOS-190013.R1 (Revision)

Review form: Reviewer 2

Is the manuscript scientifically sound in its present form?

Yes

Are the interpretations and conclusions justified by the results?

Yes

Is the language acceptable?

Yes

Is it clear how to access all supporting data?

Yes

Do you have any ethical concerns with this paper?

No

Have you any concerns about statistical analyses in this paper?

No

Recommendation?

Accept with minor revision (please list in comments)

Comments to the Author(s)

The authors did a very thorough job of addressing my previous concerns and have tightened up the language substantially. This is a well written, succinct, and informative paper that will make a good contribution to the agroecology literature and to avian conservation practice in agroecosystems.

I have provided a few minor edits and suggestions (one sentence in the abstract, some minor presentation suggestions in the results, and some changes to sentence structure and rewording for clarity in the conclusion are the most substantial of these). These should take very little time to address, after which I recommend accept as is without the need to review the MS again. The authors have demonstrated great care in responding to and addressing previous suggestions and it will be great to see this published (see Appendix E).

Review form: Reviewer 3

Is the manuscript scientifically sound in its present form?

No

Are the interpretations and conclusions justified by the results?

No

Is the language acceptable?

Yes

Is it clear how to access all supporting data?

Yes

Do you have any ethical concerns with this paper?

No

Have you any concerns about statistical analyses in this paper?

No

Recommendation?

Major revision is needed (please make suggestions in comments)

Comments to the Author(s)

This is an interesting study that uses fluctuating asymmetry (tarsus and tail feathers) and feather growth rates (ptilochronology) as biomarkers to compare birds breeding in shade coffee plantations vs native wet forests in the mountains of Ethiopia.

The topic is of general interest in that it may provide insight into whether habitat quality is equivalent between agroforestry versus native forests; not just that the species are retained in shade coffee plantations but that they are still able to sufficiently breed and not be sink populations.

While I like the idea of this study, I have a couple of comments that I think the authors would have to address in order to convince readers of the effectiveness of their technique.

1. Validation of the Biomarkers

The first of these comments is regarding the “ground truthing” of these two biomarkers – while there are studies to suggest that both Fluctuating Asymmetry and Ptilochronology can be reliably associated with measures of individual condition, there are also studies that have failed to make these connections. In the current study, there is no independent measure of condition to which the measures of FA or Tail Growth are compared so as to show they can accurately be used as biomarkers. This assumption, though, is the basis for the entire study (L104-107) and recommendation for further use in other studies (L319-321). I suspect that the authors do not have data on reproductive success or other measures of condition that they can correlate asymmetry or tail-growth measures to, but at the very least they could compare these measure to one another.

The use of the two biomarkers in tandem, as suggested (L84-86, L303-305), would indicate they show the same pattern. As a result, there should be a strong negative correlation between level of asymmetry and tail feather growth bar width (little asymmetry should correspond to large growth bars, and high asymmetry with small growth bars). If the authors could show this association, it would help validate these as potentially sensitive biomarkers of condition.

2. Ptilochronology Measure

The authors have measured growth bands on tail feathers plucked from birds at initial capture. While they have definitely refined on Grubb’s technique (e.g. digital vs manual measurement of individual vs groups of growth bars – L181-182 – and repeatability studies on measures that show remarkably high confidence in measurement –L191-194), there are still problems with measuring growth rates in feathers in this manner. First, as the authors indicate, feathers are moulted throughout adult life, so represent a snapshot of condition/food availability in the habitat during the time period of regrowth (L110-112). However, there is no indication in this study if all the birds from which feathers were drawn are of the same relative age (all adults, no juveniles) to suggest they all moulted their feathers during the same time period. Although the authors indicate all species are resident breeders in the region (L144), they don’t indication whether they are year-round residents on the exact territories in which they were captured during these studies, or whether they may have been occupying different areas during the period of feather regrowth. Without this information, you don’t have a firm basis to conclude that feather growth bars reflect growth in the habitats in which the birds are currently located. If shade coffee sites are sink populations, you could be measuring birds that grew their feathers in forested areas, then dispersed to shade coffee sites.

The advocated technique in ptilochronology studies by Grubb is to pluck feathers from the birds to induce regrowth, then recapture the birds and pluck the regrown feathers for measurement. This standardizes both the temporal period in which the feathers are regrown, as well as confirms they are regrown in the habitat the bird is currently occupying. I suspect this is not possible in the current study, but the limitation of the technique you have used needs to be addressed and also highlighted as a possible source of not detecting differences (e.g. L 267-268).

3. Native Forest and Shade Coffee sites being of “similar quality”

This is the conclusion that is drawn by the authors not finding differences in FA or Tail Growth bars (L289). First, I think you need to account for the validity of your biomarkers and limitations of your ptilochronology technique (above two points), but even then I would argue that your results hint that this is a false conclusion to draw. While not statistically significant, the FA data does show a pretty consistent pattern across 4 of the 5 species that there was slightly higher FA in the shade coffee sites than natural forest sites (Figure 2a). Only one species with extremely variable tarsal FA (Grey-backed Camaroptera) doesn't show this pattern, and two of the other species have suggestive P values in the 0.1 range. As you are using a proxy of condition, rather than a direct measure of condition, I would suggest you are a little more cautious. While you can't statistically conclude there is a difference, it doesn't mean the two habitats are necessarily equivalent.

Decision letter (RSOS-190013.R1)

21-May-2019

Dear Ms Gebremichael:

Manuscript ID RSOS-190013.R1 entitled "Fluctuating asymmetry and feather growth bars as biomarkers to assess habitat quality of shade coffee farming for avian diversity conservation" which you submitted to Royal Society Open Science, has been reviewed. The comments of the reviewer(s) are included at the bottom of this letter.

Please submit a copy of your revised paper before 13-Jun-2019. Please note that the revision deadline will expire at 00.00am on this date. If we do not hear from you within this time then it will be assumed that the paper has been withdrawn. In exceptional circumstances, extensions may be possible if agreed with the Editorial Office in advance. We do not allow multiple rounds of revision so we urge you to make every effort to fully address all of the comments at this stage. If deemed necessary by the Editors, your manuscript will be sent back to one or more of the original reviewers for assessment. If the original reviewers are not available we may invite new reviewers.

- Ethics statement

- Data accessibility

- Competing interests

- Authors' contributions

- Acknowledgements

- Funding statement

Kind regards,
Andrew Dunn
Royal Society Open Science Editorial Office

on behalf of Prof Kevin Padian (Subject Editor)
openscience@royalsociety.org

Editor comments:

Unusually, we are recommending a major revision of your paper - only in rare instances are major revisions granted more than once per paper, so we strongly urge you to take this opportunity to get your paper over the line. The commentary from the reviewers suggests that your paper has merit and that you are making good faith efforts to improve the paper, but a number of (fixable) methodological concerns remain. Please ensure that you incorporate the changes required by the new reviewer, and provide a full point-by-point response in your revision. Good luck, and we look forward to receiving the final version of your paper.

Reviewer comments to Author:
Reviewer: 3

Comments to the Author(s)

This is an interesting study that uses fluctuating asymmetry (tarsus and tail feathers) and feather growth rates (ptilochronology) as biomarkers to compare birds breeding in shade coffee plantations vs native wet forests in the mountains of Ethiopia.

The topic is of general interest in that it may provide insight into whether habitat quality is equivalent between agroforestry versus native forests; not just that the species are retained in shade coffee plantations but that they are still able to sufficiently breed and not be sink populations.

While I like the idea of this study, I have a couple of comments that I think the authors would have to address in order to convince readers of the effectiveness of their technique.

1. Validation of the Biomarkers

The first of these comments is regarding the “ground truthing” of these two biomarkers – while there are studies to suggest that both Fluctuating Asymmetry and Ptilochronology can be reliably associated with measures of individual condition, there are also studies that have failed to make these connections. In the current study, there is no independent measure of condition to which the measures of FA or Tail Growth are compared so as to show they can accurately be used as biomarkers. This assumption, though, is the basis for the entire study (L104-107) and recommendation for further use in other studies (L319-321). I suspect that the authors do not have data on reproductive success or other measures of condition that they can correlate asymmetry or tail-growth measures to, but at the very least they could compare these measure to one another.

The use of the two biomarkers in tandem, as suggested (L84-86, L303-305), would indicate they show the same pattern. As a result, there should be a strong negative correlation between level of asymmetry and tail feather growth bar width (little asymmetry should correspond to large growth bars, and high asymmetry with small growth bars). If the authors could show this association, it would help validate these as potentially sensitive biomarkers of condition.

2. Ptilochronology Measure

The authors have measured growth bands on tail feathers plucked from birds at initial capture. While they have definitely refined on Grubb’s technique (e.g. digital vs manual measurement of

individual vs groups of growth bars – L181-182 – and repeatability studies on measures that show remarkably high confidence in measurement –L191-194), there are still problems with measuring growth rates in feathers in this manner. First, as the authors indicate, feathers are moulted throughout adult life, so represent a snapshot of condition/food availability in the habitat during the time period of regrowth (L110-112). However, there is no indication in this study if all the birds from which feathers were drawn are of the same relative age (all adults, no juveniles) to suggest they all moulted their feathers during the same time period. Although the authors indicate all species are resident breeders in the region (L144), they don't indicate whether they are year-round residents on the exact territories in which they were captured during these studies, or whether they may have been occupying different areas during the period of feather regrowth. Without this information, you don't have a firm basis to conclude that feather growth bars reflect growth in the habitats in which the birds are currently located. If shade coffee sites are sink populations, you could be measuring birds that grew their feathers in forested areas, then dispersed to shade coffee sites.

The advocated technique in ptilochronology studies by Grubb is to pluck feathers from the birds to induce regrowth, then recapture the birds and pluck the regrown feathers for measurement. This standardizes both the temporal period in which the feathers are regrown, as well as confirms they are regrown in the habitat the bird is currently occupying. I suspect this is not possible in the current study, but the limitation of the technique you have used needs to be addressed and also highlighted as a possible source of not detecting differences (e.g. L 267-268).

3. Native Forest and Shade Coffee sites being of “similar quality”

This is the conclusion that is drawn by the authors not finding differences in FA or Tail Growth bars (L289). First, I think you need to account for the validity of your biomarkers and limitations of your ptilochronology technique (above two points), but even then I would argue that your results hint that this is a false conclusion to draw. While not statistically significant, the FA data does show a pretty consistent pattern across 4 of the 5 species that there was slightly higher FA in the shade coffee sites than natural forest sites (Figure 2a). Only one species with extremely variable tarsal FA (Grey-backed Camaroptera) doesn't show this pattern, and two of the other species have suggestive P values in the 0.1 range. As you are using a proxy of condition, rather than a direct measure of condition, I would suggest you are a little more cautious. While you can't statistically conclude there is a difference, it doesn't mean the two habitats are necessarily equivalent.

Reviewer: 2

Comments to the Author(s)

The authors did a very thorough job of addressing my previous concerns and have tightened up the language substantially. This is a well written, succinct, and informative paper that will make a good contribution to the agroecology literature and to avian conservation practice in agroecosystems.

I have provided a few minor edits and suggestions (one sentence in the abstract, some minor presentation suggestions in the results, and some changes to sentence structure and rewording for clarity in the conclusion are the most substantial of these). These should take very little time to address, after which I recommend accept as is without the need to review the MS again. The authors have demonstrated great care in responding to and addressing previous suggestions and it will be great to see this published.

Author's Response to Decision Letter for (RSOS-190013.R1)

See Appendix F.

RSOS-190013.R2 (Revision)

Review form: Reviewer 3

Is the manuscript scientifically sound in its present form?

Yes

Are the interpretations and conclusions justified by the results?

Yes

Is the language acceptable?

Yes

Do you have any ethical concerns with this paper?

No

Recommendation?

Accept with minor revision (please list in comments)

Comments to the Author(s)

The authors have addressed some of my initial concerns - for example they have confirmed that individual birds captured as juveniles were recaptured in subsequent years in the same locations, indicating that the biomarkers they measure (tarsal asymmetry reflecting nestling growth conditions, tail asymmetry reflecting stress on adults, and feather growth rates indicative of food availability) are accurately reflecting the conditions under the two habitats they are testing.

They have not addressed my other concern - the validation of the biomarkers themselves. In their reply, they indicate they simply assume these markers are sensitive based on other studies. This is sufficient so long as they explicitly acknowledge this in the manuscript, but I also suggested a simple means of verifying this through a quick correlation between the markers to look for a negative association. The authors appear to have attempted this (Reply letter) with a bivariate pearson correlation, with the results being in the correct direction but not significant. While they indicate this appears in the paper, the line numbers they refer to don't include this analysis, nor did I find it mentioned in the entire paper.

Despite this, the authors have qualified a bit of their language in the revised manuscript to acknowledge a bit more uncertainty of their results, particularly as their results failed to show an difference in bioindicators between samples from either habitat where others have found such differences (Buechley et al. 2015, L298).

However, without this validation, I feel that the paragraph (L315-323) recommending "future studies combine the use of both biomarkers to robustly asses the habitat quality" is overstated. This advocacy is problematic when you yourselves have assumed these are robust measures without actually testing or demonstrating this. I suggest simply omitting this paragraph.

L290-294. Sentences are poorly constructed. I would suggest something like:

"As individual birds were initially banded as juveniles, and subsequently recaptured in later years as adults in the same sites, this suggests our biomarker measures (tarsal and retrice asymmetry and feather growth rates) should reflect conditions faced within our two specific regions - shade coffee vs natural forests".

I would also suggest this would probably be better to put into the Methods than leave to the Discussion.

Decision letter (RSOS-190013.R2)

19-Jun-2019

Dear Ms Gebremichael:

On behalf of the Editors, I am pleased to inform you that your Manuscript RSOS-190013.R2 entitled "Fluctuating asymmetry and feather growth bars as biomarkers to assess habitat quality of shade coffee farming for avian diversity conservation" has been accepted for publication in Royal Society Open Science subject to minor revision in accordance with the referee suggestions. Please find the referees' comments at the end of this email.

The reviewers and Subject Editor have recommended publication, but also suggest some minor revisions to your manuscript. Therefore, I invite you to respond to the comments and revise your manuscript.

- Ethics statement

- Data accessibility

If you wish to submit your supporting data or code to Dryad (<http://datadryad.org/>), or modify your current submission to dryad, please use the following link:
<http://datadryad.org/submit?journalID=RSOS&manu=RSOS-190013.R2>

- Competing interests

- Authors' contributions

- Acknowledgements

- Funding statement

Because the schedule for publication is very tight, it is a condition of publication that you submit the revised version of your manuscript before 28-Jun-2019. Please note that the revision deadline will expire at 00.00am on this date. If you do not think you will be able to meet this date please let me know immediately.

on behalf of Kevin Padian (Subject Editor)
openscience@royalsociety.org

Reviewer comments to Author:
Reviewer: 3

Comments to the Author(s)

The authors have addressed some of my initial concerns - for example they have confirmed that individual birds captured as juveniles were recaptured in subsequent years in the same locations, indicating that the biomarkers they measure (tarsal asymmetry reflecting nestling growth conditions, tail asymmetry reflecting stress on adults, and feather growth rates indicative of food availability) are accurately reflecting the conditions under the two habitats they are testing.

They have not addressed my other concern - the validation of the biomarkers themselves. In their reply, they indicate they simply assume these markers are sensitive based on other studies. This is sufficient so long as they explicitly acknowledge this in the manuscript, but I also suggested a simple means of verifying this through a quick correlation between the markers to look for a negative association. The authors appear to have attempted this (Reply letter) with a bivariate pearson correlation, with the results being in the correct direction but not significant. While they indicate this appears in the paper, the line numbers they refer to don't include this analysis, nor did I find it mentioned in the entire paper.

Despite this, the authors have qualified a bit of their language in the revised manuscript to acknowledge a bit more uncertainty of their results, particularly as their results failed to show a difference in bioindicators between samples from either habitat where others have found such differences (Buechley et al. 2015, L298).

However, without this validation, I feel that the paragraph (L315-323) recommending "future studies combine the use of both biomarkers to robustly assess the habitat quality" is overstated. This advocacy is problematic when you yourselves have assumed these are robust measures without actually testing or demonstrating this. I suggest simply omitting this paragraph.

L290-294. Sentences are poorly constructed. I would suggest something like:

"As individual birds were initially banded as juveniles, and subsequently recaptured in later years as adults in the same sites, this suggests our biomarker measures (tarsal and retrice asymmetry and feather growth rates) should reflect conditions faced within our two specific regions - shade coffee vs natural forests".

I would also suggest this would probably be better to put into the Methods than leave to the Discussion.

Author's Response to Decision Letter for (RSOS-190013.R2)

See Appendix G.

RSOS-190013.R3 (Revision)

Review form: Reviewer 3

Is the manuscript scientifically sound in its present form?

Yes

Are the interpretations and conclusions justified by the results?

Yes

Is the language acceptable?

Yes

Do you have any ethical concerns with this paper?

No

Have you any concerns about statistical analyses in this paper?

Yes

Recommendation?

Accept as is

Comments to the Author(s)

I am satisfied that with the author's response to my final comments.

The only outstanding suggestion would be wording of their concluding sentence:

L325-330 - change to:

The lack of strong correlation between the growth bar width and FA tarsus length mean the biomarkers used in this study are not equally sensitive for assessing the effects of habitat change. Only a few studies have combined FA and feather growth bar width analysis as biomarkers of habitat fragmentation; in some of these, the two markers both both varied with habitat quality, but in others only one of the two markers showed an effect of fragmentation. We recommend future studies combine the use of multiple biomarkers when assessing changes to the habitat quality following habitat conversion."

Decision letter (RSOS-190013.R3)

23-Jul-2019

Dear Ms Gebremichael,

I am pleased to inform you that your manuscript entitled "Fluctuating asymmetry and feather growth bars as biomarkers to assess habitat quality of shade coffee farming for avian diversity conservation" is now accepted for publication in Royal Society Open Science.

on behalf of Prof Kevin Padian (Subject Editor)
openscience@royalsociety.org

Associate Editor Comments to Author:

The reviewers and editors are broadly happy to accept your manuscript for publication. The reviewer has a final recommendation of a tweak to the text of your conclusion - this is recommended for inclusion by the editors, too, though you can make this minor modification during proofing. Please ensure that, when you receive the proofs, you include the changes.

Reviewer comments to Author:
Reviewer: 3

Comments to the Author(s)
I am satisfied that with the author's response to my final comments.

The only outstanding suggestion would be wording of their concluding sentence:
L325-330 - change to:

The lack of strong correlation between the growth bar width and FA tarsus length mean the biomarkers used in this study are not equally sensitive for assessing the effects of habitat change. Only a few studies have combined FA and feather growth bar width

analysis as biomarkers of habitat fragmentation; in some of these, the two markers both both varied with habitat quality, but in others only one of the two markers showed an effect of fragmentation. We recommend future studies combine the use of multiple biomarkers when assessing changes to the habitat quality following habitat conversion."

Appendix A**ROYAL SOCIETY
OPEN SCIENCE****Fluctuating asymmetry and feather growth rate as
biomarkers to assess habitat quality of shade coffee
farming for avian diversity conservation**

Journal:	Royal Society Open Science
Manuscript ID	RSOS-180982
Article Type:	Research
Date Submitted by the Author:	18-Jun-2018
Complete List of Authors:	Gebremichael, Gelaye; Jimma University, Biology Department; Ghent University, Department of Biology Alemu, Diress ; Universitetet i Oslo Det Matematisk-naturvitenskapelige Fakultet, Department of Biosciences Bunnefeld, Nils; University of Stirling Zinner, Dietmar ; Germany primate center Atickem, Anagaw; Germany primate center , Cognitive Ethology Laboratory
Subject:	ecology < BIOLOGY
Keywords:	bird community, functional guilds, ptilochronology
Subject Category:	Biology (whole organism)

**1 Fluctuating asymmetry and feather growth rate as biomarkers to assess habitat quality**
**2 of shade coffee farming for avian diversity conservation**

Gelaye Gebremichael^{1,2}, Diress Tsegaye³, Nils Bunnefeld^{4,5}, Dietmar Zinner⁶, Anagaw
Atickem⁶

¹Jimma Universities, College of Natural Sciences, P.O. Box 378, Jimma, Ethiopia

²Terrestrial Ecology Unit (TEREC), Ghent University, K.L. Ledeganckstraat 35, 9000, Ghent,
Belgium

³University of Oslo, Department of Biosciences, Postboks 1066 Blindern, 0316 Oslo, Norway

⁴Biological and Environmental Sciences, Faculty of Natural Sciences, University of Stirling,
Stirling FK9 4LA, UK;

⁵School of Geosciences, University of Edinburgh, Edinburgh, EH9

⁶Cognitive Ethology Laboratory, German Primate Center, Leibniz Institute for Primate
Research Kellnerweg 4, 37077 Göttingen, Germany

**Fluctuating asymmetry and feather growth rate as biomarkers to assess habitat quality**
**of shade coffee farming for avian diversity conservation**

**Abstract**

Shade coffee farming has been promoted as a means for combining sustainable use of forests
and biodiversity conservation, mainly because in shade coffee forests and natural forests a
similar diversity and abundance of, e.g. birds, have been found. However, since negative
effects on such indicators for habitat quality may become obvious only after some time,
because of constant influx of birds from nearby subpopulations in natural forests, other
indicators for habitat quality of shade coffee forests should also be applied. We used two
biomarkers, fluctuating asymmetry of tarsus length (FA) and feather growth rate (relative
growth bar width), **assuming higher FA and narrower feather growth bars in shade coffee**
**forest than in natural forest.** We measured and compared **more than 350 birds** of four
functional guilds (forest specialist and generalist, as well as obligate and opportunistic
insectivores) from both forest types. The extent of FA showed no difference between the two
forest types in any of the four guilds. Similarly, no difference was found in growth bar widths
in any of these guilds with the exception of forest specialist birds, which had narrower growth
bars in shade coffee forests than in natural forests. This suggests that shade coffee forests
constitute a habitat lower value for this guild. From our results we conclude that although
shade coffee farming seems to have little negative effects on most bird species (guilds), the
promotion of this agro-forestry practice should not lead to more conversion of natural forest
into shade coffee forest, **but to support conversion of degraded forests into shade coffee**
**forests.**

**Keywords:** bird community, functional guilds, Ethiopia, ptilochronology

1. Introduction

Tropical forests host at least two-thirds of the world's terrestrial biodiversity and offer a wide range of ecosystem services [1]. However, these forests are shrinking at unprecedented rates [2,3]. Agricultural practices compatible with forest conservation have been proposed as a potential strategy to slow down, stop or even reverse the current alarming rate of deforestation [4].

Shade coffee farming, i.e. growing coffee beneath the canopy of shade trees, has recently gained support as a form of agricultural practice that is, at least partly, more compatible with biodiversity conservation than other agricultural practices [5-7]. For instance, ~~shade coffee farming seems to have no negative effects on~~ species richness, abundance and community composition ~~as compared to natural forests (8-10)~~. Therefore, it ~~was~~ concluded, that shade coffee farming can play an important role when combining biodiversity conservation and agro-forestry [10-12].

Such inference implicitly assumes that populations inhabiting shade coffee forests also share equal prospects of long-term survival or/and reproduction with conspecifics residing in pristine forest [13,14]. **However, demographic changes in the distribution of sex or age classes, changes in rates of survival or reproduction, which results from biotic changes, are possibly long term effects which are unlikely to be picked up by community-level studies on demography and α -diversity alone, if surveys are done shortly after the changes happened.** Populations can be affected without any noticeable change in diversity and demography due to time lags in responses or immigration from nearby source populations [15,16]. To overcome problems of demography based studies, **a growing number of studies applied** phenotypic or physiological proxies of fitness (biomarkers), such as body size, body mass, fat-free mass, or the rate or precision of growth, which are all thought to be able to indicate stress (e.g. nutritional stress), and thus ~~lower~~ habitat quality. ~~Lower~~ habitat quality in turn,

~~will have negative~~ effects on respective populations in the long run [17,18]. Stress-mediated
changes should preferably be detectable before direct components of fitness are compromised
(see [19] for a list of criteria to evaluate biomarkers) and, within conservation contexts,
collecting respective data should not involve lethal sampling, exhaustive training or
expensive equipment.

Fluctuating asymmetry (FA), i.e. small random deviations from left-right symmetry in
bilateral traits [20-23], and feather growth rate, measured as the width of alternating dark and
light growth-bars on bird feathers [ptilochronology, 24-27], fulfil the above criteria. The
benefits of these markers are based on their ease, low cost, and non-lethality of measurement
and their link to fundamental life-history components that are otherwise more laborious to
measure [28]. When applied in tandem [29,30], they provide an overall, integrated picture of
environmental and nutritional stress effects [17], which in turn can provide information on
habitat quality.

Ethiopian coffee mainly grows in tropical rain forest remnants belonging to the Eastern
Afromontane biodiversity hotspot. This hotspot is known for its exceptionally high level of
biodiversity and endemism [31]. The respective forests are also the origin and reservoir of
wild *Coffea arabica* [32,33]. Buechley et al. [34] showed that bird diversity in shade coffee
forests in SW Ethiopia did not significantly differ from that of ecologically comparable
natural moist evergreen Afromontane forests in the same region. The forests used for coffee
production even hosted over more than twice the number of species of the latter. While all
bird species recorded in primary forest control sites were also recorded in shade coffee forests,
shade coffee plots, ~~however,~~ showed a lower abundance of forest specialists and understory
insectivores. The authors ~~therefore~~ concluded that, while Ethiopian shade coffee farming is
perhaps the most “bird-friendly” coffee in the world, little-disturbed forest remains critical
for sustaining at-risk groups of birds, such as forest specialist birds [34].

To determine whether populations of forest-dependent species that currently persist in
Ethiopian shade coffee farms show signs of increased environmental and/or nutritional stress
compared to conspecifics from natural forests, we revisited a subset of the study plots
sampled by Buechley et al. [34] and measured FA of two phenotypic traits (length of tarsi
and mass of the second outermost rectrices) and feather growth rates of the second outermost
rectrices in 380 individual birds belonging to ten species. If shade coffee farming ~~has~~
~~negative effects on bird communities~~, we ~~assumed to find~~ higher FA and lower feather
growth rates (i.e., narrower feather growth bars) in birds from shade coffee forests as
compared to birds from undisturbed forests. **Studying both traits allowed us to assess possible**
**adverse effects of shade coffee farming during subsequent life stages**, because tarsi are
100 already fully grown before fledging, **thus providing information on stress during the nestling**
**period [35,36], while rectrices are moulted throughout adult life, providing information on**
**stress in later phases of the individuals' life-history.**

2. Materials and method

2.1 Study site

The study was conducted in south-western Ethiopia at three sites in shade coffee farming area
(422 km²): Fetcbe, Garuke, and Yebu, and at four natural forest sites in the Belete-Gera
Forest (920 km²): Afalo, Abana Buna, Qacho, and Muje (Fig. 1).

In the shade coffee production scheme, the forest was modified from the original forest
to improve the productivity of coffee plants by reducing tree and shrub density. This leads to
obvious differences in the vegetation density and canopy cover compared to the natural
forest. In the shade coffee areas seedling density, stem density and crown cover of indigenous
trees were 3000 ha⁻¹, 655 ha⁻¹ and 79%, respectively, whereas in natural forests respective
numbers were 10,000 ha⁻¹, 952 ha⁻¹ and 86.7% [37].

The Belete-Gera Forest is one of the largest remaining tracts of forest in south-western
Ethiopia. It is a typical Afromontane evergreen forest, dominated by tree species like
*Syzigium guineense*, *Olea welwitschii*, *Prunus africana* and *Pouteria adolfi-friederici* , with
over 157 plant species representing 69 families and 135 genera [38,39]. Its animal diversity
has never been thoroughly studied, but during a one month pilot study, 25 mammal, 16
amphibian, 5 reptile, 126 bird and 87 butterfly species have been identified [40]. The forest is
considered as a regional forest priority area and “Key Biodiversity Area” (KBA). Regardless
of this, deforestation and habitat modification continues in the region.

2.2 Study species

We focussed our study on **ten bird species** (Table 1). We were not able to compare respective
biomarkers in each species separately, because for several species we failed to collect data in
one of the two forest types (Tab. 1). Therefore, and since Buechley et al., [34] ~~already~~
showed that the abundance of birds of the forest-specialist-guild was negatively affected by
shade coffee farming, we also assigned our study species according to ecological properties
into four functional guilds. Each species was assigned to two of four functional guilds based
on their predominant habitat utilization (forest specialist versus forest generalist [41-47]) and
the proportion of invertebrates in their diet (obligate insectivore species OBI, with a diet
comprising of $\geq 80\%$ invertebrates and opportunistic insectivore species OPI, with $< 80\%$
invertebrate diet based on Elton Trait database (Table 1, [48]).

2.3 Data collection

**From December 2013 to June 2015, we captured 488 birds** belonging to ten species using
standardized ~~mist-nets~~. We opened the nets 30 minutes before sunrise and kept them open for
six hours. We ~~controlled~~ the nets routinely at 30-min intervals ~~to remove birds without~~

~~holding them too long~~. We released the birds immediately after we collected feather samples
and took the measurements on the spot. Juvenile birds were released immediately without
taking any measurement. We ringed the birds with Kenyan rings to recognize them in case
they were caught a second time.

2.3.1 Measurement of tarsi length and rectrix mass

Fluctuating asymmetry was determined from two traits, tarsi length and rectrix mass.
Different traits have different susceptibility for FA based on cost of growth of the trait,
defined as the amount of structural components necessary to form a unit of length of a given
character [49] hence we used both traits. Tarsi length, distance from the back of inter-tarsal
joint to the point where the toes bending an angle of 90 degree to the tarsus [50], was
measured in the field to the nearest 0.1 mm using slide callipers. Two independent
measurements of right and left tarsus lengths for each individual (sequence right-left-right-
left or left-right-left-right) were taken. Second outer most, (the fifth) right and left ~~rectri~~
(fully-grown) feathers were collected from every bird captured for rectrix mass
measurements [29]. The feathers were stored dry in paper envelopes and transported to the
laboratory, and the rectrices were measured with a calliper and weighted twice using an
analytical balance to the nearest 0.1 mg.

2.3.2 Measurement of growth bar widths

Growth bar widths and total feather lengths were measured by one of us (GG) as follow: (1)
each unwearied feather was pinned on separate, white, polystyrene boards and total feather
length was measured to the nearest 0.01 mm with a digital calliper. (2) Each feather was
marked at a distance of 7/10 from its proximal end, and the proximate and distal ends of 5
consecutive growth bars were marked with ultrafine mounting pins. (3) Each marked board

was scanned (Oce OP1130) and growth bar widths and total feather lengths were
automatically measured with an image-analysis-software (KS400 Zeiss). The entire
procedure was repeated for 66 randomly selected feathers to assess the repeatability of our
measurements.

13 171 2.4 Data validation

Possible data outliers were identified using *Proc Univariate* procedure (in SAS) to ensure
data quality for all measurements (tarsus length, feather mass, feather length and average
growth bar widths), whereby lower and higher extreme observation values were screened out
from the data.

The level of repeatability for the growth bar widths was calculated according to Lessells
and Boag [51], whereby within-group and between group mean squares were obtained from
1-way ANOVAs, and the p-values were obtained from F-tests. Repeatability estimates were
high, both for total feather length (right trait side: $r = 0.999$, $N = 66$, $P = <0.001$; left trait side:
$r = 0.999$, $N = 66$, $P < 0.001$) and for average growth bar widths (right trait side: $r = 0.992$, N
$= 66$, $P < 0.000$; left trait side: $r = 0.983$, $N = 66$, $P < 0.001$).

Since the data of the measured traits (unsigned FA values for tarsus length and rectrix
mass, average growth bar widths and total feather lengths) deviated from respective normal
distributions (Kolmogorov-Smirnov test; see Table S1), we log-transformed the data to meet
normality assumptions before further analysis.

187 2.5 Data analyses

188 2.5.1 Analysis of fluctuating asymmetry

Individual signed FA for tarsus length and rectrix mass were quantified as difference in
measures between right (R) and left (L) (using average values obtained from the two

replicated measurements of each side) divided by individual mean trait size to neutralized
trait size effect [52]. Individual unsigned FA (the magnitude of signed FA) values were the
absolute values of signed FA.

**The levels of significance of signed FA and its repeatability** in the measured traits were
calculated by the restricted maximum likelihood (REML) estimation of a mixed regression
model where we fitted the models to the repeated measurements of right and left trait sides
[53]. We included sides as fixed effects in the models and measured trait values (tarsi lengths
or rectrix masses) as response variables, intercept and **sides** as a random effects and
individual id as a subject. This procedure allows the separation of measurement error (ME)
from bilateral asymmetry analysis [53]. The presence of directional asymmetry (DA) was
assessed by the F-test of the fixed effects, with degrees of freedom corrected for statistical
dependence by Satterthwaite formulas [54]. We used random intercepts and slopes (both
estimated within individuals) to estimate the variation in individual trait value and the
individual signed FA, respectively, while the random error variance component gave the
measurement error. The significance of FA was then calculated by performing a likelihood
ratio (LR) test comparing two models: the original, full model and a reduced model without
the side (left or right) as a random effect. Then, intra-class correlation coefficients (ICC) were
calculated to evaluate the repeatability of FA [53].

Absence of anti-symmetry, i.e. a type of asymmetry that occurs in a random direction,
producing a bimodal distribution over the population, was verified by testing the signed FA
for normal distribution.

2.5.2 Analysis of unsigned FA

A linear mixed model was fitted to the data to test whether unsigned FA levels in tarsus
length differed between birds of shade coffee forests and birds of natural forests in relation to:

(1) their forest dependency type (forest specialist or forest generalist), and (2) their diet
composition (obligate insectivores or opportunistic insectivores) whereby unsigned FA was
included as a response variable, forest type as a fixed effect and each sampling location as a
random effect.

13 221 2.5.3 Analysis of growth bar width

To test whether average growth bar widths differed between birds of shade coffee forests and
those of natural forests, we added average growth bar widths in the models as a response
variable, forest type and total feather lengths as a fixed effects, and **sampling location as a**
**225 random effect.** All statistical analyses were performed with program SAS (Version 9.4., SAS
Institute 2013, Cary, NC, USA).

228 **3. Results**

We retrieved measures of tarsus length from 380 birds, and measures of rectrix mass and
average growth bar width from 344 birds (Table 1). For tarsus length, kurtosis ranges from K
= 4.20 in FF to K = 12.65 in OPI, for rectrix mass, it ranges from K = 49.76 in FF to K =
121.96 in OBI (Table S2). The signed FA of tarsus length and rectrix mass showed
leptokurtic distributions for all of the four guilds ($K > 3.0$) (Kolmogorov-Smirnov test; all $P <$
0.01; Table S1), which indicates heterogeneity in developmental stability between individuals
(53.55). DA was detected for tarsus length ($P = 0.0003$), but not for rectrix mass ($P = 0.7886$;
Table S3).

238 3.1 Fluctuating asymmetry

The signed FA variance estimation was highly significant relative to measurement error in
tarsus length ($\chi^2 = 410.9$, $P < 0.0001$), but the level of within-side measurement error was

high compared to between-side difference (ICC = 2.8 %). On the other hand, the signed FA
variance estimation was not significant relative to measurement error in rectrix mass ($\chi^2 = 1.7$,
$P = 0.1930$), and level of within-side measurement error was very high compared to between-
side difference (ICC = 5.12×10^{-4} %; Table S3). Hence, we did not include FA of rectrix mass
in further analyses.

16 247 3.2 Growth bar width

Average growth bar width of birds of the F, OBI and OPI guilds did not differ between shade
coffee forests and natural forests (for all three guilds $P \geq 0.0598$; Table 3). However, average
growth bar width for birds of the FF guild differed significantly between the two forest types
($P < 0.02$) whereby birds of FF guild from shade coffee forests had narrower growth bars
than birds of this guild from natural forests. While the difference is not significant, birds of
the OBI guild from shade coffee forests showed reduced growth bar width compared to birds
of the same guild residing in natural forest (Fig.3).

35 256 4. Discussion

We applied biomarkers (FA in tarsus length and rectrix mass, and growth bar width and total
feather growth) to re-assess the quality of the Ethiopian tropical rain forest which was
reported to support rich avian biodiversity comparable to the natural forest except forest
specialist and understorey insectivore birds [34].

Birds did not show increased levels of fluctuating asymmetry in tarsus length
irrespective of guild membership and whether they were captured in shade coffee forests or
natural forest. Similarly, the analysis of feather growth bar widths revealed no difference in
among birds from shade coffee forests or natural forests with the exception of birds of the
forest specialist guild (FF), which showed lower growth bar width in shade coffee forests.

Also in the study by Buechley et al. [34] birds of the FF guild were found to be more
negatively affected than birds of other guilds. Species respond to habitat modifications
differently. Some species are more sensitive than others for various reasons, including
ecological specialization, diet and reproductive capacity [56,57]. This puts, most likely, forest
specialist birds at high risk of extinction [58-61]. About 21% of bird species of which most
are forest specialist are currently extinction-prone and 6.5% are functionally extinct [62].
Shade coffee farm management means selective removal of trees species, which may
decrease habitat quality for forest specialist birds more than for other avian guilds [5,9,63].
Other than for birds, Ethiopian shade coffee farming has been reported to be useful for forest
tree conservation but its conservation value strongly depends on the coverage of canopy of
overstorey shade trees which varied across shade coffee plantations [12]. Other studies
showed that shade coffee farming is incompatible with epiphytic orchid conservation [64].
While the importance of the Ethiopian shade coffee farming for biodiversity conservation is
undeniable, its conservation value could not replace the natural forest. Hence, the promotion
of shade coffee farming should emphasise the conversion of degraded forest into shade coffee
forests but not converting natural forest into shade coffee farms. No human modified habitat
has been found to substitute primary forests for biodiversity conservation in general [65].

Our study showed that fluctuating asymmetry (FA) of tarsi length is unaffected by
shade coffee farming in forest specialist birds, whereas birds of this guild have reduced
growth bar widths in shade coffee forests. Only few studies combined FA and feather growth
286 bar width analysis which showed the two markers showed consistent result or only one of the
287 markers show the effect of fragmentation (21,29,66). We recommend combining the two
biomarkers to assess effects of habitat fragmentation. We skipped the FA analysis for rectrix
mass as within-side measurement error was higher than the between-side difference. FA is a
sensitive measure of habitat quality that needs rigorous test for the validity of the data used

for the analysis, and could be very useful as early warning clue for a negative effects of
habitat change before any demographic change can be observed [67].

Currently, at least in our study area, the negative effects of converting natural forest
into shade coffee forests on bird communities are not dramatic. However, in many coffee
growing countries across the globe, shade coffee management practices are increasingly
intensified, which is believed to reduce plant diversity and canopy cover [68-71], and hence,
also reducing the quality or quantity of feeding, nesting or hiding resources for animal
populations. Such tendencies urge a close monitoring of habitat quality of shade coffee
forests with biomarkers which can detect changes of habitat quality before it is degraded to
the extent that it affects avian diversity and community demography [14,67,72,73].

5. Conclusion

Our results are consistent with results of demographic studies that shade coffee farming in the
Ethiopian highland can be compatible with avian diversity conservation. No early sign of
negative effect was observed from biomarker studies different from the demographic changes
reported by Buechely et al. [34]. However, results of our study in combination with Buechely
et al. [34] show that shade coffee farming has negative effects on the guild of forest
specialists, i.e. habitat quality for these bird species seems to decrease. Therefore, the current
promotion of shade coffee farming as means of biodiversity conservation should not lead to
the conversion of natural forests into shade coffee forests. Shade coffee can be used as an
important conservation tool whenever preserving natural forest is impossible for various
reasons, but it should not replace it. Combining FA and feather growth rate as biomarkers
provide more comprehensive results in studies of habitat quality for avian biodiversity
conservation than using only one of them.

Ethics: This project was carried out in accordance to the ethical standards for research from
Jima University, Ethiopia and Ethiopian Wildlife Conservation Authority (EWCA), and the
project was endorsed by EWCA.

Data accessibility: Data available at the Dryad Digital Repository

<https://datadryad.org/review?doi=doi:10.5061/dryad.cb0d8ft>

Authors' contributions: GG did the fieldwork and collected the data. GG and AA drafted a
first version of the manuscript which was improved by DZ. GG, DT and NB did the statistical
analyses. All authors reviewed the manuscript and gave their final approval for publication.

Competing interests: We declare we have no competing interests.

Funding: This research is funded by VLIR IUC-JU project (VLIR-UOS Institutional

University Cooperation programme between Jimma University and various Flemish

universities under the umbrella of the Flemish Interuniversity Council).

Acknowledgement: We are grateful to Bahir and Reshad Abafita and Jewad Abazinab for

assisting us during our field work and all VLIR IUC-JU divers. We thank Luc Lens for

important comments on improving the proposal and designing the field work. We thank

Kasahun Eba at VLIR IUC-JU programme office at Jimma University for his support though

out our field work period. We thank Alexander von Humboldt Foundation for support given

to Anagaw Atickem.

**References**

1. Gardner T.A., Barlow J., Chazdon R., Ewers R. M., Harvey C. A., Peres C.A., Sodhi N. S.,

2009. Prospects for tropical forest biodiversity in a human-modified world. *Ecol. Lett.*

**12**, 561–582. (doi: 10.1111/j.1461-0248.2009.01294.x)

2. FAO, 2010. Global Forest Resources Assessment (FRA) 2010. Key findings Food and

Agriculture Organization of the United Nations, Rome.

<http://www.fao.org/docrep/013/i1757e/i1757e.pdf>

3. Lawrence, D., Vandecar, K., 2015. Effects of tropical deforestation on climate and

agriculture. *Nat. Clim. Change* **5**, 27–36. (doi: 10.1038/NCLIMATE2430)

4. Ranganathana, J., Daniels, R.J.R., Chandranc, M.D.S., Ehrlich, P.R., Dailya, G.C., 2008.
Sustaining biodiversity in ancient tropical countryside. *Proc. Natl. Acad. Sci. USA* **105**,
17852–17854. (doi: 10.1073/pnas.0808874105)
5. Rappole, J.H., King, D.I., Vega Rivera, J.H., 2003. Coffee and conservation. *Conserv. Biol.*
**17**, 334–336. (doi: 10.1046/j.1523-1739.2003.01548.x)
6. Caudill S.A., DeClerck F.J.A., Husband T.P., (2015). Connecting sustainable agriculture
and wildlife conservation: Does shade coffee provide habitat for mammals? *Agric.*
*Ecosyst. Environ.* **199**, 85–93. (doi: 10.1016/j.agee.2014.08.023)
7. Kinasih, I., Cahyanto, T., Widiana, A., Kurnia, D. N. I., Julita, U., Putra, R.E., 2016. Soil
invertebrate diversity in coffee-pine agroforestry system at Sumedang, West Java.
*Biodiversitas* **17**, 473–478. (doi: 10.13057/biodiv/d170211)
8. Perfecto, I., Mas, A., Dietsch, T., Vandermeer, J., 2003. Conservation of biodiversity in
coffee agroecosystems: a tri-taxa comparison in southern Mexico. *Biodiv. Conserv.* **12**,
1239–1252. (doi: 10.1023/A:1023039921916)
9. Tejeda-Cruz, C., Sutherland, W.J., 2004. Bird responses to shade coffee production. *Anim.*
*Conserv.* **7**, 169–179. (doi: 10.1017/S1367943004001258)
10. López-Gómez, A.M., Williams-Linera, G., Manson, R.H., 2008. Tree species diversity
and vegetation structure in shade coffee farms in Veracruz, Mexico. *Agric. Ecosyst.*
*Environ.* **124**, 160–172. (doi: 10.1016/j.agee.2007.09.008)
11. Leyequien, E., de Boer, W.F., Toledo, V.M., 2010. Bird community composition in a
shaded coffee agro-ecological matrix in Puebla, Mexico: The effects of landscape
heterogeneity at multiple spatial scales. *Biotropica* **42**, 236–245. (doi: 10.1111/j.1744-
7429.2009.00553.x)

12. Tadesse, G., Zavaleta, E., Shennan, C., 2014. Coffee landscapes as refugia for native
woody biodiversity as forest loss continues in southwest Ethiopia. *Biol. Conserv.* **169**,
384–391. (doi: 10.1016/j.biocon.2013.11.034)
13. Stearns, S. C., 1992. *The evolution of life histories*. Oxford University Press, Oxford, UK.
14. Johnson, M.D., 2007. Measuring habitat quality: a review. *Condor* **109**, 489–504. (doi:
10.1650/8347.1)
15. Nagelkerke K.C.J., Verboom J., van den Bosch F., van de Wolfshaar K., 2002. Time lags
in metapopulation responses to landscape change. In *Concepts and applications of land*
*scape ecology in biological conservation* (ed. KJ Gutzwiller), Springer, pp. 330–354.
16. Uezu, A., Metzger, J.P., 2016. Time-lag in responses of birds to Atlantic Forest
fragmentation: restoration opportunity and urgency. *PLoS ONE* **11**, e0147909. (doi:
10.1371/journal.pone.0147909)
17. Lens, L., Eggermont, H., 2008. Fluctuating asymmetry as a putative marker of human-
induced stress in avian conservation. *Bird Conserv. Int.* **18**, 125–143. (doi:
10.1017/S0959270908000336)
18. Huggett, R.J., Kimerle, R.A., Mehrle, Jr. PM., Bergman, H.L., 1992. *Biomarkers:*
*biochemical, physiological and histological markers of anthropogenic stress*. Boca
Raton, FL, Lewis.
19. Leung, B., Knopper, L., Mineau, P., 2001. *A critical assessment of the utility of*
*fluctuating asymmetry as a biomarker of anthropogenic stress*. In *Developmental*
*instability: causes and consequences* (ed. M Polak), pp. 415–426. Oxford, UK, Oxford
University Press.
20. Van Valen, L., 1962. A study of fluctuating asymmetry. *Evolution* **16**, 125–142. (doi:
10.1111/j.1558-5646.1962.tb03206.x)

21. Møller, A.P., Manning, J.T., 2003. Growth and developmental instability. *Vet. J.* **166**, 19–
27. (doi: 10.1016/S1090-0233(02)00262-9)
22. De Coster, G., van Dongen, S., Malaki, P., Muchane, M., Alcántara-Exposito, A.,
Matheve, H., Lens, L., 2013. Fluctuating asymmetry and environmental stress:
understanding the role of trait history. *PLoS ONE* **8**, e57966. (doi:
10.1371/journal.pone.0057966)
23. Costa, R. N., Solé, M., Nomura, F., 2017. Agro pastoral activities increase fluctuating
asymmetry in tadpoles of two neotropical anuran species. *Austral. Ecol.* **42**, 801-809.
(doi: 10.1111/aec.12502)
24. Grubb, T.C., 1989. Ptilochronology: Feather growth bars as indicators of nutritional status.
*Auk* **106**, 314–320.
25. Grubb, T.C. Jr., Cimprich, D.A., 1990. Supplementary food improves the nutritional
condition of wintering woodland birds: evidence from ptilochronology. *Ornis Scand.*
**21**, 277–281. (doi: 10.2307/3676392)
26. Grubb, T.C. Jr., Yosef, R., 1994. Habitat specific nutritional condition in loggerhead
shrikes (*Lanius ludovicianus*): evidence from ptilochronology. *Auk* **111**, 756–759.
27. Grubb, T.C., Jr., 2006. *Ptilochronology: feather time and the biology of birds*. Oxford,
UK, Oxford University Press.
28. Møller, A.P., 1997. Developmental stability and fitness: a review. *Am. Nat.* **149**, 916–932.
29. Carbonell, R., Tellería, J.L., 1999. Feather traits and ptilochronology as indicators of
stress in Iberian blackcaps *Sylvia atricapilla*. *Bird Study* **46**, 243–248. (doi:
10.1080/00063659909461136)
30. Eeva, T., Tanhuanpää, S., Råbergh, C., Airaksinen, S., Nikinmaa, M., Lehikoinen E.,
(2000). Biomarkers and fluctuating asymmetry as indicators of pollution-induced stress

in two hole-nesting passerines. *Funct. Ecol.* **14**, 235–224. (doi: 10.1046/j.1365-
2435.2000.00406.x)
31. Mittermeier, R.A., Gil, P.R., Hoffman, M., Pilgrim, J., Brooks, T., Mittermeier, C.G.,
Lamoreux, J., Da Fonseca, G.A.B., 2004. *Hotspots Revisited: Earth's Biologically*
*Richest and Most Endangered Terrestrial Ecoregions*. CEMEX, Conservation
International, and Agrupación Sierra Madre, Monterrey, Mexico
32. Kufa, T., Burkhardt, J., 2011. Plant composition and growth of wild *Coffea arabica*:
Implications for management and conservation of natural forest resources. *Int. J.*
*Biodivers. Conserv.* **3**, 131–141.
33. Reichhuber, A., Requate, T., 2012. Alternative use systems for the remaining Ethiopian
cloud forest and the role of Arabica coffee - A cost-benefit analysis. *Ecol. Econ.* **75**,
102–113. (doi: 10.1016/j.ecolecon.2012.01.006)
34. Buechley, E.R., Şekercioğlu, Ç.H., Atickem, A., Gebremichael, G., Ndungu, J.K., Abdu,
B., Beyene, T., Mekonnen, T., Lens, L., 2015. Importance of Ethiopian shade coffee
farms for forest bird conservation. *Biol. Conserv.* **188**, 50–60. (doi:
10.1016/j.biocon.2015.01.011)
35. Braziotis, S., Liordos, V., Bakaloudis, D.E., Goutner, V., Papakosta, M.A., Vlachos, C.G.,
2017. Patterns of postnatal growth in a small falcon, the lesser kestrel *Falco naumanni*
(Fleischer, 1818) (Aves: Falconidae). *Eur. Zool. J.* **84**, 277–285. (doi:
10.1080/24750263.2017.1329359)
36. Tayefeh, F.H., Amini, H., Khaleghizadeh, A., 2016. *Chick growth patterns of three*
*sympatric tern species on the Persian Gulf Islands*. Bird Numbers 2016-Birds in a
changing World Conference, Halle, Germany.
37. Hundera, K., Aerts, R., Fontaine, A., Van Mechelen, M., Gijbels, P., Honnay O., Muys,
B., 2013a. Effects of coffee management intensity on composition, structure, and

regeneration status of Ethiopian moist evergreen Afromontane forests. *Environ. Manag.*
**51**, 801–809. (doi: 10.1007/s00267-012-9976-5)
38. Gebrehiwot, K., Hundera, K., 2014. Species composition, plant community structure and
natural regeneration status of Belete moist evergreen montane forest, Oromia Regional
State, Southwestern Ethiopia. *SINET* 6, 97–101.
39. Demissew, S., Cribb, P., Rasmussen, F., 2004. *Field guide to Ethiopian orchids*. Royal
Botanic Gardens, Kew, UK.
40. De Beenhouwer, M., Aerts, R., Honnay, O., 2013. A global meta-analysis of the
biodiversity and ecosystem service benefits of coffee and cacao agroforestry. *Agric.*
*Ecosyst. Environ.* **175**, 1-7. (doi: 10.1016/j.agee.2013.05.003)
41. Brown, L. H., Urban, E. K. & Newman, K. (1982). The birds of Africa, Volume I, pp.
521. Academic Press, London
42. Urban, E. K., Fry, C. H. & Keith, S. (1986). The birds of Africa, Volume II, pp. 552.
Academic Press, London.
43. Fry, C. H., Keith, S. & Urban, E. K. (1988). The birds of Africa, Volume III, pp. 611.
Academic Press, London.
44. KEITH, S., URBAN, E. K. & FRY, C. H. (1992). The birds of Africa, Volume IV, pp.
632. Academic Press, London. London.
45. Fry, C. H., Keith, S. & Urban, E. K. (1997). The birds of Africa, Volume V, pp. 672.
Academic Press, London.
46. Fry, C. H., Keith, S. & Urban, E. K. (2000). The birds of Africa, Volume VI, pp. 704.
Academic
47. Fry, C. H. & Keith, S. (2004). The birds of Africa, Volume VII, pp. 666. Christopher
Helm, London.

48. Wilman, H., Belmaker, J., Simpson, J., de la Rosa, C., Rivadeneira, M.M., Jetz, W., 2014.
EltonTraits 1.0: Species-level foraging attributes of the world's birds and mammals.
*Ecology* **95**, 2027. (doi: 10.1890/13-1917.1)
49. Aparicio, J.M., Bonal, R., 2002. Why do some traits show higher fluctuating asymmetry
than others? A test of hypotheses with tail feathers of birds. *Heredity* **89**, 139–144. (doi:
10.1038/sj.hdy.6800118)
50. Svensson, L. (1992). *Identification Guide of European Passerines*. 4th ed. Svensson,
Stockholm.
51. Lessells, C.M., Boag, P.T., 1987. Unrepeatable repeatabilities: a common mistake. *Auk*
**104**, 116–121. (doi: 10.2307/4087240)
52. Palmer, A.R., Strobeck, C., 2003. Fluctuating Asymmetry analyses revisited. In
*Developmental Instability: Causes and Consequences* (ed. M Polak), pp. 279–319.
Oxford University Press, Oxford.
53. Van Dongen, S., Molenberghs, G., Matthysen, E., 1999. The statistical analysis of
fluctuating asymmetry: REML estimation of a mixed regression model. *J. Evol. Biol.*
**12**, 94–102. (doi: 10.1046/j.1420-9101.1999.00012.x)
54. Verbeke, G., Molenberghs, G., 2000. *Linear Mixed Models for Longitudinal Data*.
Springer, New York.
55. Palmer, A.R., Strobeck, C., 1992. Fluctuating asymmetry as a measure of developmental
stability: implications of non-normal distributions and power of statistical tests. *Acta*
*Zool. Fenn.* **191**, 57–72.
56. Posa, M.R.C., Sodhi, N.S., 2006. Effects of anthropogenic land use on forest birds and
butterflies in Subic Bay, Philippines. *Biol. Conserv.* **129**, 256–270. (doi:
10.1016/j.biocon.2005.10.041)

57. Keinath, D.A., Doak, D.F., Hodges, K.E., Prugh, L.R., Fagan, W., Sekercioglu, C.H.,
Buchart, S.H.M., Kauffman M., 2017. A global analysis of traits predicting species
sensitivity to habitat fragmentation. *Global Ecol. Biogeogr.* **26**, 115–127. (doi:
10.1111/geb.12509)
58. Owens, I.P.F., Bennett P.M., 2000. Ecological basis of extinction risk in birds: habitat
loss versus human persecution and introduced predators. *Proc. Natl. Acad. Sci. USA* **97**,
12144–12148. (doi: 10.1073/pnas.200223397)
59. Şekerciöğlü, C.H., 2007. Conservation ecology: area trumps mobility in fragment bird
extinctions. *Curr. Biol.* **17**, 283–286. (doi: 10.1016/j.cub.2007.02.019)
60. Şekerciöğlü, C.H., Ehrlich, P.R., Daily, G.C., Aygen, D., Goehring, D., Sandi, R., 2002.
Disappearance of insectivorous birds from tropical forest fragments. *Proc. Natl. Acad.*
*Sci. USA* **99**, 263–267. (doi: 10.1073/pnas.012616199)
61. Tschamtkke, T., Şekerciöğlü, C., Dietsch, T., Sodhi, N., Hoehn, P., Tylianakis, J.M., 2008.
Landscape constraints on functional diversity of birds and insects in tropical
agroecosystems. *Ecology* **89**, 944–951. (doi: 10.1890/07-0455.1)
62. Şekerciöğlü C.H., Daily G.C. and Ehrlich P.R. 2004. Ecosystem consequences of bird
declines. *Proc. Natl. Acad. Sci. USA* **101**, 18042–18047. (doi:
10.1073/pnas.0408049101)
63. Schmitt, C.B., Senbeta, F., Denich, M., Preisinger, H., Boehmer, H.J., 2010. Wild coffee
management and plant diversity in the montane rainforest of south-western Ethiopia.
*Afr. J. Ecol.* **48**, 78–86. (doi: 10.1111/j.1365-2028.2009.01084.x)
64. Hundera, K., Aerts, R., De Beenhouwer, M., Van Overtveld, K., Helsen, K., Muys, B.,
Honnay, O., 2013b. Both forest fragmentation and coffee cultivation negatively affect
epiphytic orchid diversity in Ethiopian moist evergreen Afromontane forests. *Biol.*
*Conserv.* **159**, 285–291. (doi: 10.1016/j.biocon.2012.10.029)

65. Gibson, L., Lee, T.M., Koh, L.P., Brook, B.W., Gardner, T.A., Barlow, J., Peres, C.A.,
Bradshaw, C.J.A., Laurance, W.F., Lovejoy, T.E., Sodhi, N.S., 2011. Primary forests
are irreplaceable for sustaining tropical biodiversity. *Nature* **478**, 378–381. (doi:
10.1038/nature10425)
- 66. Polo, V., Carrascal, L.M., 1999. Ptilochronology and fluctuating asymmetry in tail and
wing feathers in coal tits *Parus ater*. *Ardeola* **46**, 195–204.
- 67. Lens, L., van Dongen, S., Matthysen, E., 2002. Fluctuating asymmetry as an early
warning system in the critically endangered Taita thrush. *Conserv. Biol.* **16**, 479–487.
(doi: 10.1046/j.1523-1739.2002.00516.x)
- 68. Armbrrecht, I., 2003. Habitat changes in Colombian coffee farms under increasing
management intensification. *Endang. Spec. Update* **20**, 4–5.
- 69. Aerts, R., Hundera, K., Berecha, G., Gijbels, P., Baeten, M., Van Mechelen, M., Hermy,
525 M., Muys, B., Honnay, O., 2011. Semi-forest coffee cultivation and the conservation of
526 Ethiopian Afromontane rainforest fragments. *Forest. Ecol. Manag.* **261**, 1034–1041.
(doi: 10.1016/j.foreco.2010.12.025)
- 70. Noponen, M.R.A., Haggar, J.P., Edward-Jones, G., Healey, J.R., 2013. Intensification of
coffee systems can increase the effectiveness of REDD mechanisms. *Agric. Syst.* **119**,
1–9. (doi: 10.1016/j.agsy.2013.03.006)
- 71. Aerts R., Berecha G., Honnay, O., 2015. Protecting coffee from intensification. *Science*
**347**, 139. (doi: 10.1126/science.347.6218.139-b)
- 72. Beasley, D.A.E., Bonisoli-Alquati, A., Mousseau, T.A., 2013. The use of fluctuating
asymmetry as a measure of environmentally induced developmental instability: A
meta-analysis. *Ecol. Indic.* **30**, 218–226. (doi: 10.1016/j.ecolind.2013.02.024)
- 73. Helle, S., Huhta, E., Suorsa, P., Hakkarainen, H., 2011. Fluctuating asymmetry as a
biomarker of habitat fragmentation in an area-sensitive passerine, the Eurasian

treecreeper (*Certhia familiaris*). *Ecol. Indic.* **11**, 861–867. (doi:
10.1016/j.ecolind.2010.11.004)
74. Gebremichael, G., Tsegaye D., Bunnefeld N., Zinner D., Atickem A., 2018. Data from:
Fluctuating asymmetry and feather growth rate as biomarkers to assess habitat quality of
shade coffee farming for avian biodiversity conservation. Dryad Digital Repository. (doi:
10.5061/dryad.cb0d8ft)

Fig. 1. Geographic positions of study sites in natural forests (Muje, Afalo, Ababa Buna, Qacho) and in the shade coffee farm area (Yebu, Garuke, Fetche) in the highlands of western Ethiopia.

364x258mm (96 x 96 DPI)

Fig. 2. Average fluctuating asymmetry (FA) of tarsi length in birds of four functional guilds from shade coffee forests and natural forests (guilds: OPI = generalist insectivores; OBI = obligate insectivores; F = forest generalists; FF = forest specialists).

Fig. 3. Average widths of feather growth bars of birds of four functional guilds from shade coffee forests and natural forests (guilds: OPI = generalist insectivores; OBI = obligate insectivores; F = forest generalists; FF = forest specialists).

Table 1. The number of birds assessed for growth bar widths and tarsus length per species and forest type (natural forest [NF], shade coffee forest [SC]). Species are grouped into functional guilds (forest dependency types [FD]: forest generalist [F] and forest specialist [FF]; feeding habits [DT]: obligate insectivores [OBI] and opportunistic insectivores [OPI]).

species	guilds		growth bar width			tarsus length		
	DT	FD	SC	NF	total	SC	NF	total
Melaenornis chocolatinus	OBI	F	46	1	47	40	5	45
Muscicapa adusta	OBI	F	19	0	19	10	17	27
Terpsiphone viridis	OBI	F	20	0	20	16	9	25
Camaroptera brevicaudata	OBI	F	14	0	14	12	9	21
Cossypha semirufa	OBI	FF	51	45	96	65	35	100
Turdus pelios	OPI	F	7	11	18	12	9	21
Turtur tympanistria	OPI	F	33	44	77	38	46	84
Pogoniulus chrysoconus	OPI	F	9	6	15	16	6	22
Geokichla piaggiae	OPI	FF	0	15	15	6	9	15
Pseudoalcippe abyssinica	OPI	FF	0	18	18	5	15	20
complete sample			199	140	339	220	160	380

Table 2. Unsigned FA estimates in tarsus length for bird community in two habitat types (natural forest and shade coffee farms) based on their forest dependency and diet. Linear mixed models were fitted to the data where unsigned FA were included in the models as a response variable, forest types as fixed effect and each sampling location as random effect for forest specialist (FF), forest generalist (F), and opportunistic insectivores (OPI) and obligate insectivore (OBI).

Community	Effect	Estimate	se	df	t	p
FF	Intercept	-6.339	0.514	6.250	-12.340	<0.0001
	Forest type (shade coffee)	0.982	0.684	6.670	1.430	0.1965
F	Intercept	-5.814	0.663	8.180	-8.760	<0.0001
	Forest type (shade coffee)	1.728	0.857	8.600	2.020	0.0760
OPI	Intercept	-6.176	0.618	6.080	-10.000	<0.0001
	Forest type (shade coffee)	1.788	0.849	7.710	2.110	0.0700
OBI	Intercept	-6.266	0.805	7.130	-7.790	<0.0001
	Forest type (shade coffee)	1.621	1.016	7	1.590	0.1549

Table 3. Estimates of average growth bar width from mixed linear models, where average growth bar widths included as a response variable, forest types as fixed effect, whereby feather length included as fixed covariate and each sampling location as random effect for forest specialist (FF), forest generalist (F), and opportunistic insectivores (OPI) and obligate insectivore (OBI).

Community	Effect	Estimate	se	df	t	p
FF	Intercept	0.057	0.201	121	0.29	0.7761
	Forest type(shade coffee)	-0.675	0.279	127	-2.42	0.0171
	Feather length	0.595	0.110	122	5.40	<0.0001
	Forest type(shade coffee): Feather length	0.365	0.154	127	2.37	0.0191
F	Intercept	-0.467	0.107	209	-4.35	<0.0001
	Forest type(shade coffee)	0.012	0.144	209	0.09	0.9312
	Feather length	0.911	0.058	207	15.79	<0.0001
	Forest type (shade coffee): Feather length	-0.029	0.078	209	-0.38	0.7060
OPI	Intercept	-0.323	0.132	143	-2.45	0.0156
	Forest type (shade coffee)	-0.339	0.179	143	-1.90	0.0598
	Feather length	0.835	0.070	142	11.98	<0.0001
	Forest type (shade coffee): Feather length	0.177	0.095	143	1.87	0.0630
OBI	Intercept	-0.207	0.118	193	-1.76	0.0796
	Forest type (shade coffee)	0.225	0.148	193	1.52	0.1308
	Feather length	0.736	0.066	193	11.15	<0.0001
	Forest type (shade coffee): Feather length	-0.130	0.083	193	-1.58	0.1161

Appendix B

Dear Sirs,

Thank you for your review and for letting us resubmit. We found the comments helpful and constructive.

We incorporate comments and suggestions of the reviewers in the new version of manuscript. We also did additional editorial work on top of the reviewers comment. The main concern of the reviewer was the analysis of fluctuating symmetry of the avian community on the bases of functional guilds, diet and forest specialization. We now updated the analysis of fluctuating asymmetry on the bases of species residing in the two habitat types as suggested by the reviewers.

Please find more detail of the changes we have made,

Sincerely,

Gelaye Gebremichae

Reviewer(s)' Comments to Author:

Reviewer: 1

Reviewers Comments to Author:

Reviewer: 1

Comments to the Author(s)

Overview:

This is a very interesting concept, applying morphometrics and fluctuating asymmetry in a fairly novel fashion to study potential impacts of environmental quality of shade coffee and natural forest on bird communities in SW Ethiopia. The study is novel and has the potential to make a significant and important contribution to biodiversity conservation in Ethiopia, as well as to encourage similar studies worldwide to evaluate early indicators of environmental stresses on populations. On this level, I think it is suitable for publication in Royal Society Open Science.

However, I also have some significant methodological concerns with the manuscript that I hope can be addressed prior to publication. My biggest concern is regarding sample size and guild classifications, as best outlined in Table 1. Is it really suitable to compare fluctuating asymmetry and feather growth bar width of different species, as lumped into very general guilds?

Our reply: Our updated version consider only species comparison between natural forest and shade coffee habitat instead of the comparing avian community based on functional guilds.

The difference in samples sizes of species between shade coffee and natural forest for growth bar width measurements seems particularly great. For example the top 4 species were almost exclusively sampled in shade coffee. Thus how can you compare differences between habitats??? I'm not sure that lumping them into guilds addresses this underlying issue in the dataset. If you have a strong argument for why this is valid to compare, I think it should be very prominently laid out in the methods and discussion. As is, there is really almost no discussion of such lack of uneven sampling and how that might impact results. Otherwise, perhaps the analysis should be limited to species that you have a reasonable sample size in both habitat types...

Our reply: Yes, as we mentioned above, we have now updated the analysis based on species comparison

I also have a bit of a concern with the guild classification for obligate insectivore vs opportunistic insectivore, which is based on a seemingly arbitrary 80% insect-diet cut off. Is there no way to make a more meaningful guild distinction, perhaps based on the classifications in Buechley et al 2015 (i.e. insectivore / granivore / frugivore).

Our reply: This problem is also solved as we change the analysis to species level in the updated manuscript version.

A couple of other fairly major comments are as follows:

- English was good throughout, up until the discussion, which could use some detailed work simplifying and correcting grammatical errors.

Our reply: in our updated version, we have improved the language and fix grammatical errors.

- Discussion has several bold claims about coffee forests in Ethiopia that I strongly suggest toning back. This is an interesting study, but I don't think it is definitive enough to say things like “the negative effects of converting natural forest into shade coffee forests on bird communities are not dramatic [line 294]”

Our reply: we agreed and in the new version of our manuscript, we have modified this particular sentence other sentence which appeared to be broad and debatable.

Line 71: “random” – if they are random, then how can they be an index of anything?

Our reply: FA is originally defined as “random deviations from left-right symmetry in bilateral traits [24-27, reference in the main manuscript]”. To our understanding, from large number of the sample size of the given traits, a random sample of the traits potentially can reveal the deviation, either as length or weight measurements.

Line 78: What about using other bio-markers like mass?

Our reply: That is also a possibility and here we used a specific trait, rectrix mass

Lines 81-82: “the origin and reservoir” – i take issue with the wording here, as arabica also originated and is still found in SW Arabia

Our reply: To our knowledge, the origin of the coffee arabica is Ethiopia and it crossed the red sea to Yeman and other Arabian countries around the 7th century.

Line 94: tarsi length can be a hard measurement to standardize – perhaps address how this was done in methods

Our reply: we have now provided in detail of how we measured the Tarsi length in the method section on page 7, lines 147 and 148.

Line 95: Mass of the retrace? This seems an odd measurement!

Our reply: Okay we now replaced it with rectrices weight.

Line 97: replace assumed with hypothesized

Our reply: We have now made the change

Line 118: comma spacing

Our reply: we have now made the change

Line 123: has this directly impacted the study sites? If so how?

Our reply: yes, much of the natural forest is now modified to shade coffee. In the new version of the manuscript, we have tried to explain this further

Line 134: this 80% cutoff seems arbitrary. Could you not find a more well-defined guild separation, perhaps by following the definitions used in Buechley et al 2015?

Our reply: We have now excluded the analysis that based on functional guilds

Line 143: clarify where the rings are from (i.e. museum of nat hist)

Our reply: ring were obtained from National Museum of Kenya, and we now have mentioned this in the manuscript on page 7, line 138.

Line 144: is this the only reason?

Our reply: for this particular research question, yes.

Table 1: I'm concerned about unbalanced sampling of the study species between habitat types. Is it valid to compare fluctuating asymmetry / feather growth across species?

Our reply: we fully acknowledge this concern, and we now change our analysis to a species comparison in the two habitat forms rather than guilds.

Lines 154 – 158: I'm a bit skeptical about this method. Retrix feathers are often soiled by feces. Were the feathers cleaned prior to weighing? If so, how? Retrix feathers also undergo major wear. It is not clear to me that tropical birds would necessarily have the same moult pattern, thus different individuals may be under different levels of feather wear at any given time, which could impact feather mass. I strongly suggest adding more detail of methods and citations validating these indices here.

Our reply: we caught 488 individual birds of the ten species included in the previous analysis. But our analysis was only fixed to feathers from individuals birds with undamaged/unworn and clean feathers (this is the reason why our sample size was small for some species). Any feather those soiled with feces/ worn were not included in the analysis page 7, lines 155 and 156.

Line 162: “unworn” – what does this mean??

Our reply: it is to mean damaged/worn out

Line 194: I like that repeatability of the measurements was included in the model

Our reply: Thanks. This is one of the most important precaution in ptilochronology studies

Line 231: spell out acronyms (e.g. FF, DA, etc) at start of new section

Our reply: we have done as suggested

Lines 241 – 245: was this caused by scale accuracy? If the scale is accurate, it seems this would be the easier measurement to replicate. Can you explain this result more, perhaps in Discussion?

Our reply: As we change the analysis from avian community comparison based on functional guilds to species, some of the results including this section also changed.

Lines 252-254: show p value for this comment

Our reply: This is only descriptive, to show the raw data.

Lines 259 – 260: I would cut this sentence off at “natural forest”. It was shown to support high richness, including of forest specialists and understory insectivores.

Our reply: we did as suggested

Line 261: I would lead with the significant result (FF with lower growth bar width) and then follow with null findings.

Our reply: we have now modified the earlier analysis which was based on guilds to a species level and we modified the discussion accordingly

Line 270-271: “most” – need to add percentage for this claim

Our reply: We have modified the sentence

Lines 278-279: I would rephrase this sentence away from bold claims

Our reply: we have now modified this sentence

Lines 283 – 284: again, I would back away from this overly-bold claim. I suggest rephrasing to something like “our study identified no difference in FA between...”

Our reply: Following our new analysis on FA comparison between the species, we have modified this sentence

Lines 293 – 294: again, I think an overly bold statement. Your study may add some support to this idea, but I wouldn’t state that conversion does not have a dramatic impact.

Our reply: we agree, and we have now modified the sentence

Figure 2 and 3: check lower left heading “insectivore dependy on diet” for misspelling

Our reply: Okay we did that, and corrected the errors

Evan R. Buechley

Reviewer: 2

Comments to the Author(s)

This study aims to assess avian habitat quality between natural forests and shade coffee plantations in Ethiopia via individual measures of functional asymmetry and feather growth. The overall aim of the study is excellent and attempts to address an area where much work is needed. However, I am unable to evaluate the study and thus did not proceed to review the results or discussion for one major reason.

The paper does not clarify the phenological life history of the 10 study species used. This is crucial because the measures of local site habitat quality are entirely dependent on the location of 1) where the measured tarsi were developed as nestlings, and 2) where the measured feathers were grown. For example, if all 10 species were known to breed on site, and it was assumed that the individuals did not disperse to or from elsewhere, then the tarsus growth could be assumed to be influenced by the conditions of the study site. However, if the bird was captured after it had migrated from its breeding location elsewhere onto the study site, then the growth differences of the tarsus would reflect the habitat quality or conditions of the site where the development took place. It is worrisome, for example, that juveniles were not included in the study even though these would be an example of individuals for which tarsi development location would have been known. Similarly, it is not clarified where the rectrices feather growth took place and this is typically different depending on species and age. In first year birds, the rectrices would have grown in on the breeding grounds. If the captured birds were after-first-year adults, the rectrices might have molted in during different time periods and at different locations depending on the species. If this information is unknown then the only way to know for certain that a feather was grown at the study site would be to pluck the feather of unknown origin and then pluck the regrown feather from the recaptured bird. Thus the feather growth rate would reflect conditions at that specific site and not the location where the original plucked feather was grown.

It is my assumption that this information was taken into account and it is known but simply not reported. For example, it could be that the ten species are known to breed (and thus develop their tarsi) on site. It could be that the species are migrants from another breeding site but they moult rectrices during the time period in which they are captured on site, or they are year round residents and thus perform their moulting cycles on site (and do not move between the forest and shade coffee). I am happy to review the paper again if this is the case. However, if this type of information is unavailable then I am afraid it is impossible to know whether the tarsi and feather measures reflect what the authors assume they reflect.

Our reply: We ring birds for more than three years in this particular study sites and we never come across individual birds those move from one fragment to another. Individual birds were caught first as immature later as adult rewrap in the same sites year after years. From this we conclude most of the birds born and grow on the same sites, and not migrate from one forest to other. We have this experience from our previous study on bird species richness, diversity

and community structure in the same study sites using mist netting procedures where all captured individuals ringed (Buechley et al., 2015).

Buechley ER, Şekercioglu ÇH, Atickem A, Gebremichael G, Ndungu JK, Abdu B, Beyene T, Mekonnen T, Lens L. (2015). Importance of Ethiopian shade coffee farms for forest bird conservation. *Biological Conservation* 188:50–60.

I have made other less crucial comments in-text of the pdf that should also be addressed if revised.

Sincerely,

Sacha Heath, PhD

Ecology Graduate Group

University of California, Davis

Line 27 and 28, I suggest making explicitly clear the two step assumption and prediction here, which then leads to your final indirect assumption (prediction):

1. You *assume* (presumably based on previous work that will be detailed in the text) that higher FA and narrower feather growth bars indicate comparatively lower Habitat Quality, and
2. You *predict* lower habitat quality in shade coffee vs. natural forest (as indicated by higher FA and narrower growth bars).

Our reply: we have now clarified the two assumptions clearly in the updated manuscript version

Line 28, page 2

Line 28 page 2, I think I understand what you are getting at here, but perhaps add a sentence or a few words to clarify. The reader might wonder why it is suggested that these habitats be converted to shade coffee vs. restored to healthy forests. So adding something about degraded forests being converted into shade coffee as compensation for lost coffee production due to forest protection would clarify. Also, is shade coffee considered to be of higher quality than degraded forest?

Our reply: Okay we have now clarified this sentence

Line 28 page 2, Replace with the actual number

Our reply: Okay, we have made the change accordingly

Line 50, insert bird before species

Our reply: we have inserted bird as suggested

Line 51, page 3, in shade coffee farming have been shown to be comparable to natural forests in some cases (8-10), and not in others (several citations - see reviewr comments).

Our reply: we fully understand this, and have tried to clarify the sentence

Line 51, change was to has been

Our reply: we have made the changes accordingly

Line 51, in shade coffee farming have been shown to be comparable to natural forests in some cases (8-10), and not in others (several citations - see reviewr comments).

Our reply: Okay, we have done as suggested [11-14] line 48.

Line 53, I made a few suggestions for how to tone down this statement. There has been evidence presented, as the introduction suggests, that remnant forest and shade coffee are comparable in terms of these bird measures, but I would not go so far to state “no negative effect” because the story is more nuanced. There are several examples where shade coffee measures of bird species richness, etc, were lower than native forest remnants (and this should be noted). For example, see this review and citations therein:

Komar. 2006. Ecology and conservation of birds in coffee plantations: a critical review. *Bird Conservation International* 16:1-23:

“As expected, bird species-richness and diversity in shaded coffee plantations tend to be lower than in nearby forest patches (Terborgh and Weske 1969, Beehler et al. 1987, Thiollay 1995, Wunderle and Latta 1996, Estrada et al. 1997, Greenberg et al. 1997a, Petit et al. 1999), although studies in some landscapes have documented the same or even higher species richness as natural forest (Aguilar- Ortiz 1982, Greenberg et al. 1997b, Shahabuddin 1997, Tejeda-Cruz and Sutherland 2004). Species diversity in shaded plantations is nearly always reported to be considerably higher than in open-sun plantations or other types of monoculture (Beehler et al. 1987, Wunderle and Latta 1996, Estrada et al. 1997, González 1999, Petit et al. 1999, Tejeda-Cruz and Sutherland 2004).”

Our reply: Okay, we have now modified the paragraph to show the importance of shade coffee for biodiversity conservation is not always the case

Line 56 This sentence could be broken into two or three sentences for clarity.

Our reply: Okay, we have broken the sentence in to two.

Line 62, Please provide a few citations of examples.

Our reply: Okay we have included few citations

Line 65, Insert “relative levels of”

Our reply: we have inserted “relative levels of” as suggested

Line 65, be careful with how this is stated. Quality could be considered lower or higher depending on the specific comparisons of these stress measures. For example, it is not stated that body mass, size, etc. is explicitly lower and thus equates with lower quality... stay more general here.

Our reply: Okay. We did as suggested

Line 65, delete lower

Our reply: We did the change as suggested

Line 66 to 70, These are two important points that aren't necessarily connected; please break up the long sentence into two.

Our reply: We did the changes as suggested

Line 71 to 78, No need to go into an exhaustive review of these two measures, but since the entire study is based on FA and feather growth, it would help the reader to have a bit more justification for the validity of the methods. Please more explicitly explain the links between FA and Feather Growth to fundamental life history components. Also, what do you think about the criticisms of ptilochronology? (for example, see citations below). Again, no need for too much detail but please add in a few more clarifying sentences to make the life history links and very briefly address the criticisms of Murphy.

Murphy and King. 1991. Ptilochronology: a critical evaluation of assumptions and utility. *The Auk* 108:695-704

Murphy. 1992. Ptilochronology: accuracy and reliability of the technique. *The Auk* 109:676-680.

Our reply: Okay we have include this debate in the discussion section

Line 87, delte however

Our reply: we have deleted the word “delete”

Line 88, delete therefore

Our reply: we have made the change as suggested

Line 97, replace ,, by “is of lower quality than natural forests”,

Our reply: We did the change as suggested

97, replace ,, by “predicted”

Our reply: we did as suggested

Line 99, Based on next two comments, please rework this section. I can see why nestling growth effects might affect bird health in later years, but that isn't described here.

Our reply: Okay, we have done as suggested lines 101 & 102.

Line 101, so this demonstrates previous effects, not subsequent effects as stated earlier in the sentence.

Our reply: We did as suggested.

Line 102, But stress induced during one feather growth period does not necessarily indicate anything about stress during future molt periods. It only indicates stress levels during the particular growth phase measured.

Our reply: Yes, that is true. In the current version we have addressed this issue lines 102&103.

Line 126 , Do these birds occupy the sites year round? During what time period do they moult their rectrices? Do they grow their rectrices on site or during another season at another site? If measured feathers were grown offsite, then feathers must be plucked and feathers regrown onsite must again be plucked and measured to quantify local conditions. This point is crucial to the entire study so please clarify.

Our reply: The species included in this study are resident bird species and they stay in the sites year round. They grow the rectrices on the sites lines 131&132.

Line 128, delete already

Our reply: we have made the change

Line 138, Birds were captured year round? So breeding, migrant/transient, and overwintering birds were all included or were only birds that developed as nestlings and/or moulted in rectrices on site used for analysis? Crucial point, please clarify.

Our reply: No, birds were captured from 9 December, 2013 to 30 March, 2014 and 19 December, 2014 to 10 June, 2015. Migrant birds/transient, and overwintering birds were not included this analysis. We only analyzed the data of bird that were developed as nestling and/or moulted rectrices on the site.

Line 139, mist-netting procedures.

Our reply: we have made the change accordingly

Line 140, replace controlled by Checked

Our reply: we have made the change accordingly

Line 140, delete

Our reply: we have made the change accordingly

Line 142, why?

Our reply: We, fully admit this is a mistake. And corrected it in the current version, even the immature individual were also included in tarsus analysis.

Line 152, By two different observers?

Our reply: the observer was the same person

Line 154, replace ,, ,, by retrix

Our reply: we did the change as suggested

Line 155, what is the life history of these birds? When were these feathers grown? Where they grown on the study site or were these birds migrants and thus the feather growth took place in a different location? It is impossible to evaluate this study without knowing this.

Our reply: Our study species were resident, stay round in a site where they were captured. Feather growth and moult were takes place in the same sites (in our study sites). These species do not migrate from one forest fragment to the other. Even if we made many years ringing effort, we did not captured individuals that captured in one forest fragment in other.

Line 160, Again. This procedure completely depends on when these feathers were grown. Are these birds year round residents of their various plots and therefore the feather growth reflects the conditions of that habitat? Or, are any of these birds migrants that only use the study sites during a certain time period? Do they occupy the sites when rectrices molt occurs? If the feathers grow in offsite, then the growth bars do not represent the quality of the habitat in question.

Our reply: Bird species of our study are resident and stay year round in sampling sites. Previous work of Buechley et al., 2015 have showed that these bird species have breeding population in this particular study plots. Gebremichael et al., unpublished data also showed that these bird species were breed and moult in the study plots of our study area.

Line 194, perhaps begin this section by stating that you followed the analysis procedures described in citation 53.

Our reply: we have made the changes accordingly

Line 198: is this the random slope for side?

Our reply: yes, the random side is for random slop that estimates FA within individual.

Line 224, because different species were grouped into functional groups, species should probably also be included as a random effect, otherwise a particular species within a group might bias the results.

Our reply: This is true, and in the updated manuscript we re-analysed the data as species comparison rather than the former analysis which is based on guilds of functional guilds

Appendix C**ROYAL SOCIETY
OPEN SCIENCE****Fluctuating asymmetry and feather growth bars as
biomarkers to assess habitat quality of shade coffee
farming for avian diversity conservation**

Journal:	Royal Society Open Science
Manuscript ID	RSOS-190013
Article Type:	Research
Date Submitted by the Author:	02-Jan-2019
Complete List of Authors:	Gebremichael, Gelaye; Jimma University, Biology Department; Ghent University, Department of Biology Alemu, Diress ; Universitetet i Oslo Det Matematisk-naturvitenskapelige Fakultet, Department of Biosciences Bunnefeld, Nils; University of Stirling Zinner, Dietmar ; Germany primate center Atickem, Anagaw; Germany primate center , Cognitive Ethology Laboratory
Subject:	ecology < BIOLOGY
Keywords:	bird community, bird species, Ethiopia, fluctuating asymmetry, ptilochronology
Subject Category:	Biology (whole organism)

1 **Fluctuating asymmetry and feather growth bars as biomarkers to assess habitat quality**
2 **of shade coffee farming for avian diversity conservation**

Gelaye Gebremichael^{1,2}, Diress Tsegaye³, Nils Bunnefeld^{4,5}, Dietmar Zinner⁶, Anagaw
Atickem⁶

¹Terrestrial Ecology Unit (TEREC), Ghent University, K.L. Ledeganckstraat 35, 9000,
Ghent, Belgium

²Jimma Universities, College of Natural Sciences, P.O. Box 378, Jimma, Ethiopia

³University of Oslo, Department of Biosciences, Postboks 1066 Blindern, 0316 Oslo, Norway

⁴Biological and Environmental Sciences, Faculty of Natural Sciences, University of Stirling,
Stirling FK9 4LA, UK

⁵School of Geosciences, University of Edinburgh, Edinburgh, EH9

⁶Cognitive Ethology Laboratory, German Primate Center, Leibniz Institute for Primate
Research Kellnerweg 4, 37077 Göttingen, Germany

**Fluctuating asymmetry and feather growth bars as biomarkers to assess habitat quality**
**of shade coffee farming for avian diversity conservation**

**Abstract**

Shade coffee farming has been promoted as a means for combining sustainable use of forests
and biodiversity conservation. Both in shade coffee and natural forests a similar diversity and
abundance of birds have been found. However, the negative effects on such indicators for
habitat quality may become obvious only after some time, because of a constant influx of
birds from nearby subpopulations in natural forests. Therefore other indicators for habitat
quality of shade coffee forests should possibly be applied. We used two biomarkers,
fluctuating asymmetry (FA) of tarsus length and rectrix mass, and feather growth bars
(average growth bar width), assuming higher FA and narrower feather growth bars in shade
coffee forest than in a natural forest. We measured and compared 320 and 223 individual
birds of five species ~~for tarsus length and rectrix mass, respectively~~ from both forest types.
The extent of FA in both ~~traits~~ (tarsus length and rectrix mass) ~~showed no difference~~ between
the two forest types in any of the five species. Similarly, ~~no difference was found in growth~~
~~bar widths in any of these species between shade coffee and natural forest~~. Therefore, we
conclude that shade coffee farming in Afrotropical rain forests of Ethiopia is compatible with
**avian conservation.**

Keywords: bird community, bird species, Ethiopia, fluctuating asymmetry, ptilochronology

Introduction

Tropical forests host at least two-thirds of the world's terrestrial biodiversity and offer a wide
range of ecosystem services [1]. However, these forests are shrinking at unprecedented rates
[2,3]. Agricultural practices compatible with forest conservation have been proposed as a
potential strategy to slow down, stop or even reverse the current alarming rate of
deforestation [4].

Shade coffee farming, i.e. growing coffee beneath the canopy of shade trees, has
recently gained support as a form of agricultural practice that is, at least partly, more
compatible with biodiversity conservation than other agricultural practices [5-7]. However,
the results of various studies are contradictory. For instance, species richness, abundance and
community composition of birds in shade coffee forests have been shown to be similar to
those in natural forests in some studies [8-10], but not in others [11-14]. But compared to
nearby open-sun plantation, bird species diversity in shade coffee forests is notably higher
[9,12,13]. Therefore, ~~it has been concluded, that~~ shade coffee farming can ~~play~~ an important
~~role when combining~~ biodiversity conservation ~~and~~ agro-forestry [15,16].

Such inference implicitly assumes that populations inhabiting shade coffee forests also
share equal prospects of reproduction or/and long-term survival with conspecifics residing in
pristine forests [17,18]. However, demographic changes in the distribution of sex or age
classes, changes in rates of survival or reproduction, which results from biotic changes, are
possibly long-term effects. These are unlikely to be picked up by community-level studies on
demography and α -diversity alone if surveys are done shortly after natural forests have been
transformed into shade coffee forests. Communities can be affected without any noticeable
change in diversity and abundance due to time lags in responses or immigration from nearby
source populations [19,20]. To overcome problems of diversity and demography abundance
based studies, a growing number of studies applied phenotypic or physiological proxies of

62 fitness (biomarkers), such as body size, body mass, fat-free mass, or the rate or precision of
63 growth. These markers are all thought to be able to indicate stress (e.g. nutritional stress), and
64 thus relative levels of habitat quality [21,22]. Habitat quality, in turn, have effects on
respective populations in the long run [21,22]. Stress-mediated changes should preferably be
detectable before direct components of fitness are compromised (see [23] for a list of criteria
to evaluate biomarkers). Most important, conservation contexts, collecting respective data
should not involve lethal sampling, exhaustive training or expensive equipment.

Fluctuating asymmetry (FA), i.e. small random deviations from left-right symmetry in
bilateral traits [24-27], and feather growth rate, measured as the width of alternating dark and
light growth-bars on bird feathers [ptilochronology, 28-31], fulfilling the above criteria. The
benefits of these markers are based on their ease, low cost, and non-lethality of measurement
and their link to fundamental life-history components that are otherwise more laborious to
measure [32]. When applied in tandem [33,34], they provide an overall, integrated picture of
environmental and nutritional stress effects [21], which in turn can provide information on
habitat quality.

[revised manuscript text omitted]

2.2 Study species

Our study focussed on five bird species byssinian Ground Thrush, Rüppell’s Robin-chat,
Grey-backed Camaroptera, African dusky Flycatcher and Tambourine Dove which are found
in both forest types (Table 1). Based on their primary habitat utilization, the former two
species are considered forest species while the latter three species are considered as non-
forest species [50-56]. Based on their feeding habit; Rüppell’s Robin-chat, Grey-backed
Camaroptera and African dusky Flycatcher are regarded as insectivore, Abyssinian Ground
Thrush as omnivore and Tambourine Dove as granivores [57]. These species are resident
[58], and have breeding populations at our study sites [38].

2.3 Data collection

From December 2013 to June 2015, we captured ~~351~~ birds belonging to five bird species
using mist-netting procedures. We opened the nets 30 minutes before sunrise and kept them
open for six hours. We checked the nets routinely at 30-min intervals to remove birds without

holding them too long. Upon capture, 1) ~~we~~ ringed each individual bird with rings from
National Museum of Kenya, ~~aged, weighted~~ (to the nearest 0.01g using digital balance), 2)
measured tarsus length (to the nearest 0.1 mm with Vernier callipers), and wing length (to the
nearest 0.5 mm using wing ruler), and 3) plucked the second outermost (the fifth) of right and
left rectrices feathers from individual captured birds. We released the birds immediately after
we collected feather samples and took the measurements on the spot.

2.3.1 Measurement of tarsi length and rectrix mass

Fluctuating asymmetry was determined from two traits, tarsi length, and rectrix mass.
Different traits have different susceptibility for FA based on cost of growth of the trait,
defined as the amount of structural components necessary to form a unit of length of a given
character [59] hence we used both traits. ~~Tarsi length~~, distance from the back of inter-tarsal
joint to the point where the toes bending an angle of 90 degree to the tarsus [60], ~~was~~
~~149 measured in the field to the nearest 0.1 mm using Vernier calliper~~  Two independent
measurements of right and left tarsus lengths for each individual (sequence right-left-right-
left or left-right-left-right) were taken. ~~The second outer most right and left rectrices (fully-~~
~~152 grown) rectrices feathers were collected from every bird captured for rectrix mass and growth~~
~~153 bar measurements~~  The feathers were stored in separately labelled dry paper envelopes
and transported to the laboratory, and the rectrices were weighted twice using an analytical
balance to the nearest 0.1 mg. Tarsi with anomalies were not measured, and rectrices with
damaged tips or soiled with faeces were not collected.

49 157 2.3.2 Measurement of growth bar widths

Growth bar widths and total feather lengths were measured by one of us (GG) as follow: (1)
each unwearied feather was pinned on separate, white, polystyrene boards and total feather
length was measured to the nearest 0.01 mm with a digital calliper. (2) Each feather was
marked at a distance of 7/10 from its proximal end, and the proximate and distal ends of 5

consecutive growth bars were marked with ultrafine mounting pins. (3) Each marked board
was scanned (Oce OP1130) and growth bar widths and total feather lengths were
automatically measured with an image-analysis-software (KS400 Zeiss). The entire
procedure was repeated for 66 randomly selected feathers to assess the repeatability of our
measurements.

2.4 Data validation

Possible data outliers were detected using Cook distance (Cookd) (in SAS) to ensure data
quality for all measurements (tarsus length, feather mass, feather length, and average growth
171 bar widths), whereby influential observation values were screened out from the data.

The level of repeatability for the growth bar widths was calculated according to Lessells
and Boag [61], whereby within-group and between group mean squares were obtained from
1-way ANOVAs, and the p-values were obtained from F-tests. Repeatability estimates were
high, both for total feather length (right trait side: $r = 0.999$, $N = 66$, $P = <0.001$; left trait
side: $r = 0.999$, $N = 66$, $P < 0.001$) and for average growth bar widths (right trait side: $r =$
0.992 , $N = 66$, $P < 0.000$; left trait side: $r = 0.983$, $N = 66$, $P < 0.001$).

Since the residuals of the measured traits (unsigned FA values for tarsus length and
rectrix mass) deviated from respective normal distributions (Shapiro-Wilk test; see Table S1),
the data were transformed using Box-Cox transformation to meet normality assumptions
before further analysis.

2.5 Data analyses

2.5.1 Analysis of fluctuating asymmetry

~~Individual signed~~ FA for tarsus length and rectrix mass were quantified as a difference in
measures between right (R) and left (L) (using average values obtained from the two

replicated measurements of each side) [62]. ~~Individual unsigned FA (the magnitude of signed~~
~~FA) values were the absolute values of signed FA.~~

189 Following Van Dongen et al. (1999), the levels of significance of signed FA and its
repeatability in the measured traits were calculated by the restricted maximum likelihood
(REML) estimation of a mixed regression model where we fitted the models to the repeated
measurements of right and left trait sides [63]. We included sides as fixed effects in the
models and measured trait values (tarsi lengths or rectrix masses) as response variables,
intercept and sides as a random effect and individual id as a subject. This procedure allows
the separation of measurement error (ME) from bilateral asymmetry analysis [63]. The
presence of directional asymmetry (DA) was assessed by the F-test of the fixed effects, with
degrees of freedom corrected for statistical dependence by Satterthwaite formulas [64]. We
used random intercepts and slopes (both estimated within individuals) to estimate the
variation in individual trait value and the individual signed FA, respectively, while the
random error variance component gave the measurement error. The significance of FA was
then calculated by performing a likelihood ratio (LR) test comparing two models: the
original, full model and a reduced model without the side (left or right) as a random effect.
Then, intra-class correlation coefficients (ICC) were calculated to evaluate the repeatability
of FA [63].

Absence of anti-symmetry, i.e. a type of asymmetry that occurs in a random direction,
producing a bimodal distribution over the population, was verified by testing the signed FA
for normal distribution.

2.5.2 Analysis of unsigned FA

The linear mixed model was fitted to the data to test whether unsigned FA levels in tarsus
length differed between birds of shade coffee forests and birds of natural forests whereby

unsigned FA was included as a response variable, forest type as a fixed effect and each
sampling location as a random effect.

Model selection was done using Akaike Information Criterion (AIC) for the species with
moderate sample sizes (Rüppell's Robin-chat and Tambourine Dove). For the species with
small sample sizes (Abyssinian Ground Thrush, Grey-backed Camaroptera, and African
Dusky Flycatcher), model selection was done using AICc (the form of AIC corrected for
small sample size).

19 219 2.5.3 Analysis of growth bar width

To test whether average growth bar widths differed between birds of shade coffee forests and
those of natural forests, we added average growth bar widths in the models as a response
variable, forest type and total feather lengths as fixed effects, and sampling location as a
random effect. All statistical analyses were performed with program SAS (Version 9.4., SAS
Institute 2013, Cary, NC, USA).

33 225 **2. Results**

We retrieved measures of tarsus length from 320 birds, and measures of rectrix mass and
average growth bar width from 223 birds belongs to  species (Table 1). For tarsus length,
kurtosis ranges from $K = 0.33$ in Tambourine Dove to $K = 6.82$ in Abyssinian Ground
Thrush, for rectrix mass, it ranges from $K = 3.551$ in Abyssinian Ground Thrush to $K = 24.75$
in Tambourine Dove (Table S1). The signed FA showed leptokurtic distribution in tarsus
length and rectrix mass for Abyssinian Ground Thrush, African dusky Flycatcher and Grey-
backed Camaroptera, whereas Rüppell's Robin-chat and Tambourine Dove showed such
distribution in rectrix mass only ($K > 3.0$) (Shapiro Wilk test; all $P \leq 0.0257$; Table S1), which
indicates heterogeneity in developmental stability between individuals [63,65]. DA was
detected for tarsus length ($P = 0.0005$), but not for rectrix mass ($P = 0.0967$; Table S2).

58 236 3.1 Fluctuating asymmetry

The signed FA variance estimation was highly significant relative to measurement error in
both tarsus length ($\chi^2 = 789.7$, $df = 1$, $P < 0.0001$) and rectrix mass ($\chi^2 = 449.1$, $df = 1$, $P <$
0.0001) (Table S₂). But, the level of within-side measurement error was high compared to
between-side difference in both traits (tarsus: ICC = 22.3 % and rectrix mass: ICC = $2.5 \times 10^{-}$
⁵). Furthermore, the level of unsigned FA both in tarsus length and rectrix mass did not differ
between natural forests and shade coffee forests for all species, Abyssinian Ground Thrush,
African dusky Flycatcher, Grey-backed Camaroptera, Tambourine Dove and Rüppell's
Robin-chat (all: $P \geq 0.0521$; Table 2 & 3). In spite of this, Abyssinian Ground Thrush,
African dusky Flycatcher, and Tambourine Dove show higher FA both in tarsus length and
rectrix mass in shade coffee than their conspecifics in natural forest (Fig. 2 a & b). On the
other hand, Rüppell's Robin-chat showed higher FA only in tarsus length in shade coffee
forests.

3.2 Growth bar width

Average growth bar width of the five bird species did not differ between shade coffee and
natural forests (all species: $P \geq 0.1409$; Table 4). Furthermore, mean rectrix length did not
show variation between shade coffee and natural forests (all species: $P \geq 0.1587$; Table S₃).
Despite this, Tambourine Dove showed wider average growth bar in the natural forests,
whereas Grey-backed Camaroptera showed narrower growth bar in the natural forest (Fig.3).

3. Discussion

Several studies based on species richness, abundance and community composition revealed
that shade coffee farming has an important role in biodiversity conservation [8-10,15,16].
Yet, the intensification of coffee management urges a close monitoring of habitat quality of
shade coffee farms with methods which can detect changes of habitat quality before it is
degraded to the extent that it affects avian community demography [66-68]. Here, we applied
biomarkers (FA in tarsus length and rectrix mass, and growth bar width and total feather

length) to re-assess the quality of the Ethiopian tropical rain forest which was reported to
support rich avian biodiversity comparable to the natural forest ~~based on demographic studies~~
[38].

FA in both traits (tarsus length and rectrix mass) did not show a significant difference
in birds of shade coffee forests and natural forest ~~including in forest species~~. Similarly, the
analysis of feather growth bar widths revealed no significant difference between birds of the
two forest types. Our results are in contrast to Buechley et al. [38] who reported that forest
specialist guilds and understory-insectivore were **more negatively** affected than birds of other
guilds.

~~Currently, in our study area, the negative effects of shade coffee farming is not evident~~
~~in any of the bird species~~. Yet, the long-term impact of the on-going shade coffee plantation
modification and management may be closely monitored. Shade coffee farm management
which is selective removal of trees species may decrease habitat quality of shade coffee
plantation for avian biodiversity [5,69]. In many coffee growing countries across the globe,
shade coffee management practices are also increasingly intensified, which is believed to
reduce plant diversity and canopy cover [68,70-72], and hence, also reducing the quality or
quantity of feeding, nesting or hiding resources for animal populations. Such tendencies urge
a close monitoring of habitat quality of shade coffee forests with biomarkers which can detect
changes of habitat quality before it is degraded to the extent that it affects avian diversity and
community demography [18,66,73,74]. Globally, about 21% of bird species are currently
extinction-prone and 6.5% are functionally extinct [75] which urges enhanced conservation
and monitoring effort.

Future studies also may combine FA analysis in tarsus length and rectrix mass as
implemented in this study to provide a more comprehensive result. Only a few studies
combined FA and feather growth bar width analysis which showed the two markers showed

consistent result or only one of the markers show the effect of fragmentation [25,33,76]. We
recommend combining the two biomarkers to assess the effects of habitat fragmentation. FA
is a sensitive measure of habitat quality that needs a rigorous test for the validity of the data
used for the analysis and could be very useful as early warning clue for negative effects of
habitat change before any demographic change can be observed [66].

15 292 **4. Conclusion**

Our results are consistent with results of many of the demographic studies that shade coffee
farming in the Ethiopian highland can be compatible with avian diversity conservation. No
early sign of negative effect was observed from biomarker studies. Buechley et al. [38]
showed that shade coffee farming has an important contribution for the avian biodiversity
conservation though forest specialist birds are disproportionately affected in shade coffee
farms. While the current biomarker-based study did not reveal any single of negative effect, it
is important to continue monitoring the Ethiopian shade coffee for the avian biodiversity
conservation as the shade coffee plantation is still under pressure of modification and subject
for the increasing level of fragmentation. We also recommend combining FA and feather
growth bars as biomarkers to provide more comprehensive results in studies of habitat quality
for avian biodiversity conservation than using only one of them.

**Ethics:** This project was carried out in accordance with the ethical standards for research
from Jima University, Ethiopia and Ethiopian Wildlife Conservation Authority (EWCA), and
the project was endorsed by EWCA.

Data accessibility: Data available at the Dryad Digital Repository

<https://datadryad.org/review?doi=doi:10.5061/dryad.cb0d8ft>

Authors' contributions: GG did the fieldwork and collected the data. GG and AA drafted a
first version of the manuscript which was improved by DZ. GG, DT, and NB did the
statistical analyses. All authors reviewed the manuscript and gave their final approval for
publication.

Competing interests: We declare we have no competing interests.

Funding: This research is funded by the VLIR IUC-JU project (VLIR-UOS Institutional
University Cooperation programme between Jimma University and various Flemish
universities under the umbrella of the Flemish Interuniversity Council).

Acknowledgement: We are grateful to Bahir and Reshad Abafita and Jewad Abazinab for
assisting us during our fieldwork and all VLIR IUC-JU divers. We thank Luc Lens for
important comments on improving the proposal and designing the field work. We thank
Kasahun Eba at VLIR IUC-JU programme office at Jimma University for his support
throughout our fieldwork period. We thank Alexander von Humboldt Foundation for the
support given
to Anagaw Atickem.

**References**

1. Gardner T.A., Barlow J., Chazdon R., Ewers R. M., Harvey C. A., Peres C.A., Sodhi N. S.,
2009. Prospects for tropical forest biodiversity in a human-modified world. *Ecol. Lett.*
**12**, 561–582. (doi:10.1111/j.1461-0248.2009.01294.x)
2. FAO, 2010. Global Forest Resources Assessment (FRA) 2010. Key findings Food and
Agriculture Organization of the United Nations, Rome.
3. Lawrence, D., Vandecar, K., 2015. Effects of tropical deforestation on climate and
agriculture. *Nat. Clim. Change* **5**, 27–36. (doi:10.1038/NCLIMATE2430)
4. Ranganathana, J., Daniels, R.J.R., Chandranc, M.D.S., Ehrlich, P.R., Dailya, G.C., 2008.
Sustaining biodiversity in ancient tropical countryside. *Proc. Natl. Acad. Sci. USA* **105**,
17852–17854. (doi:10.1073/pnas.0808874105)
5. Rappole, J.H., King, D.I., Vega Rivera, J.H., 2003. Coffee and conservation. *Conserv.*
*Biol.* **17**, 334–336. (doi:10.1046/j.1523-1739.2003.01548.x)
6. Caudill S.A., DeClerck F.J.A., Husband T.P., (2015). Connecting sustainable agriculture
and wildlife conservation: Does shade coffee provide habitat for mammals? *Agric.*
*Ecosyst. Environ.* **199**, 85–93. (doi:10.1016/j.agee.2014.08.023)

7. Kinasih, I., Cahyanto, T., Widiana, A., Kurnia, D. N. I., Julita, U., Putra, R.E., 2016. Soil
invertebrate diversity in coffee-pine agroforestry system at Sumedang, West Java.
*Biodiversitas* **17**, 473–478. (doi:10.13057/biodiv/d170211)
8. Perfecto, I., Mas, A., Dietsch, T., Vandermeer, J., 2003. Conservation of biodiversity in
coffee agroecosystems: a tri-taxa comparison in southern Mexico. *Biodivers. conserv* **12**,
1239–1252. (doi:10.1023/A:1023039921916)
9. Tejeda-Cruz, C., Sutherland, W.J., 2004. Bird responses to shade coffee production. *Anim.*
*Conserv.* **7**, 169–179. (doi:10.1017/S1367943004001258)
10. López-Gómez, A.M., Williams-Linera, G., Manson, R.H., 2008. Tree species diversity
and vegetation structure in shade coffee farms in Veracruz, Mexico. *Agric. Ecosyst.*
*Environ.* **124**, 160–172. (doi:10.1016/j.agee.2007.09.008)
11. Terborgh, J., Weske, J. S., 1969. Colonization of secondary habitats by Peruvian birds.
*Ecology* **50**, 765-782.
12. Petit, L. J., Petit, D. R., Christian, D. G., Powell, H. D., 1999. Bird communities of
natural and modified habitats in Panama. *Ecography* **22**, 292-304.
13. Wunderle Jr, J. M., Latta, S. C., 1996. Avian abundance in sun and shade coffee
plantations and remnant pine forest in the Cordillera Central, Dominican Republic.
*Ornit. Neotrop.* **7**, 19-34.
14. Estrada, A., Coates-Estrada, R., Meritt, D.A., 1997. Anthropogenic landscape changes
and avian diversity at Los Tuxlas Mexico. *Biodiv & Cons.* **6**, 19-43.
15. Leyequien, E., de Boer, W.F., Toledo, V.M., 2010. Bird community composition in a
shaded coffee agro-ecological matrix in Puebla, Mexico: The effects of landscape
heterogeneity at multiple spatial scales. *Biotropica* **42**, 236–245. (doi:10.1111/j.1744-
7429.2009.00553.x)

16. Tadesse, G., Zavaleta, E., Shennan, C., 2014. Coffee landscapes as refugia for native
woody biodiversity as forest loss continues in southwest Ethiopia. *Biol. Conserv.* **169**,
384–391. (doi:10.1016/j.biocon.2013.11.034)
17. Stearns, S. C., 1992. *The evolution of life histories*. Oxford University Press, Oxford, UK.
18. Johnson, M.D., 2007. Measuring habitat quality: a review. *Condor* **109**, 489–504. (doi:
10.1650/8347.1)
19. Nagelkerke K.C.J., Verboom J., van den Bosch F., van de Wolfshaar K., 2002. Time lags
in metapopulation responses to landscape change. In *Concepts and applications of land*
*scape ecology in biological conservation* (ed. KJ Gutzwiller), Springer, pp. 330–354.
20. Uezu, A., Metzger, J.P., 2016. Time-lag in responses of birds to Atlantic Forest
fragmentation: restoration opportunity and urgency. *PLoS ONE* **11**, e0147909.
(doi:10.1371/journal.pone.0147909)
21. Lens, L., Eggermont, H., 2008. Fluctuating asymmetry as a putative marker of human-
induced stress in avian conservation. *Bird Conserv. Int.* **18**, 125–143.
(doi:10.1017/S0959270908000336)
22. Huggett, R.J., Kimerle, R.A., Mehrle, Jr. PM., Bergman, H.L., 1992. *Biomarkers:*
*biochemical, physiological and histological markers of anthropogenic stress*. Lewis.
23. Leung, B., Knopper, L., Mineau, P., 2001. *A critical assessment of the utility of*
*fluctuating asymmetry as a biomarker of anthropogenic stress*. In *Developmental*
*Instability: Causes and Consequences* (ed. M Polak), pp. 415–426. Oxford University
Press, Oxford, UK,
24. Van Valen, L., 1962. A study of fluctuating asymmetry. *Evolution* **16**, 125–142.
(doi:10.1111/j.1558-5646.1962.tb03206.x)
25. Møller, A.P., Manning, J.T., 2003. Growth and developmental instability. *Vet. J.* **166**, 19–
27. (doi:10.1016/S1090-0233(02)00262-9)

26. De Coster, G., van Dongen, S., Malaki, P., Muchane, M., Alcántara-Exposito, A.,
Matheve, H., Lens, L., 2013. Fluctuating asymmetry and environmental stress:
understanding the role of trait history. *PLoS ONE* **8**, e57966.
(doi:10.1371/journal.pone.0057966)
27. Costa, R. N., Solé, M., Nomura, F., 2017. Agro pastoral activities increase fluctuating
asymmetry in tadpoles of two neotropical anuran species. *Austral. Ecol.* **42**, 801-809.
(doi:10.1111/aec.12502)
28. Grubb, T.C., 1989. Ptilochronology: Feather growth bars as indicators of nutritional
status. *Auk* **106**, 314–320.
29. Grubb, T.C. Jr., Cimprich, D.A., 1990. Supplementary food improves the nutritional
condition of wintering woodland birds: evidence from ptilochronology. *Ornis Scand.*
**21**, 277–281. (doi:10.2307/3676392)
30. Grubb, T.C. Jr., Yosef, R., 1994. Habitat specific nutritional condition in loggerhead
shrikes (*Lanius ludovicianus*): evidence from ptilochronology. *Auk* **111**, 756–759.
31. Grubb, T.C., Jr., 2006. *Ptilochronology: feather time and the biology of birds*. Oxford
University Press, Oxford, UK.
32. Møller, A.P., 1997. Developmental stability and fitness: a review. *Am. Nat.* **149**, 916–
932.
33. Carbonell, R., Tellería, J.L., 1999. Feather traits and ptilochronology as indicators of
stress in Iberian blackcaps *Sylvia atricapilla*. *Bird Study* **46**, 243–248.
(doi:10.1080/00063659909461136)
34. Eeva, T., Tanhuanpää, S., Råbergh, C., Airaksinen, S., Nikinmaa, M., Lehikoinen E.,
(2000). Biomarkers and fluctuating asymmetry as indicators of pollution-induced stress
in two hole-nesting passerines. *Funct. Ecol.* **14**, 235–224. (doi:10.1046/j.1365-
2435.2000.00406.x)

35. Mittermeier, R.A., Gil, P.R., Hoffman, M., Pilgrim, J., Brooks, T., Mittermeier, C.G.,
Lamoreux, J., Da Fonseca, G.A.B., 2004. *Hotspots Revisited: Earth's Biologically*
*Richest and Most Endangered Terrestrial Ecoregions*. CEMEX, Conservation
International, and Agrupación Sierra Madre, Monterrey, Mexico
36. Kufa, T., Burkhardt, J., 2011. Plant composition and growth of wild *Coffea arabica*:
Implications for management and conservation of natural forest resources. *Int. J.*
*Biodivers. Conserv.* **3**, 131–141.
37. Reichhuber, A., Requate, T., 2012. Alternative use systems for the remaining Ethiopian
cloud forest and the role of Arabica coffee - A cost-benefit analysis. *Ecol. Econ.* **75**,
102–113. (doi:10.1016/j.ecolecon.2012.01.006)
38. Buechley, E.R., Şekercioglu, Ç.H., Atickem, A., Gebremichael, G., Ndungu, J.K., Abdu,
B., Beyene, T., Mekonnen, T., Lens, L., 2015. Importance of Ethiopian shade coffee
farms for forest bird conservation. *Biol. Conserv.* **188**, 50–60.
(doi:10.1016/j.biocon.2015.01.011)
39. Braziotis, S., Liordos, V., Bakaloudis, D.E., Goutner, V., Papakosta, M.A., Vlachos,
C.G., 2017. Patterns of postnatal growth in a small falcon, the lesser kestrel *Falco*
*naumanni* (Fleischer, 1818) (Aves: Falconidae). *Eur. Zool. J.* **84**, 277–285.
(doi:10.1080/24750263.2017.1329359)
40. Tayefeh, F.H., Amini, H., Khaleghizadeh, A., 2016. *Chick growth patterns of three*
*sympatric tern species on the Persian Gulf Islands*. Bird Numbers 2016-Birds in a
changing World Conference, Halle, Germany.
41. Møller, A. P. 1996. Parasitism and developmental instability of hosts: a review. *Oikos*,
**77**, 189-196. (doi:10.2307/3546057)

42. Møller, A. P. 1996. Sexual selection, viability selection, and developmental stability in
the domestic fly *Musca domestica*. *Evol.* **50**, 746-752. (doi:10.1111/j.1558-
5646.1996.tb03884.x)
43. Møller, A. P., Swaddle, J. P. 1997. *Asymmetry, developmental stability and evolution*.
Oxford University Press, UK.
44. Manning, J. T. 1995. Fluctuating asymmetry and body weight in men and women:
implications for sexual selection. *Ethol. Sociobiol.* **16**, 145-153. (doi:10.1016/0162-
3095(94)00074-H)
45. Møller, A. P. 1997. Developmental stability and fitness: a review. *Am.Nat.* **149**, 916-932.
(doi:10.1086/286030)
46. Hundera, K., Aerts, R., Fontaine, A., Van Mechelen, M., Gijbels, P., Honnay O., Muys,
B., 2013. Effects of coffee management intensity on composition, structure, and
regeneration status of Ethiopian moist evergreen Afromontane forests. *Environ. Manag.*
**51**, 801–809. (doi:10.1007/s00267-012-9976-5)
47. Gebrehiwot, K., Hundera, K., 2014. Species composition, plant community structure and
natural regeneration status of Belete moist evergreen montane forest, Oromia Regional
State, Southwestern Ethiopia. *SINET* **6**, 97–101.
48. Demissew, S., Cribb, P., Rasmussen, F., 2004. *Field guide to Ethiopian orchids*. Royal
Botanic Gardens, Kew, UK.
49. De Beenhouwer, M., Aerts, R., Honnay, O., 2013. A global meta-analysis of the
biodiversity and ecosystem service benefits of coffee and cacao agroforestry. *Agric.*
*Ecosyst. Environ.* **175**, 1-7. (doi:10.1016/j.agee.2013.05.003)
50. Brown, L. H., Urban, E. K., Newman, K., 1982. *The birds of Africa*, Volume I, pp. 521.
Academic Press, London

51. Urban, E. K., Fry, C. H., Keith, S., 1986. *The birds of Africa*, Volume II, pp. 552.
Academic Press, London.
52. Fry, C. H., Keith, S. Urban, E. K., 1988. *The birds of Africa*, Volume III, pp. 611.
Academic Press, London.
53. Keith, S., Urban, E. K., Fry, C. H., 1992. *The birds of Africa*, Volume IV, pp. 632.
Academic Press, London. London.
54. Fry, C. H., Keith, S., Urban, E. K., 1997. *The birds of Africa*, Volume V, pp. 672.
Academic Press, London.
55. Fry, C. H., Keith, S., Urban, E. K., 2000. *The birds of Africa*, Volume VI, pp. 704.
Academic press.
56. Fry, C. H., Keith, S., 2004. *The birds of Africa*, Volume VII, pp. 666. Christopher Helm,
London.
57. Wilman, H., Belmaker, J., Simpson, J., de la Rosa, C., Rivadeneira, M.M., Jetz, W.,
2014. EltonTraits 1.0: Species-level foraging attributes of the world's birds and
mammals. *Ecology* **95**, 2027. (doi:10.1890/13-1917.1)
58.. Ash, C. P., Atkins, J. D. (2009). Birds of Ethiopia and Eritrea: an atlas of distribution.
A&C Black.
59. Aparicio, J.M., Bonal, R., 2002. Why do some traits show higher fluctuating asymmetry
than others? A test of hypotheses with tail feathers of birds. *Heredity* **89**, 139–144.
(doi:10.1038/sj.hdy.6800118)
60. Svensson, L., 1992. *Identification Guide of European Passerines*. 4th ed. Svensson,
Stockholm.
61. Lessells, C.M., Boag, P.T., 1987. Unrepeatable repeatabilities: a common mistake. *Auk*
**104**, 116 –121. (doi:10.2307/4087240)

62. Palmer, A.R., Strobeck, C., 2003. Fluctuating Asymmetry analyses revisited. In
*Developmental Instability: Causes and Consequences* (ed. M Polak), pp. 279–319.
Oxford University Press, Oxford.
63. Van Dongen, S., Molenberghs, G., Matthysen, E., 1999. The statistical analysis of
fluctuating asymmetry: REML estimation of a mixed regression model. *J. Evol. Biol.*
**12**, 94–102. (doi:10.1046/j.1420-9101.1999.00012.x)
64. Verbeke, G., Molenberghs, G., 2000. *Linear Mixed Models for Longitudinal Data*.
Springer, New York.
65. Palmer, A.R., Strobeck, C., 1992. Fluctuating asymmetry as a measure of developmental
stability: implications of non-normal distributions and power of statistical tests. *Acta*
*Zool. Fenn.* **191**, 57–72.
66. Lens, L., van Dongen, S., Matthysen, E., 2002. Fluctuating asymmetry as an early
warning system in the critically endangered Taita thrush. *Conserv. Biol.* **16**, 479–487.
(doi:10.1046/j.1523-1739.2002.00516.x)
67. Niemi, G. J., McDonald, M. E., 2004. Application of ecological indicators. *Annu. Rev.*
*Ecol. Evol. Syst.* **35**, 89–111. (doi: 10.1 146/annurev.ecolsys.35.112202.30000005)
68. Aerts R., Berecha G., Honnay, O., 2015. Protecting coffee from intensification. *Science*
**347**, 139. (doi:10.1126/science.347.6218.139-b)
69. Schmitt, C.B., Senbeta, F., Denich, M., Preisinger, H., Boehmer, H.J., 2010. Wild coffee
management and plant diversity in the montane rainforest of south-western Ethiopia.
*Afr. J. Ecol.* **48**, 78–86. (doi:10.1111/j.1365-2028.2009.01084.x)
70. Armbrrecht, I., 2003. Habitat changes in Colombian coffee farms under increasing
management intensification. *Endang. Spec. Update* **20**, 4–5.
71. Aerts, R., Hundera, K., Berecha, G., Gijbels, P., Baeten, M., Van Mechelen, M., Hermy,
509 M., Muys, B., Honnay, O., 2011. Semi-forest coffee cultivation and the conservation of

510 Ethiopian Afromontane rainforest fragments. *For. Ecol. Manag.* **261**, 1034–1041.
(doi:10.1016/j.foreco.2010.12.025)
72. Noponen, M.R.A., Hagggar, J.P., Edward-Jones, G., Healey, J.R., 2013. Intensification of
coffee systems can increase the effectiveness of REDD mechanisms. *Agric. Syst.* **119**,
1–9. (doi:10.1016/j.agsy.2013.03.006)
73. Beasley, D.A.E., Bonisoli-Alquati, A., Mousseau, T.A., 2013. The use of fluctuating
asymmetry as a measure of environmentally induced developmental instability: A meta-
analysis. *Ecol. Indic.* **30**, 218–226. (doi:10.1016/j.ecolind.2013.02.024)
74. Helle, S., Huhta, E., Suorsa, P., Hakkarainen, H., 2011. Fluctuating asymmetry as a
biomarker of habitat fragmentation in an area-sensitive passerine, the Eurasian
treecreeper (*Certhia familiaris*). *Ecol. Indic.* **11**, 861–867.
(doi:10.1016/j.ecolind.2010.11.004)
75. Şekercioğlu C.H., Daily G.C. and Ehrlich P.R. 2004. Ecosystem consequences of bird
declines. *Proc. Natl. Acad. Sci. USA* **101**, 18042–18047.
(doi:10.1073/pnas.0408049101)
76. Polo, V., Carrascal, L.M., 1999. Ptilochronology and fluctuating asymmetry in tail and
wing feathers in coal tits *Parus ater*. *Ardeola* **46**, 195–204.
77. Gebremichael, G., Tsegaye D., Bunnefeld N., Zinner D., Atickem A., 2018. Data from:
Fluctuating asymmetry and feather growth rate as biomarkers to assess habitat quality
of shade coffee farming for avian biodiversity conservation. Dryad Digital Repository.
(doi:10.5061/dryad.cb0d8ft)

Figure 1. Geographic positions of study sites in natural forests (Muje, Afalo, Ababa Buna, and Qacho) and in the shade coffee forests (Yebu, Garuke, Fetche) in the highlands of western Ethiopia.

364x258mm (96 x 96 DPI)

Figure 2. Unsigned FA (mean \pm SE) in five bird species (species: ADF = African dusk Flycatcher; AGT = Abyssinian Ground Thrush; GBC = Grey-backed Camaroptera, RRC = Rüppell's Robin-chat; TD = Tambourine Dove).

235x118mm (96 x 96 DPI)

Figure 3. Average growth bar width (mean \pm SE) in five bird species (species: ADF = African dusk Flycatcher; AGT = Abyssinian Ground Thrush; GBC = Grey-backed Camaroptera, RRC = Rüppell's Robin-chat; TD = Tambourine Dove).

130x104mm (96 x 96 DPI)

Table 1. The number of individual birds assessed for growth bar widths (rectrices masses) and tarsus lengths per species and forest type (natural forest [NF], shade coffee forest [SC]).

species	Common name	growth bar width		tarsus length			
		SC	NF	total	SC	NF	total
Muscicapa adusta	African dusky Flycatcher	10	10	20	21	22	43
Camaroptera brevicaudata	Grey-backed Camaroptera	8	7	15	16	12	28
Cossypha semirufa	Rüppell's Robin-chat	66	30	96	85	40	125
Turtur tympanistria	Tambourine Dove	37	40	77	47	50	97
Zoothera piaggiae	Abyssinian Ground Thrush	5	10	15	13	13	26
Total		123	97	220	183	137	320

Table 2. Unsigned FA estimates in tarsus length for five bird species in two habitat types (natural forest and shade coffee forest). Linear mixed models were fitted to the data where unsigned FA were included in the models as a response variable, forest types as fixed effect and each sampling location as random effect for Abyssinian Ground Thrush (AGT), African dusky Flycatcher (ADF), Grey-backed Camaroptera (GBC), Rüppell's Robin-chat (RRC) and Tambourine Dove (TD).

species	Effect	Estimate	SE	df	t	p
AGT	Intercept	-2.566	0.2491	23	-10.3	<0.0001
	Forest type (natural)	-0.5699	0.3454	23	-1.65	0.1126
ADF	Intercept	0.4219	0.01029	17	41.0	<0.0001
	Forest type (natural)	-0.0210	0.021	40	-1.67	0.1031
GBC	Intercept	0.7796	0.149	28	5.24	<0.0001
	Forest type (natural)	-0.0602	0.227	28	-0.27	0.7926
RRC	Intercept	0.0195	0.004	10	5	<0.0034
	Forest type (natural)	-0.0091	0.0085	11	-1.08	0.3018
TD	Intercept	3.3257	0.3577	86	9.30	<0.0001
	Forest type (natural)	-0.3899	0.4948	86	-0.81	0.4225

Table 3. Unsigned FA estimates in of rectrix weights for five bird species in two habitat types (natural forest and shade coffee forest). Linear mixed models were fitted to the data where unsigned FA were included in the models as a response variable, forest types as fixed effect and each sampling location as random effect for Abyssinian Ground Thrush (AGT), African dusky Flycatcher (ADF), Grey-backed Camaroptera (GBC), Rüppell's Robin-chat (RRC) and Tambourine Dove (TD).

species	Effect	Estimate	SE	df	t	p
AGT	Intercept	1.51×10^{-5}	0.00037	15	4.12	0.0009
	Forest type (natural)	-9×10^{-4}	0.00045	15	-2.11	0.0521
ADF	Intercept	9.68×10^{-4}	0.000568	20	1.7	0.0391
	Forest type (natural)	-11.00066	0.000733	20	-0.89	0.3816
GBC	Intercept	1.35×10^{-6}	0	13	-0.86	<0.0001
	Forest type (natural)	1.8×10^{-5}	1.4×10^{-6}	13	-0.86	0.4038
RRC	Intercept	7.9×10^{-5}	2.3×10^{-5}	94	3.5	0.0007
	Forest type (natural)	1.8×10^{-5}	2.7×10^{-5}	94	0.67	0.5021
TD	Intercept	1.4×10^{-2}	0.004	66	3.69	0.005
	Forest type (natural)	-6.5×10^{-3}	0.005	66	-1.22	0.223

Table 4. Estimates of average growth bar width from mixed linear models, where average growth bar widths included as a response variable, forest types as fixed effect, whereby feather length included as fixed covariate and each sampling location as random effect for Abyssinian Ground Thrush (AGT), African dusky Flycatcher (ADF), Grey-backed Camaroptera (GBC), Rüppell's Robin-chat (RRC) and Tambourine Dove (TD). Natural=Natural forest and TFL=mean of the second out most right and left rectrices.

species	Effect	Estimate	SE	df	t	p
AGT	Intercept	-7.012	7.9878	13	0.880	0.3959
	Forest type (natural)	1.456	0.9286	13	1.57	0.1409
	TFL	0.267	0.093	13	2.88	0.0128
ADF	Intercept	6.029	7.428	16	0.81	0.429
	Forest type (natural)	-0.287	0.5471	16	-0.53	0.429
	TFL	-0.287	0.547	16	0.77	0.451
GBC	Intercept	-4.687	6.44	8.42	-0.73	0.486
	Forest type (natural)	-0.83	1.12	6.1	-0.74	0.486
	TFL	0.38	0.105	8	2.3	0.0496
RRC	Intercept	6.70	4.9647	91	2.36	0.0206
	Forest type (natural)	-5.098	4.9647	88	-1.1	0.276
	TFL	0.098	0.0443	91	2.22	0.0292
	Natural*TFL	0.088	0.0776	88	2.0	0.2574
TD	Intercept	-0.316	0.138	66	1.87	0.979
	Forest type (natural)	0.984	5.038	69	0.07	0.9480
	TFL	0.2582	0.138	65	1.87	0.066
	Natural*TFL	-0.0163	0.175	69	-0.09	0.92

Dear Sirs,

Thank you for your review and for letting us resubmit. We found the comments helpful and
constructive.

We incorporate comments and suggestions of the reviewers in the new version of manuscript.
We also did additional editorial work on top of the reviewers comment. The main concern of
the reviewer was the analysis of fluctuating symmetry of the avian community on the bases
of functional guilds, diet and forest specialization. We now updated the analysis of
fluctuating asymmetry on the bases of species residing in the two habitat types as suggested
by the reviewers.

Please find more detail of the changes we have made,

Sincerely,

Gelaye Gebremichae

Reviewer(s)' Comments to Author:
Reviewer: 1

**Reviewers Comments to Author:**

Reviewer: 1

Comments to the Author(s)

Overview:

This is a very interesting concept, applying morphometrics and fluctuating asymmetry in a
fairly novel fashion to study potential impacts of environmental quality of shade coffee and
natural forest on bird communities in SW Ethiopia. The study is novel and has the potential
to make a significant and important contribution to biodiversity conservation in Ethiopia, as
well as to encourage similar studies worldwide to evaluate early indicators of environmental
stresses on populations. On this level, I think it is suitable for publication in Royal Society
Open Science.

However, I also have some significant methodological concerns with the manuscript that
I hope can be addressed prior to publication. My biggest concern is regarding sample size and
guild classifications, as best outlined in Table 1. Is it really suitable to compare fluctuating
asymmetry and feather growth bar width of different species, as lumped into very general
guilds?

**Our reply:** Our updated version consider only species comparison between natural forest and
shade coffee habitat instead of the comparing avian community based on functional guilds.

The difference in samples sizes of species between shade coffee and natural forest for growth
6 bar width measurements seems particularly great. For example the top 4 species were almost
7 exclusively sampled in shade coffee. Thus how can you compare differences between
8 habitats??? I'm not sure that lumping them into guilds addresses this underlying issue in the
9 dataset. If you have a strong argument for why this is valid to compare, I think it should be
very prominently laid out in the methods and discussion. As is, there is really almost no
discussion of such lack of uneven sampling and how that might impact results. Otherwise,
perhaps the analysis should be limited to species that you have a reasonable sample size in
both habitat types...

**Our reply:** Yes, as we mentioned above, we have now updated the analysis based on species
comparison

I also have a bit of a concern with the guild classification for obligate insectivore vs
opportunistic insectivore, which is based on a seemingly arbitrary 80% insect-diet cut off. Is
there no way to make a more meaningful guild distinction, perhaps based on the
classifications in Buechley et al 2015 (i.e. insectivore / granivore / frugivore).

**Our reply:** This problem is also solved as we change the analysis to species level in the
updated manuscript version.

A couple of other fairly major comments are as follows:

• English was good throughout, up until the discussion, which could use some detailed
work simplifying and correcting grammatical errors.

**Our reply:** in our updated version, we have improved the language and fix grammatical
errors.

• Discussion has several bold claims about coffee forests in Ethiopia that I strongly
suggest toning back. This is an interesting study, but I don't think it is definitive enough to
say things like “the negative effects of converting natural forest into shade coffee forests on
bird communities are not dramatic [line 294]”

**Our reply:** we agreed and in the new version of our manuscript, we have modified this
particular sentence other sentence which appeared to be broad and debatable.

Line 71: “random” – if they are random, then how can they be an index of anything?

**Our reply:** FA is originally defined as “random deviations from left-right symmetry in
bilateral traits [24-27, reference in the main manuscript]”. To our understanding, from large
number of the sample size of the given traits, a random sample of the traits potentially can
reveal the deviation, either as length or weight measurements.

Line 78: What about using other bio-markers like mass?

**Our reply:** That is also a possibility and here we used a specific trait, rectrix mass

Lines 81-82: “the origin and reservoir” – i take issue with the wording here, as arabica also
originated and is still found in SW Arabia

**Our reply:** To our knowledge, the origin of the coffee arabica is Ethiopia and it crossed the
red sea to Yeman and other Arabian countries around the 7th century.

Line 94: tarsi length can be a hard measurement to standardize – perhaps address how this
was done in methods

**Our reply:** we have now provided in detail of how we measured the Tarsi length in the
method section on page 7, lines 147 and 148.

Line 95: Mass of the retrace? This seems an odd measurement!

**Our reply:** Okay we now replaced it with rectrices weight.

Line 97: replace assumed with hypothesized

**Our reply:** We have now made the change

Line 118: comma spacing

**Our reply:** we have now made the change

Line 123: has this directly impacted the study sites? If so how?

**Our reply:** yes, much of the natural forest is now modified to shade coffee. In the new
version of the manuscript, we have tried to explain this further

Line 134: this 80% cutoff seems arbitrary. Could you not find a more well-defined guild
separation, perhaps by following the definitions used in Buechley et al 2015?

**Our reply:** We have now excluded the analysis that based on functional guilds

Line 143: clarify where the rings are from (i.e. museum of nat hist)

**Our reply:** ring were obtained from National Museum of Kenya, and we now have
mentioned this in the manuscript on page 7, line 138.

Line 144: is this the only reason?

**Our reply:** for this particular research question, yes.

Table 1: I'm concerned about unbalanced sampling of the study species between habitat
types. Is it valid to compare fluctuating asymmetry / feather growth across species?

**Our reply:** we fully acknowledge this concern, and we now change our analysis to a species
comparison in the two habitat forms rather than guilds.

Lines 154 – 158: I'm a bit skeptical about this method. Retrix feathers are often soiled by
feces. Were the feathers cleaned prior to weighing? If so, how? Retrix feathers also undergo
major wear. It is not clear to me that tropical birds would necessarily have the same moult
pattern, thus different individuals may be under different levels of feather wear at any given
time, which could impact feather mass. I strongly suggest adding more detail of methods and
citations validating these indices here.

**Our reply:** we caught 488 individual birds of the ten species included in the previous
analysis. But our analysis was only fixed to feathers from individuals birds with
undamaged/unworn and clean feathers (this is the reason why our sample size was small
for some species). Any feather those soiled with feces/ worn were not included in the
analysis page 7, lines 155 and 156.

Line 162: “unworn” – what does this mean??

**Our reply:** it is to mean damaged/worn out

Line 194: I like that repeatability of the measurements was included in the model

**Our reply:** Thanks. This is one of the most important precaution in ptilochronology studies

Line 231: spell out acronyms (e.g. FF, DA, etc) at start of new section

**Our reply:** we have done as suggested

Lines 241 – 245: was this caused by scale accuracy? If the scale is accurate, it seems this
would be the easier measurement to replicate. Can you explain this result more, perhaps in
Discussion?

**Our reply:** As we change the analysis from avian community comparison based on
functional guilds to species, some of the results including this section also changed.

Lines 252-254: show p value for this comment

**Our reply:** This is only descriptive, to show the raw data.

Lines 259 – 260: I would cut this sentence off at “natural forest”. It was shown to support
high richness, including of forest specialists and understory insectivores.

**Our reply:** we did as suggested

Line 261: I would lead with the significant result (FF with lower growth bar width) and then
follow with null findings.

**Our reply:** we have now modified the earlier analysis which was based on guilds to a species
level and we modified the discussion accordingly

Line 270-271: “most” – need to add percentage for this claim

**Our reply:** We have modified the sentence

Lines 278-279: I would rephrase this sentence away from bold claims

**Our reply:** we have now modified this sentence

Lines 283 – 284: again, I would back away from this overly-bold claim. I suggest rephrasing
to something like “our study identified no difference in FA between...”

**Our reply:** Following our new analysis on FA comparison between the species, we have
modified this sentence

Lines 293 – 294: again, I think an overly bold statement. Your study may add some support
to this idea, but I wouldn’t state that conversion does not have a dramatic impact.

**Our reply:** we agree, and we have now modified the sentence

Figure 2 and 3: check lower left heading “insectivore dependy on diet” for misspelling

**Our reply:** Okay we did that, and corrected the errors

Evan R. Buechley

Reviewer: 2

Comments to the Author(s)

This study aims to assess avian habitat quality between natural forests and shade coffee
plantations in Ethiopia via individual measures of functional asymmetry and feather growth.
The overall aim of the study is excellent and attempts to address an area where much work is
needed. However, I am unable to evaluate the study and thus did not proceed to review the
results or discussion for one major reason.

The paper does not clarify the phenological life history of the 10 study species used. This is
crucial because the measures of local site habitat quality are entirely dependent on the
location of 1) where the measured tarsi were developed as nestlings, and 2) where the
measured feathers were grown. For example, if all 10 species were known to breed on site,
and it was assumed that the individuals did not disperse to or from elsewhere, then the tarsus
growth could be assumed to be influenced by the conditions of the study site. However, if the
bird was captured after it had migrated from its breeding location elsewhere onto the study
site, then the growth differences of the tarsus would reflect the habitat quality or conditions of
the site where the development took place. It is worrisome, for example, that juveniles were
not included in the study even though these would be an example of individuals for which
tarsi development location would have been known. Similarly, it is not clarified where the
rectrices feather growth took place and this is typically different depending on species and
age. In first year birds, the rectrices would have grown in on the breeding grounds. If the
captured birds were after-first-year adults, the rectrices might have molted in during different
time periods and at different locations depending on the species. If this information is
unknown then the only way to know for certain that a feather was grown at the study site
would be to pluck the feather of unknown origin and then pluck the regrown feather from the
recaptured bird. Thus the feather growth rate would reflect conditions at that specific site and
not the location where the original plucked feather was grown.

It is my assumption that this information was taken into account and it is known but simply
not reported. For example, it could be that the ten species are known to breed (and thus
develop their tarsi) on site. It could be that the species are migrants from another breeding
site but they moult rectrices during the time period in which they are captured on site, or they
are year round residents and thus perform their moulting cycles on site (and do not move
between the forest and shade coffee). I am happy to review the paper again if this is the case.
However, if this type of information is unavailable then I am afraid it is impossible to know
whether the tarsi and feather measures reflect what the authors assume they reflect.

**Our reply:** We ring birds for more than three years in this particular study sites and we never
come across individual birds those move from one fragment to another. Individual birds were
caught first as immature later as adult rewrap in the same sites year after years. From this we
conclude most of the birds born and grow on the same sites, and not migrate from one forest
to other. We have this experience from our previous study on bird species richness, diversity

and community structure in the same study sites using mist netting procedures where all
captured individuals ringed (Buechley et al., 2015).

Buechley ER, Şekerciöglu ÇH, Atickem A, Gebremichael G, Ndungu JK, Abdu B, Beyene T,
Mekonnen T, Lens L. (2015). Importance of Ethiopian shade coffee farms for forest
bird conservation. *Biological Conservation* 188:50–60.

I have made other less crucial comments in-text of the pdf that should also be addressed if
revised.

Sincerely,

Sacha Heath, PhD

Ecology Graduate Group

University of California, Davis

*****

Line 27 and 28, I suggest making explicitly clear the two step assumption and prediction
here, which then leads to your final indirect assumption (prediction):

- 1. You *assume* (presumably based on previous work that will be detailed in the text) that
higher FA and narrower feather growth bars indicate comparatively lower Habitat Quality,
and
2. You *predict* lower habitat quality in shade coffee vs. natural forest (as indicated by higher
FA and narrower growth bars).

**Our reply:** we have now clarified the two assumptions clearly in the updated manuscript
version

Line 28, page 2

Line 28 page 2, I think I understand what you are getting at here, but perhaps add a sentence
or a few words to clarify. The reader might wonder why it is suggested that these habitats be
converted to shade coffee vs. restored to healthy forests. So adding something about degraded
forests being converted into shade coffee as compensation for lost coffee production due to
forest protection would clarify. Also, is shade coffee considered to be of higher quality than
degraded forest?

**Our reply:** Okay we have now clarified this sentence

Line 28 page 2, Replace with the actual number

**Our reply:** Okay, we have made the change accordingly

Line 50, insert bird before species

**Our reply:** we have inserted bird as suggested

Line 51, page 3, in shade coffee farming have been shown to be comparable to natural forests
in some cases (8-10), and not in others (several citations - see reviewr comments).

**Our reply:** we fully understand this, and have tried to clarify the sentence

Line 51, change was to has been

**Our reply:** we have made the changes accordingly

Line 51, in shade coffee farming have been shown to be comparable to natural forests in
some cases (8-10), and not in others (several citations - see reviewr comments).

**Our reply:** Okay, we have done as suggested [11-14] line 48.

Line 53, I made a few suggestions for how to tone down this statement. There has been
evidence presented, as the introduction suggests, that remnant forest and shade coffee are
comparable in terms of these bird measures, but I would not go so far to state “no negative
effect” because the story is more nuanced. There are several examples where shade coffee
measures of bird species richness, etc, were lower than native forest remnants (and this
should be noted). For example, see this review and citations therein:

Komar. 2006. Ecology and conservation of birds in coffee plantations: a critical review. Bird
Conservation International 16:1-23:

“As expected, bird species-richness and diversity in shaded coffee plantations tend to be
lower than in nearby forest patches (Terborgh and Weske 1969, Beehler et al. 1987, Thiollay
1995, Wunderle and Latta 1996, Estrada et al. 1997, Greenberg et al. 1997a, Petit et al. 1999),
although studies in some landscapes have documented the same or even higher species
richness as natural forest (Aguilar- Ortiz 1982, Greenberg et al. 1997b, Shahabuddin 1997,
Tejada-Cruz and Sutherland 2004). Species diversity in shaded plantations is nearly always
reported to be considerably higher than in open-sun plantations or other types of monoculture
(Beehler et al. 1987, Wunderle and Latta 1996, Estrada et al. 1997, González 1999, Petit et al.
1999, Tejada-Cruz and Sutherland 2004).”

**Our reply:** Okay, we have now modified the paragraph to show the importance of shade
coffee for biodiversity conservation is not always the case

Line 56 This sentence could be broken into two or three sentences for clarity.

**Our reply:** Okay, we have broken the sentence in to two.

Line 62, Please provide a few citations of examples.

**Our reply:** Okay we have included few citations

Line 65, Insert “relative levels of”

**Our reply:** we have inserted “relative levels of” as suggested

Line 65, be careful with how this is stated. Quality could be considered lower or higher
depending on the specific comparisons of these stress measures. For example, it is not stated
that body mass, size, etc. is explicitly lower and thus equates with lower quality... stay more
general here.

**Our reply:** Okay. We did as suggested

Line 65, delete lower

**Our reply:** We did the change as suggested

Line 66 to 70, These are two important points that aren't necessarily connected; please break
up the long sentence into two.

**Our reply:** We did the changes as suggested

Line 71 to 78, No need to go into an exhaustive review of these two measures, but since the
entire study is based on FA and feather growth, it would help the reader to have a bit more
justification for the validity of the methods. Please more explicitly explain the links between
FA and Feather Growth to fundamental life history components. Also, what do you think
about the criticisms of ptilochronology? (for example, see citations below). Again, no need
for too much detail but please add in a few more clarifying sentences to make the life history
links and very briefly address the criticisms of Murphy.

Murphy and King. 1991. Ptilochronology: a critical evaluation of assumptions and utility.
The Auk 108:695-704

Murphy. 1992. Ptilochronology: accuracy and reliability of the technique. The Auk 109:676-
680.

**Our reply:** Okay we have include this debate in the discussion section

Line 87, delte however

**Our reply:** we have deleted the word “delete”

Line 88, delete therefore

**Our reply:** we have made the change as suggested

Line 97, replace ,,,, by “is of lower quality than natural forests”,

**Our reply:** We did the change as suggested

97, replace ,,,,, by “predicted”

**Our reply:** we did as suggested

Line 99, Based on next two comments, please rework this section. I can see why nestling
growth effects might affect bird health in later years, but that isn't described here.

**Our reply:** Okay, we have done as suggested lines 101 & 102.

Line 101, so this demonstrates previous effects, not subsequent effects as stated earlier in the
sentence.

**Our reply:** We did as suggested.

Line 102, But stress induced during one feather growth period does not necessarily indicate
anything about stress during future molt periods. It only indicates stress levels during the
particular growth phase measured.

**Our reply:** Yes, that is true. In the current version we have addressed this issue lines
102&103.

Line 126 , Do these birds occupy the sites year round? During what time period do they
moult their rectrices? Do they grow their rectrices on site or during another season at another
site? If measured feathers were grown offsite, then feathers must be plucked and feathers
regrown onsite must again be plucked and measured to quantify local conditions. This point
is crucial to the entire study so please clarify.

**Our reply:** The species included in this study are resident bird species and they stay in the
sites year round. They grow the rectrices on the sites lines 131&132.

Line 128, delete already

**Our reply:** we have made the change

Line 138, Birds were captured year round? So breeding, migrant/transient, and overwintering
birds were all included or were only birds that developed as nestlings and/or moulted in
rectrices on site used for analysis? Crucial point, please clarify.

**Our reply:** No, birds were captured from 9 December, 2013 to 30 March, 2014 and 19
December, 2014 to 10 June, 2015. Migrant birds/transient, and overwintering birds were not
included this analysis. We only analyzed the data of bird that were developed as nestling
and/or moulted rectrices on the site.

Line 139, mist-netting procedures.

**Our reply:** we have made the change accordingly

Line 140, replace controlled by Checked

**Our reply:** we have made the change accordingly

Line 140, delete

**Our reply:** we have made the change accordingly

Line 142, why?

**Our reply:** We, fully admit this is a mistake. And corrected it in the current version, even the
immature individual were also included in tarsus analysis.

Line 152, By two different observers?

**Our reply:** the observer was the same person

Line 154, replace ,,,,, by retrix

**Our reply:** we did the change as suggested

Line 155, what is the life history of these birds? When were these feathers grown? Where
they grown on the study site or were these birds migrants and thus the feather growth took
place in a different location? It is impossible to evaluate this study without knowing this.

**Our reply:** Our study species were resident, stay round in a site where they were captured.
Feather growth and moult were takes place in the same sites (in our study sites). These
species do not migrate from one forest fragment to the other. Even if we made many years
ringing effort, we did not captured individuals that captured in one forest fragment in other.

Line 160, Again. This procedure completely depends on when these feathers were grown.
Are these birds year round residents of their various plots and therefore the feather growth
reflects the conditions of that habitat? Or, are any of these birds migrants that only use the
study sites during a certain time period? Do they occupy the sites when rectrices molt occurs?
If the feathers grow in offsite, then the growth bars do not represent the quality of the habitat
in question.

**Our reply:** Bird species of our study are resident and stay year round in sampling sites.
Previous work of Buechley et al., 2015 have showed that these bird species have breeding
population in this particular study plots. Gebremichael et al., unpublished data also showed
that these bird species were breed and moult in the study plots of our study area.

Line 194, perhaps begin this section by stating that you followed the analysis procedures
described in citation 53.

**Our reply:** we have made the changes accordingly

Line 198: is this the random slope for side?

Our reply: yes, the random side is for random slop that estimates FA within individual.

Line 224, because different species were grouped into functional groups, species should
probably also be included as a random effect, otherwise a particular species within a group
might bias the results.

**Our reply:** This is true, and in the updated manuscript we re-analysed the data as species
comparison rather than the former analysis which is based on guilds of functional guilds

Appendix D

Fluctuating asymmetry and feather growth bars as biomarkers to assess habitat quality of shade coffee farming for avian diversity conservation

Reviewer comments to Author:

Reviewer: 2

Comments to the Author(s)

I am happy to once again review this paper which aims to compare avian habitat quality between shade coffee and nearby natural forests in Ethiopia. The paper is an important contribution in that unlike most studies of this sort, it aims to compare biomarkers rather than community measures. Both approaches are important, but as the authors note, occupancy alone is not always the best measure of habitat quality.

This version and the author responses addressed my major concern from my previous review. I also agreed with the primary comment of the 2nd reviewer that it is more appropriate to compare these biomarkers between habitat types within the same species, rather than between guilds.

Our reply: Thanks. Yes, we also found this comment very constructive and did the change accordingly.

The reworked paper is much sounder, though I recommend that a major revision is still needed. I think that all of the pieces are here (in terms of the data, the design, and the statistical analysis) to present a simple but important result, but I am afraid that the grammar and writing make the story more complicated than it needs to be. I think it would be very helpful for the authors to request assistance from a copy editor. I am very familiar with the general topic of the paper, thus with some effort I was able to decipher what the paper was explicitly attempting to convey. I fear that readers less familiar with the topic will not be able to do so. I provided a few comments and suggestions for sentences where I found this to be especially true, but I would recommend a thorough check on both grammar and sentence structure so that the primary message is conveyed more clearly.

Our reply: In our resubmitted version, we incorporate all changes suggested by the reviewers and improve the language and presentation in general.

I am not familiar with the details of analysis of FA, but I think that too much detail of this analysis is provided in the results section. The main aim of the paper is to compare habitat quality (via biomarkers) between the two habitat types; I suggest leading the results section with this result. As it currently stands, this main result is not addressed until the end of the second paragraph.

Our reply: That is true, and we also noticed a bit complex presentation for the result section in other published papers focused on the FA. This is mainly due to the need of clarifying the validity of several parameters in the result section before providing the final conclusion of the FA. In our updated manuscript, we slightly modify the language and presentation but keeps the analysis which are needed to show the validity of the data.

In reference to a comment from my previous review, the authors stated that they addressed this in the discussion, however this version does not contain any mention of this in the discussion. I actually suggest that this instead be briefly addressed in the methods instead. The original comment was: “No need to go into an exhaustive review of these two measures, but since the entire study is based on FA and feather growth, it would help the reader to have a bit more justification for the validity of the methods. Please more explicitly explain the links between FA and Feather Growth to fundamental life history components.

Our reply: we misunderstood the comment in our previous revision, and emphasis the need of combining the two methods in the discussion. We now include the validity and links between FA and Feather Growth to fundamental life history components in the introduction as follow,

While there are exceptions [32], studies have shown that asymmetric individuals exhibit lower fitness affecting populations in the long term [33-37].

For example, in a study of Barn Swallows (*Hirundo rustica*), Møller [38] reported a significant increase in male tail feather FA in birds captured in Chernobyl after the 1986 nuclear accident compared to pre-1986 museum specimens from the same site. Furthermore,

Møller [38] indicated that more asymmetric males of this species bred later than symmetric ones. Also, birds living in low quality sites (food less abundant) are expected to show narrower growth bars than birds in sites with higher food availability [28,29,39]. Carlson [39] demonstrated a positive correlation between feather growth bar widths with food availability in the territories of White-backed Woodpeckers (*Dendrocopos leucotos*). Although the use of growth bars as an index of nutritional condition in wild birds seems plausible and has been used in many studies [e.g. 28, 39–41], others have criticized the validity of this technique for the lack of precision [42,43].

Also, what do you think about the criticisms of ptilochronology? (for example, see citations below). Again, no need for too much detail but please add in a few more clarifying sentences to make the life history links and very briefly address the criticisms of Murphy.

Murphy and King. 1991. Ptilochronology: a critical evaluation of assumptions and utility. The Auk 108:695-704

Murphy. 1992. Ptilochronology: accuracy and reliability of the technique. The Auk 109:676-680.”

Our reply: That is correct, and we briefly mentioned the controversial criticisms of ptilochronology in the introduction section as stated above

Finally, I see that the paper does link to the Dryad Digital Repository. However, this is done in the literature cited section and I just happened to run across it. I don't believe the citation (#77) is referred to anywhere in the text. This is the reason that I checked "no" for the two questions pertaining to the accessibility of the supporting data. I recommend making this link more clear by adding it as a supplement or mentioning it in the acknowledgements.

Our reply: We have now incorporated the citation to the Data accessibility section

“Data accessibility: Data available at the Dryad Digital Repository [87]”

I am happy to review this paper again.

Our reply: Thanks you for your constructive comments and your interest in improving the manuscript further

Reviewer one (Comments here are Copied from the PDF file)

1. *Line 19: The English could use some work in the abstract; I have provided a few suggestions.*

Our reply: we have now incorporated all comments as suggested here and substantially improve the language across the manuscript.

2. *Line 20: Shouldn't this be ..."combining sustainable coffee production and biodiversity conservation"?*

Our reply: we did the change as suggested

3. *Line 22: Correct and clarify the language here. These indicators don't have "negative effects" as indicators for habitat quality. I think what you are trying to say is that indices such as abundance and diversity are not always good indicators of habitat quality, because lower quality individuals may be occupying a site only because they have been pushed out of higher quality sites by dominant or higher quality individuals. Seek a way to explain this more clearly. -or simply, "indices such as diversity and abundance are not always good indicators of habitat quality" would suffice (and then back up a bit in the intro with citations).*

Our reply: However, diversity and abundance are not always good indicators of habitat quality because there may be a lag before population effects are observed following habitat conversion. Therefore other indicators of habitat quality should be tested. In this paper, we investigate the use of two biomarkers: fluctuating asymmetry (FA) of tarsus length and rectrix mass, and feather growth bars (average growth bar width) to characterise habitat quality of shade coffee and natural forests.

4. *Line 26: For clarity, I suggest separating these sentences and rewording to something such as "...width) as indicators of habitat quality. We predicted higher FA and narrower feather growth bars in shade coffee versus natural forest, indicating higher quality in the latter."*

Our reply: We did the change accordingly

5. **Line 28:** Insert “FA and feather growth in”.

Our reply: We did the change accordingly

6. **Line 29:** Delete “for tarsus length and rectrix mass”

Our reply: We did the change accordingly

7. **Line 29:** Replace “both” by *shade coffee and natural forest*

Our reply: We did the change accordingly

8. **Line 30:** Delete “traits”

Our reply: We did the change accordingly

9. **Line 30:** Replace “showed no difference” with “was not different”

Our reply: We did the change accordingly

10. **Line 31:** Replace “no difference was found in growth bar widths in any of these species between shade coffee and natural forest” by “we found no difference in feather growth between shade coffee and natural forests for any species”

Our reply: We did the change accordingly

11. **Line 34:** Replace “avian conservation” by “avian conservation for the species we examined”

Our reply: We did the change as suggested

12. **Line 37:** *Introduction section, Also work on English grammar in the introduction.*

Our reply: We did the change as suggested

13. **Line 50:** *Delete “it has been concluded that”*

Our reply: We did the change as suggested

14. **Line 50:** *Replace “play” by “be”*

Our reply: We did the change as suggested

15. **Line 51:** *Replace role by tool for*

Our reply: We did the change as suggested

16. **Line 51:** *Delete when combining and replace by in*

Our reply: We did the change as suggested

17. **Line 56:** *clarify: long term effects of what?*

Our reply: We have clarified this in the introduction section as follow

“demographic changes in the distribution of sex or age classes, changes in rates of survival or reproduction, which results from biotic changes, are possibly long-term effects. Demographic changes are unlikely to be picked up by community-level studies on demography and α -diversity alone if surveys are done shortly after natural forests have been transformed into shade coffee forests”

18. **Line 56:** *What are "these" referring to?*

Our reply: *"these" is to referring* demographic changes, and we now clarify it in the updated manuscript version

19. **Line 69:** *Please correct grammar here.*

Our reply: Okay, we have now corrected the grammar as follow

Two potential useful biomarkers are fluctuating asymmetry (FA), i.e. measuring small random deviations from left-right symmetry in bilateral traits [24-27] and feather growth bars, i.e., measuring width of alternating dark and light growth-bars on bird feathers [ptilochronology, 28-31].

20. **Line 125:** *Please include a sentence or two here describing why you selected these 5 species.*

Our replay: In the current version we addressed this as follows

The bird species included were chosen if 1) they represented a breeding population in our study area, and 2) their sample sizes from both forest types were adequate and comparable. These criteria were met by five species: Abyssinian Ground Thrush, Rüppell's Robin-chat, Grey-backed Camaroptera, African dusky Flycatcher and Tambourine Dove (Table 1).

Line 134: *delete 351*

Our reply: We have made the change

24. **Line 134:** Presumably you captured more than 5 species. This might be the a good place to explain why you selected the 5 (as suggested above)

Our reply: We addressed why we selected five species for two reasons as mentioned above.

26. Line 137: change the place

Our reply: we did the changes accordingly

21. **Line 138:** Replace “aged, weighted” to “and aged and weighed them”

Our reply: We did the change accordingly

22. **Line 147:** Replace “Tarsi length”, by “We measured tarsi length as the”

Our reply: We did the change accordingly

23. **Line 149:** delete “was measured in the field to the nearest 0.1 mm using Vernier callipers” Already stated above.

Our reply: we now deleted the sentence as suggested

24. **Line 151: Delete** “The second outer most right and left rectrices (fully-grown) rectrices feathers were collected from every bird captured for rectrix mass and growth bar measurements [33]”. Already stated above.

Our reply: the sentence is now deleted

27. **Line 153:** replace “the” by collected

Our reply: We did the change accordingly

25. **Line 158:** Replace “follow” by “follows”

Our reply: We did the change accordingly

26. **Line 183:** Data analysis “I don't think it is necessary to separate these out into different subsections. Combine and reduce repeated wording when necessary”

Our reply: Yes, we also have tried to make the analysis part reduced and with of no subsection. This however makes the readers a bit confusing as the FA statistical analysis is not straightforward and needs several analysis as validity testing before the

final analysis of FA. Hence, we prefer to keep it separated unless the reviewer found it really important to make it without subsections.

27. **Line 185:** *Delete Individual signed*

Our reply: We did the change accordingly

28. **Line 187:** *For clarity, I suggest referring to "unsigned FA" instead as "absolute FA" - which is a clearer description of this measure.*

Our reply: Across the published materials of FA, unsigned FA is the most common word and we prefer to keep it, but we also mention absolute value in this sentence for clarity.

29. **Line 187:** *Replace "Individual unsigned FA (the magnitude of signed FA) values were the absolute values of signed FA" by "We then calculated the absolute value of differences in measurements (absolute FA) to describe the magnitude of FA".*

Our reply: We have made the change accordingly, except we mention unsigned FA in the bracket as this is the word used across the published materials in FA studies

30. **Line 194:** *Please clarify this. I think you mean that you used random intercepts for side nested within individual id (which would make the most sense).*

Our reply: we learn the reviewer find it okay in his/her latter comment, line 197.

31. **Line 225:** *Not knowing much about FA comparisons, it is not clear to me that these sorts of details are necessary or not. Even so, the reader is much more interested in the comparison between the two habitat types, then the statistical details of normality, etc. I suggest declaring the most interesting results first and following up with detail in the next, or relegating those to the tables.*

Our reply: We agree on the reviewer comment here on the long descriptions of method and analysis section before the main analysis which determines the presence or absence of the FA. Unlike most of other studies, the FA is very much dependent on analysis of the validity

which makes FA papers a bit long and not easy to read. Yet, we have tried to clearly show the validity of the data to be used for the FA analysis. We prefer to keep it like this unless we get the reviewer find the change is really important.

32. **Line 228:** *Why is Kurtosis being highlighted here? Is this a measure that is important for comparing FA? If so, please describe earlier.*

Our reply: It was highlighted in our previous submission by mistake, and we now have corrected it.

33. **Line 241:** *This is the result that should be highlighted first. Also, because "all species" is used, no need to list all 5 of them again.*

Our reply: Yes, we deleted the list of the five species. For the reason we have mentioned above, the test of the validity of the data for FA analysis, we prefer to keep them as they were.

34. **Line 244:** *I think what is meant here is that there were no significant differences (not significant when using $\alpha = 0.5$) between measures from different forest types for any species, but there was a tendency for absolute FA to be higher in shade coffee than in natural forests for 3 species. I would be more explicit about this. As this is now written, the two results seem to contradict each other. It took me a few reads to realize that you were simply saying that there tended to be a difference, even though it was not a statistically significant difference.*

Our reply: That is right. We now mentioned the differences as in spite of the lack of a significant difference. But in the current version, we have tried to improve the presentation to make this clear.

37. **Line 247:** same as the pervious comment.

Our reply: same as above

35. **Line 253:** *Again, perhaps reword this (as suggested in previous comment) to highlight the trends observed (even if they were not statistically significant).*

Our reply: We now make change accordingly.

36. **Line 263:** *delete “based on demographic studies”*

Our reply: Our study of the effect of shade coffee is different from the previous study because of our approach using biomarkers while previous studies used demographic studies, diversity and abundance. We still think mentioning this aspect is important.

37. **Line 266:** *delete including in forest species.*

Our reply: we did the change as suggested

38. **Line 269:** *negatively affected by what?*

Our reply: we now make it clear; it is by modification of forest to shade coffee

39. **Line 271:** *Replace “Currently, in our study area, the negative effects of shade coffee farming is not evident in any of the bird species” by Based on the biomarkers we compared, it appears that shade coffee farms and natural forests are of similar quality for the five species we examined.*

Our reply: we did the change as suggested

Comment of reviewer for the Supplementary materials

Rectix by Rectrix in table S1

Our reply: We did the change

Appendix E**ROYAL SOCIETY
OPEN SCIENCE****Fluctuating asymmetry and feather growth bars as
biomarkers to assess habitat quality of shade coffee
farming for avian diversity conservation**

Journal:	Royal Society Open Science
Manuscript ID	RSOS-190013.R1
Article Type:	Research
Date Submitted by the Author:	02-May-2019
Complete List of Authors:	Gebremichael, Gelaye; Jimma University, Biology Department; Ghent University, Department of Biology Alemu, Diress ; Universitetet i Oslo Det Matematisk-naturvitenskapelige Fakultet, Department of Biosciences Bunnefeld, Nils; University of Stirling Zinner, Dietmar ; Germany primate center Atickem, Anagaw; Germany primate center , Cognitive Ethology Laboratory
Subject:	ecology < BIOLOGY
Keywords:	bird community, bird species, Ethiopia, fluctuating asymmetry, ptilochronology
Subject Category:	Biology (whole organism)

Author-supplied statements

Relevant information will appear here if provided.

Ethics

Does your article include research that required ethical approval or permits?:

Yes

Statement (if applicable):

This project was carried out in accordance with the ethical standards for research from Jima University, Ethiopia and Ethiopian Wildlife Conservation Authority (EWCA), and the project was endorsed by EWCA.

Data

It is a condition of publication that data, code and materials supporting your paper are made publicly available. Does your paper present new data?:

Yes

Statement (if applicable):

Data available at the Dryad Digital Repository
<https://datadryad.org/review?doi=doi:10.5061/dryad.cb0d8ft>

Conflict of interest

I/We declare we have no competing interests

Statement (if applicable):

CUST_STATE_CONFLICT :No data available.

Authors' contributions

This paper has multiple authors and our individual contributions were as below

Statement (if applicable):

Authors' contributions: GG did the fieldwork and collected the data. GG and AA drafted a first version of the manuscript which was improved by DZ. GG, DT, and NB did the statistical analyses. All authors reviewed the manuscript and gave their final approval for publication.

**Fluctuating asymmetry and feather growth bars as biomarkers to assess habitat quality**
**of shade coffee farming for avian diversity conservation**

Gelaye Gebremichael^{1,2}, Diress Tsegaye³, Nils Bunnefeld^{4,5}, Dietmar Zinner⁶, Anagaw
Atickem⁶

¹Terrestrial Ecology Unit (TEREC), Ghent University, K.L. Ledeganckstraat 35, 9000,
Ghent, Belgium

²Jimma Universities, College of Natural Sciences, P.O. Box 378, Jimma, Ethiopia

³University of Oslo, Department of Biosciences, Postboks 1066 Blindern, 0316 Oslo, Norway

⁴Biological and Environmental Sciences, Faculty of Natural Sciences, University of Stirling,
Stirling FK9 4LA, UK

⁵School of Geosciences, University of Edinburgh, Edinburgh, EH9

⁶Cognitive Ethology Laboratory, German Primate Center, Leibniz Institute for Primate
Research Kellnerweg 4, 37077 Göttingen, Germany

**15 Abstract**

Shade coffee farming has been promoted as a means of combining sustainable coffee
production and biodiversity conservation. Supporting this idea, similar levels of diversity and
abundance of birds have been found in shade coffee and natural forests. However, diversity
and abundance are not always good indicators of habitat quality because there may be a lag
before population effects are observed following habitat conversion. Therefore other
indicators of habitat quality should be tested. In this paper, we investigate the use of two
biomarkers: fluctuating asymmetry (FA) of tarsus length and rectrix mass, and feather growth
bars (average growth bar width) to characterise habitat quality of shade coffee and natural
forests. We predicted higher FA and narrower feather growth bars in shade coffee forest
versus natural forest, indicating higher quality in the latter. We measured and compared FA
in tarsus length and rectrix mass and average growth bar width in more than 200 individuals
of five bird species. The extent of FA in both tarsus length and rectrix mass was not different
between the two forest types in any of the five species. Similarly, we found no difference in
feather growth between shade coffee and natural forests for any species. Therefore, we
33 30 conclude that shade coffee farming in Afrotropical rain forests of Ethiopia seems to have no
34
31 negative consequences on avian populations for the species we examined.

[revised manuscript text omitted]

between-side difference in both traits (tarsus: $ICC = 22.3\%$ and rectrix mass: $ICC = 2.5 * 10^{-5}$).
Furthermore, the level of unsigned FA both in tarsus length and rectrix mass did not differ

between natural forests and shade coffee forests for all species (all: $P = 0.0521$; Table 2 & 3).

Although not significant, ~~Abyssinian Ground Thrush~~, African dusky Flycatcher, and
Tambourine Dove demonstrated higher FA both in tarsus length and rectrix mass in shade
coffee than their conspecifics in natural forest (Fig. 2 a & b). Again, although not significant,
Rüppell's Robin-chat showed higher FA only in tarsus length in shade coffee forests.

3.2 Growth bar width

Average growth bar width of the five species did not differ between shade coffee and natural
forests (all species: $P = 0.4$; Table 4). Furthermore, mean rectrix length did not vary between
shade coffee and natural forests (all species: $P = 0.2$; Table S3). Tambourine Doves showed
a tendency for wider average growth bar in the natural forests, whereas Grey-backed
Camaroptera showed narrower growth bar in the natural forest (Fig. 3).

4. Discussion

Several studies based on species richness, abundance and community composition have
revealed that shade coffee farming can have an important role in biodiversity conservation [8-
10,15,16]. Yet, the intensification of coffee management necessitates further monitoring of
habitat quality in shade coffee farms with methods that can detect changes before the habitat
quality is degraded to such an extent that it affects avian community demography [40,78,79].
Here, we applied biomarkers to re-assess the quality of Ethiopian tropical rain forest used for
shade coffee farming which has previously been reported to support as rich avian biodiversity
as natural forest based on demographic studies [50].

FA in both traits (tarsus length and rectrix mass) were not significantly different in
birds from shade coffee or natural forests. Similarly, the analysis of feather growth bar widths
revealed no significant difference between birds of the two forest types. Our results are in

contrast to Buechley et al. [50] who reported that forest specialist guilds and understory-
insectivores were more negatively affected than birds of other guilds in shade coffee when
compared to those in natural forest.

Based on the biomarkers we compared, it appears that shade coffee farms and natural
forests are of similar quality for the five species we examined. Yet, the long-term impact of
the on-going shade coffee plantation modification and management should still be closely
monitored. Shade coffee farm management, involving the selective removal of certain tree
species, may decrease habitat quality of shade coffee plantation for avian biodiversity [5,80].
In many coffee growing countries across the globe, shade coffee management practices are
intensifying, which is believed to be reducing plant diversity and canopy cover [79,81-83],
and hence, also reducing the quality or quantity of feeding, nesting or hiding resources for
animal populations. Such tendencies underline the need for close monitoring of habitat
quality of shade coffee forests, for example, with biomarkers such as FA and feather growth
bars, which can detect changes of habitat quality before it is degraded to the extent that it
affects avian diversity and community demography [18,40,84,85]. Globally, about 21% of
bird species are currently at risk of extinction and 6.5% are functionally extinct [86],
underlining the urgency of monitoring efforts.

Future studies also may combine FA analysis in tarsus length and rectrix mass as
implemented in this study to provide a more comprehensive result. Only a few studies
combined FA and feather growth bar width analysis which showed the two markers showed
consistent result or only one of the markers show the effect of fragmentation [25,33,76]. We
recommend future studies combine the use of both biomarkers to robustly assess the habitat
quality following habitat conversion. FA in particular is a sensitive measure of habitat quality
and could be very useful in providing an early warning sign of negative effects of habitat
change before any demographic changes in affected populations can be observed [40].

**5. Conclusion**

Our results are consistent with results of many of the demographic studies that shade coffee
farming in the Ethiopian highland can be compatible with avian diversity conservation. ~~No~~
early sign of negative effect ~~was observed from biomarker studies~~. Buechley et al. [50]
showed that shade coffee farming ~~has an important contribution for the~~ avian biodiversity
conservation ~~though forest specialist birds are~~ disproportionately affected in shade coffee
farms. While the current biomarker-based study did not reveal any single of negative effect, it
is important to continue ~~monitoring~~ the Ethiopian shade coffee ~~for the avian biodiversity~~
~~conservation as the shade coffee plantation is~~ still under pressure of modification and subject
~~for the~~ increasing level of fragmentation. We also recommend combining FA and feather
growth bars as biomarkers to provide ~~more comprehensive results in studies~~ of habitat quality
for avian biodiversity conservation ~~than using only one of them~~.

**Ethics:** This project was carried out in accordance with the ethical standards for research
from Jimma University, Ethiopia and Ethiopian Wildlife Conservation Authority (EWCA),
and the project was endorsed by EWCA.

**Data accessibility:** Data available at the Dryad Digital Repository [87]

<https://datadryad.org/review?doi=doi:10.5061/dryad.cb0d8ft>

**Authors' contributions:** GG did the fieldwork and collected the data. GG and AA
[revised manuscript text omitted]

364x258mm (96 x 96 DPI)

Figure 2. Unsigned FA (mean \pm SE) in five bird species (species: ADF = African dusk Flycatcher; AGT = Abyssinian Ground Thrush; GBC = Grey-backed Camaroptera, RRC = Rüppell's Robin-chat; TD = Tambourine Dove).

Figure 2. Unsigned FA (mean \pm SE) in five bird species (species: ADF = African dusk Flycatcher; AGT = Abyssinian Ground Thrush; GBC = Grey-backed Camaroptera, RRC = Rüppell's Robin-chat; TD = Tambourine Dove).

240x155mm (96 x 96 DPI)

Figure 3. Average growth bar width (mean \pm SE) in five bird species (species: ADF = African dusk Flycatcher; AGT = Abyssinian Ground Thrush; GBC = Grey-backed Camaroptera, RRC = Rüppell's Robin-chat; TD = Tambourine Dove).

130x104mm (96 x 96 DPI)

Table 1. The number of individual birds assessed for growth bar widths (rectrices masses) and tarsus lengths per species and forest type (natural forest [NF], shade coffee forest [SC]).

species	Common name	growth bar width			tarsus length		
		SC	NF	total	SC	NF	total
Muscicapa adusta	African dusky Flycatcher	10	10	20	21	22	43
Camaroptera brevicaudata	Grey-backed Camaroptera	8	7	15	16	12	28
Cossypha semirufa	Rüppell's Robin-chat	66	30	96	85	40	125
Turtur tympanistria	Tambourine Dove	37	40	77	47	50	97
Zoothera piaggiae	Abyssinian Ground Thrush	5	10	15	13	13	26
Total		126	97	223	182	137	319

Table 2. Unsigned FA estimates in tarsus length for five bird species in two habitat types (natural forest and shade coffee forest). Linear mixed models were fitted to the data where unsigned FA were included in the models as a response variable, forest types as fixed effect and sampling location as random effect for Abyssinian Ground Thrush (AGT), African dusky Flycatcher (ADF), Grey-backed Camaroptera (GBC), Rüppell's Robin-chat (RRC) and Tambourine Dove (TD).

species	Effect	Estimate	SE	df	t	p
AGT	Intercept	-2.566	0.2491	23	-10.3	<0.0001
	Forest type (natural)	-0.5699	0.3454	23	-1.65	0.1126
ADF	Intercept	0.4219	0.01029	17	41.0	<0.0001
	Forest type (natural)	-0.0210	0.021	40	-1.67	0.1031
GBC	Intercept	0.7796	0.149	28	5.24	<0.0001
	Forest type (natural)	-0.0602	0.227	28	-0.27	0.7926
RRC	Intercept	0.0195	0.004	10	5	<0.0034
	Forest type (natural)	-0.0091	0.0085	11	-1.08	0.3018
TD	Intercept	3.3257	0.3577	86	9.30	<0.0001
	Forest type (natural)	-0.3899	0.4948	86	-0.81	0.4225

Table 3. Unsigned FA estimates in of rectrix masses for five bird species in two habitat types (natural forest and shade coffee forest). Linear mixed models were fitted to the data where unsigned FA were included in the models as a response variable, forest types as fixed effect and sampling location as random effect for Abyssinian Ground Thrush (AGT), African dusky Flycatcher (ADF), Grey-backed Camaroptera (GBC), Rüppell's Robin-chat (RRC) and Tambourine Dove (TD).

species	Effect	Estimate	SE	df	t	p
AGT	Intercept	1.51×10^{-5}	0.00037	15	4.12	0.0009
	Forest type (natural)	-9×10^{-4}	0.00045	15	-2.11	0.0521
ADF	Intercept	9.68×10^{-4}	0.000568	20	1.7	0.0391
	Forest type (natural)	-11.00066	0.000733	20	-0.89	0.3816
GBC	Intercept	1.35×10^{-6}	0	13	-0.86	<0.0001
	Forest type (natural)	1.8×10^{-5}	1.4×10^{-6}	13	-0.86	0.4038
RRC	Intercept	7.9×10^{-5}	2.3×10^{-5}	94	3.5	0.0007
	Forest type (natural)	1.8×10^{-5}	2.7×10^{-5}	94	0.67	0.5021
TD	Intercept	1.4×10^{-2}	0.004	66	3.69	0.005
	Forest type (natural)	-6.5×10^{-3}	0.005	66	-1.22	0.223

Table 4. Estimates of average growth bar width from mixed linear models, where average growth bar widths included as a response variable, forest types as fixed effect, whereby feather length included as fixed covariate and sampling location as random effect for Abyssinian Ground Thrush (AGT), African dusky Flycatcher (ADF), Grey-backed Camaroptera (GBC), Rüppell's Robin-chat (RRC) and Tambourine Dove (TD). Natural=Natural forest and TFL=mean of the second out most right and left rectrices.

species	Effect	Estimate	SE	df	t	p
AGT	Intercept	-7.012	7.9878	13	0.880	0.3959
	Forest type (natural)	1.456	0.9286	13	1.57	0.1409
	TFL	0.267	0.093	13	2.88	0.0128
ADF	Intercept	6.029	7.428	16	0.81	0.429
	Forest type (natural)	-0.287	0.5471	16	-0.53	0.429
	TFL	-0.287	0.547	16	0.77	0.451
GBC	Intercept	-4.687	6.44	8.42	-0.73	0.486
	Forest type (natural)	-0.83	1.12	6.1	-0.74	0.486
	TFL	0.38	0.105	8	2.3	0.0496
RRC	Intercept	6.70	4.9647	91	2.36	0.0206
	Forest type (natural)	-5.098	4.9647	88	-1.1	0.276
	TFL	0.098	0.0443	91	2.22	0.0292
	Natural*TFL	0.088	0.0776	88	2.0	0.2574
TD	Intercept	-0.316	0.138	66	1.87	0.979
	Forest type (natural)	0.984	5.038	69	0.07	0.9480
	TFL	0.2582	0.138	65	1.87	0.066
	Natural*TFL	-0.0163	0.175	69	-0.09	0.92

Appendix F

Reviewer comments to Author

Reviewer: 3

Comments to the Author(s)

This is an interesting study that uses fluctuating asymmetry (tarsus and tail feathers) and feather growth rates (ptilochronology) as biomarkers to compare birds breeding in shade coffee plantations vs native wet forests in the mountains of Ethiopia.

The topic is of general interest in that it may provide insight into whether habitat quality is equivalent between agroforestry versus native forests; not just that the species are retained in shade coffee plantations but that they are still able to sufficiently breed and not be sink populations.

Our reply: Thank you very much.

While I like the idea of this study, I have a couple of comments that I think the authors would have to address in order to convince readers of the effectiveness of their technique.

1. Validation of the Biomarkers

The first of these comments is regarding the “ground truthing” of these two biomarkers – while there are studies to suggest that both Fluctuating Asymmetry and Ptilochronology can be reliably associated with measures of individual condition, there are also studies that have failed to make these connections. In the current study, there is no independent measure of condition to which the measures of FA or Tail Growth are compared so as to show they can accurately be used as biomarkers. This assumption, though, is the basis for the entire study (L104-107) and recommendation for further use in other studies (L319-321). I suspect that the authors do not have data on reproductive success or other measures of condition that they can correlate asymmetry or tail-growth measures to, but at the very least they could compare these measure to one another.

The use of the two biomarkers in tandem, as suggested (L84-86, L303-305), would indicate they show the same pattern. As a result, there should be a strong negative correlation between level of asymmetry and tail feather growth bar width (little asymmetry should correspond to large growth bars, and high asymmetry with small growth bars). If the authors could show this association, it would help validate these as potentially sensitive biomarkers of condition.

Our reply: Thanks for the suggestion. Yes, many studies provide a strong support of the use of FA and Ptilochronology as sensitive biomarkers to study habitat modifications, though other studies do not get support for the use of the biomarkers.

Our study was not designed to test the validity of the biomarkers. We make the assumption that FA and Ptilochronology are sensitive biomarkers to effects of shade coffee management after reading several papers on the FA and Ptilochronology (Grubb and Cimprich 1990, Leung et al., 2001, De Coster et al., 2013, Costa et al., 2017, and many other studies cited in our paper).

Analysis of correlation between biomarkers also may not provide much information on the validity of the biomarkers. Biomarkers sensitivity varied a lot between traits used, species under

study and the type of the nature of the habitat change or stress factor. The presence or absence of correlations provides little information on the validity of the biomarkers.

In relation to our recommendation in using combined biomarkers of FA and growth bars is to emphasise the importance of using multiple biomarkers due to the fact that biomarkers varied a lot in their sensitivity for habitat changes. One may not see any difference in a given biomarker and may get a significant difference on the other biomarker for a given study comparing fragmented and continuous habitat.

In this specific study, we did “Bivariate Person correlation (r) between signed FA (in tarsus length and rectrix mass) and average growth bar width. Signed FA both in tarsus length and rectrix mass showed negative correlation with the average growth bar width (tarsus: $r = -0.121$, $N = 180$, $p = 0.106$; rectrix: $r = -0.073$, $N = 222$, $p = 0.330$)” (see, lines 202-205) revealing both of our biomarkers reflect the same trend when we come to effects of habitat fragmentation, increasing FA and decreasing of growth bars in fragmented habitat.

2. *Ptilochronology Measure*

The authors have measured growth bands on tail feathers plucked from birds at initial capture. While they have definitely refined on Grubb’s technique (e.g. digital vs manual measurement of individual vs groups of growth bars – L181-182 – and repeatability studies on measures that show remarkably high confidence in measurement – L191-194), there are still problems with measuring growth rates in feathers in this manner. First, as the authors indicate, feathers are moulted throughout adult life, so represent a snapshot of condition/food availability in the habitat during the time period of regrowth (L110-112). However, there is no indication in this study if all the birds from which feathers were drawn are of the same relative age (all adults, no juveniles) to suggest they all moulted their feathers during the same time period.

Our reply: That is right. We have now clarified that all birds we used in this study are adults as follow “Individual adult birds were used for the further analysis of average growth bar width, rectrix mass and mean rectrix length.” (See, lines 209 and 210).

Although the authors indicate all species are resident breeders in the region (L144), they don’t indication whether they are year-round residents on the exact territories in which they were captured during these studies, or whether they may have been occupying different areas during the period of feather regrowth. Without this information, you don’t have a firm basis to conclude that feather growth bars reflect growth in the habitats in which the birds are currently located. If shade coffee sites are sink populations, you could be measuring birds that grew their feathers in forested areas, then dispersed to shade coffee sites.

Our reply: Yes, we fully acknowledge the reviewer’s concern here. In our three years of study, we ringed all captured birds and never encountered a bird which was trapped in the other nearest fragment. On the other hand, adult and immature birds were re-trapped in the same site year after year revealing birds of our study are in fact resident and do not cross even to the nearest possible fragment site we studied. Our fragmented habitat and the natural forest are over 50 km apart which makes the movement of the birds we studied unlikely.

The advocated technique in ptilochronology studies by Grubb is to pluck feathers from the birds to induce regrowth, then recapture the birds and pluck the regrown feathers for measurement. This standardizes both the temporal period in which the feathers are regrown, as well as confirms they are regrown in the habitat the bird is currently occupying.

Our reply: Yes, as we have tried previously, our field data showed strong evidence for the study species habitat use and resident behavior. It is unlikely that our result is affected by growth or FA occurred other than the captured study. The sites of fragmented and natural forest are also far apart (over 50 km).

I suspect this is not possible in the current study, but the limitation of the technique you have used needs to be addressed and also highlighted as a possible source of not detecting differences (e.g. L 267-268).

Our reply: We have now included detailed information on the methodological problems and give more detail on our experience on the bird species habitat use and resident behavior.

And in the current version, we added the following sentences “Individual birds were caught first as immatures and later as adults in the same sites year after year. Therefore, technical limitation could not account for the absence of differences between habitat types” (see, lines: 312-314).

3. Native Forest and Shade Coffee sites being of “similar quality”

This is the conclusion that is drawn by the authors not finding differences in FA or Tail Growth bars (L289). First, I think you need to account for the validity of your biomarkers and limitations of your ptilochronology technique (above two points), but even then I would argue that your results hint that this is a false conclusion to draw. While not statistically significant, the FA data does show a pretty consistent pattern across 4 of the 5 species that there was slightly higher FA in the shade coffee sites than natural forest sites (Figure 2a). Only one species with extremely variable tarsal FA (Grey-backed Camaroptera) doesn't show this pattern, and two of the other species have suggestive P values in the 0.1 range. As you are using a proxy of condition, rather than a direct measure of condition, I would suggest you are a little more cautious. While you can't statistically conclude there is a difference, it doesn't mean the two habitats are necessarily equivalent.

Our reply: That is correct and we slightly modify our presentation of the effects of habitat fragmentation as follows “Although we did not find significant differences between habitats, there is a certain tendency of increasing FA in bird species in the fragmented habitat. Hence, it is crucial to closely monitor the avian community in the shade coffee habitat. (see, lines: 314-318).

Reviewer: 2

Comments to the Author(s)

The authors did a very thorough job of addressing my previous concerns and have tightened up the language substantially. This is a well written, succinct, and informative paper that will make a good contribution to the agroecology literature and to avian conservation practice in agroecosystems.

Our reply: Thanks.

I have provided a few minor edits and suggestions (one sentence in the abstract, some minor presentation suggestions in the results, and some changes to sentence structure and rewording for clarity in the conclusion are the most substantial of these). These should take very little time to address, after which I recommend accept as is without the need to review the MS again. The authors have demonstrated great care in responding to and addressing previous suggestions and it will be great to see this published.

Our reply: Thank you! We have incorporated all the comments and editorial suggestions in the current submitted version of the manuscript.

Journal Name: Royal Society Open Science

Journal Code: RSOS

Online ISSN: 2054-5703

Journal Admin Email: openscience@royalsociety.org

Journal Editor: Andrew Dunn

Journal Editor Email: openscience@royalsociety.org

MS Reference Number: RSOS-190013.R1

Article Status: SUBMITTED

MS Dryad ID: RSOS-190013.R1

MS Title: Fluctuating asymmetry and feather growth bars as biomarkers to assess habitat quality of shade coffee farming for avian diversity conservation

MS Authors: Gebremichael, Gelaye; Alemu, Diress ; Bunnefeld, Nils; Zinner, Dietmar ; Atickem, Anagaw

Contact Author: Gelaye Gebremichael

Contact Author Email: gelayegmd@gmail.com

Contact Author Address 1:

Contact Author Address 2:

Contact Author Address 3:

Contact Author City: Ghent

Contact Author State:

Contact Author Country: Ethiopia

Contact Author ZIP/Postal Code: 378

Keywords: bird community, bird species, Ethiopia, fluctuating asymmetry, ptilochronology

Abstract: Shade coffee farming has been promoted as a means of combining sustainable coffee production and biodiversity conservation. Supporting this idea, similar levels of diversity and abundance of birds have been found in shade coffee and natural forests. However, diversity and

abundance are not always good indicators of habitat quality because there may be a lag before population effects are observed following habitat conversion. Therefore other indicators of habitat quality should be tested. In this paper, we investigate the use of two biomarkers: fluctuating asymmetry (FA) of tarsus length and rectrix mass, and feather growth bars (average growth bar width) to characterise habitat quality of shade coffee and natural forests. We predicted higher FA and narrower feather growth bars in shade coffee forest versus natural forest, indicating higher quality in the latter. We measured and compared FA in tarsus length and rectrix mass and average growth bar width in more than 200 individuals of five bird species. The extent of FA in both tarsus length and rectrix mass was not different between the two forest types in any of the five species. Similarly, we found no difference in feather growth between shade coffee and natural forests for any species. Therefore, we conclude that shade coffee farming in Afrotropical rain forests of Ethiopia seems to have no negative consequences on avian populations for the species we examined.

EndDryadContent

Attachments area

Lines 30&31: I suggest instead of "no negative effects" something similar to what is stated on line 288: "...our comparison of biomarkers suggests that shade coffee farms and natural forests are of similar quality for the five species we examined." Just tone it down a bit, I don't think you can say there are "non negative consequences".... but your data suggests habitat quality is similar.

Our reply: Thanks, we did the change as suggested

Line 170: (such as x and x)

Our reply: okay we now use such as broken or deformed tarsi

Line 225: insert between habitat types

Our reply: we did the change as suggested

Line 226: replace fitted by fit.

Our reply: Done

Line 228: insert full stop next to forests, remove whereby and replace small u by capital u.

Our reply: Done

Line 235: insert between habitat types

Our reply: we did the change as suggested.

Line 251: I suggest stating "Directional asymmetry" again, for clarity instead of DA.

Our reply: Okay, we did that.

Line 257: is it 2?

Our reply: that is correct.

Line 260: I suggest these changes because the $p = 0.052$ threw me off. I thought these changes would clarify that for the most part, there was substantial evidence for no biological difference.

Our reply: Ok, that is true. Thanks.

Line 260: replace all by four, replace = by > and 0.0521 by 0.14.

Our reply: we did the change as suggested.

*Line 261: Cancel Abyssinian Ground Thrush,. Though in Abyssinian Ground thrush, the Fa 8in retriix mass was arguably stastically different between shade coffee and natural forest (p = 0.052), the effect size was near zero (beta= -9*10-4; table 3). (negative values mean lower FA in natural forest, right ?).*

Or reply: Yes.

Line 268: replace = by >, 1 by 14

Our reply: we did the change as suggested.

Line 269: replace = by >, 2 by 15

Our reply: we did the change as suggested.

Line 312: replace No by Similar to these other biomarker studies, we found no.....

Our reply: we did the change as suggested.

Line 313: insert a before negative, replace was observed from biomarker studies by “of shade coffee for the species we examined”.

Our reply: we did the change as suggested.

Line 314: replace has an important contribution for by “can contribute to”

Our reply: we did the change as suggested.

Line 315: insert in shade coffee agroecosystems. Replace are by “were” and next to disproportionally insert “negatively”.

Our reply: we did the change as suggested.

Line 316: Conversely, biomarkers for the two forest species in our study (Abyssinian Ground Thrush and Ruppell’s’ Robin-chat), were not demonstrably different between shade and natural forests.

Our reply: we did the change as suggested.

Line 316: cancel e and insert “a” and “of shade coffee” before and after negative effect, respectively.

Our reply: Okay we did that.

Line 317: replace monitoring by evaluating and insert avian biodiversity conservation value of before Ethiopia,, and remove for the avian biodiversity conservation.

Our reply: we did the change as suggested.

Line 318: delete the.

Our reply: Done.

Line 319: replace for the by “to”. Add change level to “levels”.

Our reply: we did the change as suggested.

Line 320: replace more comprehensive results in studies by “complementary measures.”

Our reply: we did the change as suggested.

Appendix G

Reviewer comments to Author:

Reviewer: 3

Comments to the Author(s)

The authors have addressed some of my initial concerns - for example they have confirmed that individual birds captured as juveniles were recaptured in subsequent years in the same locations, indicating that the biomarkers they measure (tarsal asymmetry reflecting nestling growth conditions, tail asymmetry reflecting stress on adults, and feather growth rates indicative of food availability) are accurately reflecting the conditions under the two habitats they are testing.

Our reply: Thank you!

They have not addressed my other concern - the validation of the biomarkers themselves. In their reply, they indicate they simply assume these markers are sensitive based on other studies. This is sufficient so long as they explicitly acknowledge this in the manuscript, but I also suggested a simple means of verifying this through a quick correlation between the markers to look for a negative association. The authors appear to have attempted this (Reply letter) with a bivariate pearson correlation, with the results being in the correct direction but not significant. While they indicate this appears in the paper, the line numbers they refer to don't include this analysis, nor did I find it mentioned in the entire paper.

Our reply: Thanks, we now included the correlation analysis in our updated version both in the method section and briefly in the discussion. We were hesitating if we should include this to the manuscript or explain this in the reply letter for the reviewer.

Despite this, the authors have qualified a bit of their language in the revised manuscript to acknowledge a bit more uncertainty of their results, particularly as their results failed to show an difference in bioindicators between samples from either habitat where others have found such differences (Buechley et al. 2015, L298).

However, without this validation, I feel that the paragraph (L315-323) recommending "future studies combine the use of both biomarkers to robustly asses the habitat quality" is overstated. This advocacy is problematic when you yourselves have assumed these are robust measures without actually testing or demonstrating this. I suggest simply omitting this paragraph.

Our reply: Thanks for this comment. We recommend the use of combining biomarkers to emphasize the need of using different traits as biomarkers varied a lot on their sensitivity for environmental changes. Including the correlation in our new version manuscript, we hope keeping this sentence is fine.

L290-294. Sentences are poorly constructed. I would suggest something like:

"As individual birds were initially banded as juveniles, and subsequently recaptured in later years as adults in the same sites, this suggests our biomarker measures (tarsal and retrice asymmetry and feather growth rates) should reflect conditions faced within our two specific regions - shade coffee vs natural forests".

I would also suggest this would probably be better to put into the Methods than leave to the Discussion.

Our reply: Ok, thanks. And we did as suggested.